# Evidence of the complexity of aerosol transport in the lower troposphere on the Namibian coast during AEROCLO-sA

Patrick Chazette[1], Cyrille Flamant[2], Julien Totems[1], Marco Gaetani[2,3], Gwendoline Smith[1,3], Alexandre Baron[1], Xavier Landsheere[3], Karine Desboeufs[3], Jean-François Doussin[3], and Paola Formenti[3]

[1]Laboratoire des Sciences du Climat et de l'Environnement (LSCE), Laboratoire mixte CEA-CNRS-UVSQ, UMR CNRS 1572, CEA Saclay, 91191 Gif-sur-Yvette, France
[2]LATMOS/IPSL, Sorbonne Université, CNRS, UVSQ, Paris, France
[3] Laboratoire Interuniversitaire des Systèmes Atmosphériques (LISA) UMR CNRS 7583, Université Paris-Est-Créteil, Université de Paris, Institut Pierre Simon Laplace, Créteil, France.

*Correspondence to*: Patrick Chazette (patrick.chazette@lsce.ipsl.fr)

**Abstract.** The evolution of the vertical distribution and optical properties of aerosols in the free troposphere, above stratocumulus, is characterized for the first time over the Namibian coast, a region where uncertainties on aerosol-cloud coupling in climate simulations are significant. We show the high variability of atmospheric aerosol composition in the lower and middle troposphere during the AEROCLO-sA field campaign (22 August - 12 September 2017) around the Henties Bay supersite, using a combination of ground-based, airborne and space-borne lidar measurements. Three distinct periods of 4 to 7 days are observed, associated with increasing aerosol loads (aerosol optical thickness at 550 nm ranging from ~ 0.2 to ~0.7), as well as increasing lofted aerosol layer depth and top altitude. Aerosols are observed up to 6 km above mean sea level during the later period. Aerosols transported within the free troposphere are mainly polluted dust (predominantly dust mixed with smoke from fires) for the first 2 periods (22 August-1 September 2017) and smoke for the last part (3-9 September) of the field campaign. As shown by Lagrangian back trajectory analyses, the main contribution to the aerosol optical thickness over Henties Bay is shown to be due to biomass burning over Angola. Nevertheless, in early September, the highest aerosol layers (between 5 and 6 km above mean sea level) seem to come from South America (southern Brazil, Argentina and Uruguay) and to reach Henties Bay after 3 to 6 days. Aerosols appear to be transported eastward by the mid latitude westerlies and towards Southern Africa by the equatorward moving cut-off low originating from within the westerlies. All the observations show a very complex mixture of aerosols over the coastal regions of Namibia that must be taken into account when investigating aerosols radiative effects above stratocumulus clouds in the south east Atlantic Ocean.

**Keywords:** dust, biomass burning aerosols, regional transport, atmospheric dynamics, back trajectories, lidar

## 1    Introduction

The western coast of southern Africa is a complex area in terms of both atmospheric composition, circulation, and climate, with aerosol-radiation-cloud interactions playing a significant role. A large part of this complexity is related to atmospheric circulation associated with a low-laying coastal strip next to an elevated continental plateau covering most of the sub-continent, as well as fast-evolving meteorological synoptic patterns largely controlled by

the St Helena anticyclone over the Atlantic and the mid-latitude westerlies on the poleward edge of this high-pressure system (Tyson and Preston-White, 2000).

The region is characterized by a complex aerosol composition linked to the variety of the sources. Biomass burning aerosols (BBA) regions over equatorial Africa (from both man-set fires and wild-fires) contribute to the regional and seasonal haze with the highest recorded aerosol optical thickness (Swap et al., 2003). Natural aerosols include i) mineral dust from point sources along the Namibian coast lines, as well as in the Etosha Pan in Namibia and in the Makgadikgadi Pan in Botswana (Ginoux et al., 2012; Vickery et al., 2013), and ii) marine sea spray and biogenic aerosols due to the strong productivity of the northern Benguela Upwelling System of the coast of Namibia (Andreae et al., 2004; Bates et al., 2001). Additional regional anthropogenic pollution is related to industrial emissions from South Africa and port activities in Namibia, together with ship emissions along the Namibian coast (Johansson et al., 2017).

The atmosphere over the coastal region of southern Africa is also characterized by a quasi-permanent stratocumulus deck, topping the marine boundary layer, and by a considerable thermodynamical stratification (Keil and Haywood, 2003), that limits the aerosol vertical mixing and exchange. Nevertheless, various authors (e.g. Diamond et al., 2018; Formenti et al., 2018; Zuidema et al., 2018) have provided evidence that BBA and dust aerosols emitted over the elevated continental plateau and transported in layers above the stratocumulus deck might penetrate and mix in the marine boundary layer (MBL). Others have also shown that the stratification of the aerosol layers over the south east Atlantic evolves with the distance from the coastline, increasing their ability to penetrate the stratocumulus deck (e.g. Adebiyi and Zuidema, 2016; Gordon et al., 2018).

Marine stratocumulus are particularly sensitive to aerosol perturbations due to relatively low background aerosol concentrations (Oreopoulos and Platnick, 2008). As a matter of fact, the vertical distribution of aerosols (and absorbing aerosols in particular) as well as their location with respect to bright low-level clouds (above or below) is of paramount importance as it significantly influences the indirect radiative effect (e.g. Ramanathan et al., 2007), the vertical profile of radiative heating in the atmosphere (e.g. Léon et al., 2002; Ramanathan et al., 2007; Raut and Chazette, 2008) and, in turn, the stability of the atmosphere, thereby modifying convective and turbulent motions and clouds (e.g. Ackerman et al., 2000; McFarquhar and Wang, 2006).

In this context, the coastal southern Africa region is arguably one of the regions where the aerosol-radiation-cloud interactions are strongest in the world (Adebiyi et al., 2015; Fuchs et al., 2017). However, state-of-the-art climate models diverge by several W m$^{-2}$ when attempting to calculate the regional direct radiative effect over coastal Southern Africa (Myhre et al., 2013; Stier et al., 2013) ranging from negative (-3 W m$^{-2}$) to strong positive forcing (+5 Wm$^{-2}$) for mean seasonal averages. These model shortcomings, that can also affect the simulation of climate features in distant areas (e.g., rainfall anomalies in Brazil, the position of the Intertropical Convergence Zone; Jones et al., 2009; Jones and Haywood, 2012), are mainly due to a limited knowledge of the aerosol properties, the vertical position of aerosol and cloud layers, and the distribution of cloud properties with and without aerosol present (Zuidema et al., 2016).

The main purpose of this article is to characterise the temporal and spatial evolutions of the vertical distribution of aerosol optical properties observed along the coastline of Namibia, in Henties Bay, in August and September 2017 during the Aerosols, Radiation and Clouds in southern Africa (AEROCLO-sA) field campaign (Formenti et al., 2019). The evolution of the vertical distribution of aerosols properties is examined as a function of the synoptic conditions and aerosol source emissions. The investigation is conducted by analysing a combination of ground-

based, airborne and space-borne lidar measurements, together with back-trajectory and numerical weather forecast model analyses, as well as complementary space-borne passive sensors observations.

Section 2 presents the observations and provides a description of the ground-based, airborne and space-borne active and passive remote sensing instruments used during the field campaign, together with complementary numerical simulation tools. Section 3 presents the evolution of the vertical profiles of aerosols during the campaign, together with the main optical and geometrical characteristics of the lofted aerosol layers and identifies three distinct periods with increasing aerosol load. The variability of the vertical distribution of aerosols around Henties Bay during the later period is assessed using lidar and dropsonde measurements acquired over the ocean, as detailed in Section 4. In Section 5, we investigate the different origins and transport pathways of aerosols in the free troposphere towards Henties Bay during the three periods. The last section is dedicated to the summary and conclusion. The description of the ground-based lidar is given in Appendix A, together with the calibration and data inversion processes.

## 2    Observations and simulations

The AEROCLO-sA supersite of Henties Bay (-22° 6' S, 14° 17' E, Figure 1) belongs to the Sam Nujoma Marine and Coastal Resources Research Centre (SANUMARC) of the University of Namibia in the Orongo region. It has been selected because of its geographical position: bounded by the Atlantic Ocean on its western side and by the Namib desert, ~800 m above the mean sea level (AMSL), on its eastern side (Formenti et al., 2019). The analysis presented here relies mainly on active and passive remote sensing observations acquired from i) ground-based instruments deployed in Henties Bay, namely an Aerosol Lidar System (ALS) 450® (Leosphere Inc, Saclay, France) operating at a wavelength of 355 nm and a sun photometer from the National Aeronautics and Space Administration Aerosol Robotic Network (AERONET), ii) the airborne lidar LEANDRE (Lidar Embarqué pour l'Etude des Aérosols, Nuages, Dynamique, Rayonnement et Espèces minoritaires) nouvelle Génération (LNG), working in the Rayleigh-Mie scattering mode, installed on the Service des Avions Français Instrumentés pour la Recherche en Environnement (SAFIRE) Falcon 20 and iii) space-borne instruments, namely the Cloud-Aerosol Lidar with Orthogonal Polarization (CALIOP), the Cloud-Aerosol Transport System (CATS) lidar and the Moderate-Resolution Imaging Spectroradiometer (MODIS). The available measurements are summarized in **Table 1** against the date and the universal time count (UTC). The synergy between ground-based lidar measurements, space-borne observations (aerosol typing and aerosol optical thickness (AOT)) and those of the sun photometer (AOT and Ångström exponent) is used to better constrain the retrieval of the aerosol optical parameters (see Appendix A): aerosol extinction coefficient (AEC), lidar ratio (LR) and particle depolarisation ratio (PDR). The space-borne lidar-derived aerosol types are associated with prescribed LRs (see Section 2.4) that are used for the inversion of the ground-based lidar.

Table 1: Data available during the field campaign on August and September 2017 from: the ground-based ALS lidar and AERONET sun photometer in Henties Bay, the airborne LNG lidar, dropsonde released from the Falcon 20, as well as the CATS and CALIOP space-borne lidars. The line highlighted in bold indicates when the AERONET inversion allows the retrieval of a relevant value for the lidar ratio (level 2 data). The aerosol typing as provided by CALIOP and CATS is also indicated for overpasses in the vicinity of Henties Bay.

| Date | ALS measurement | F20 flight LNG & dropsonde | Coupling ALS/ | CALIOP Orbit close to the site | CATS |
|---|---|---|---|---|---|

| | time (UTC) | measurement time (UTC) | **AERONET** | | **Overpass time (UTC)** |
|---|---|---|---|---|---|
| 22 Aug | 1400-2300 | - | Yes | - | - |
| 23 Aug | 1645-2330 | - | Yes | - | 0342-0357 Smoke |
| 27 Aug | 1545-1700 | - | Yes | - | - |
| 28 Aug | 1030-1230 | - | Yes | 10.2017-08-28T00-08-17ZN 10.2017-08-28T12-26-48ZD Polluted dust/Smoke | - |
| 29 Aug | 1730-2250 | - | No | 10.2017-08-29T23-55-43ZN Smoke | 0122-0207 Smoke |
| 30 Aug | 1800-2000 | - | No | | 0047-0102 Smoke |
| 31 Aug | 1430-2100 | - | Yes | 10.2017-08-31T12-57-28ZD Smoke/Polluted dust | 1452-1507 Smoke/Dust |
| 02 Sep | 0930-1130 1715-1900 | - | Yes | 10.2017-09-02T12-44-54ZD Smoke/Polluted dust | - |
| **03 Sep** | **1400-1540** | - | Yes | - | - |
| 04 Sep | 2330-2400 | - | No | 10.2017-09-04T00-13-44ZN Smoke | - |
| 05 Sep | 1400-1500 | Flight 6 LNG: ~1000 Dropsonde #5: 0952 | No | - | 2204-2219 Smoke |
| 06 Sep | 0830-1030 | Flight 8 | Yes | - | 1258-1313 |

| | | LNG: ~0830 and ~0900<br><br>Dropsondes #3 and #4: 0843 and 0908 | | | Smoke/dust |
|---|---|---|---|---|---|
| 07 Sep | 1600-1900 | - | No | - | 2156-2211<br>Smoke |
| 08 Sep | 1300-1500 | - | No | - | 2052-2107<br>Smoke |
| 09 Sep | 0900-1200 | - | Yes | - | 2001-2016<br>Smoke |
| 11 Sep | 1040-1140 | - | Yes | - | - |

114

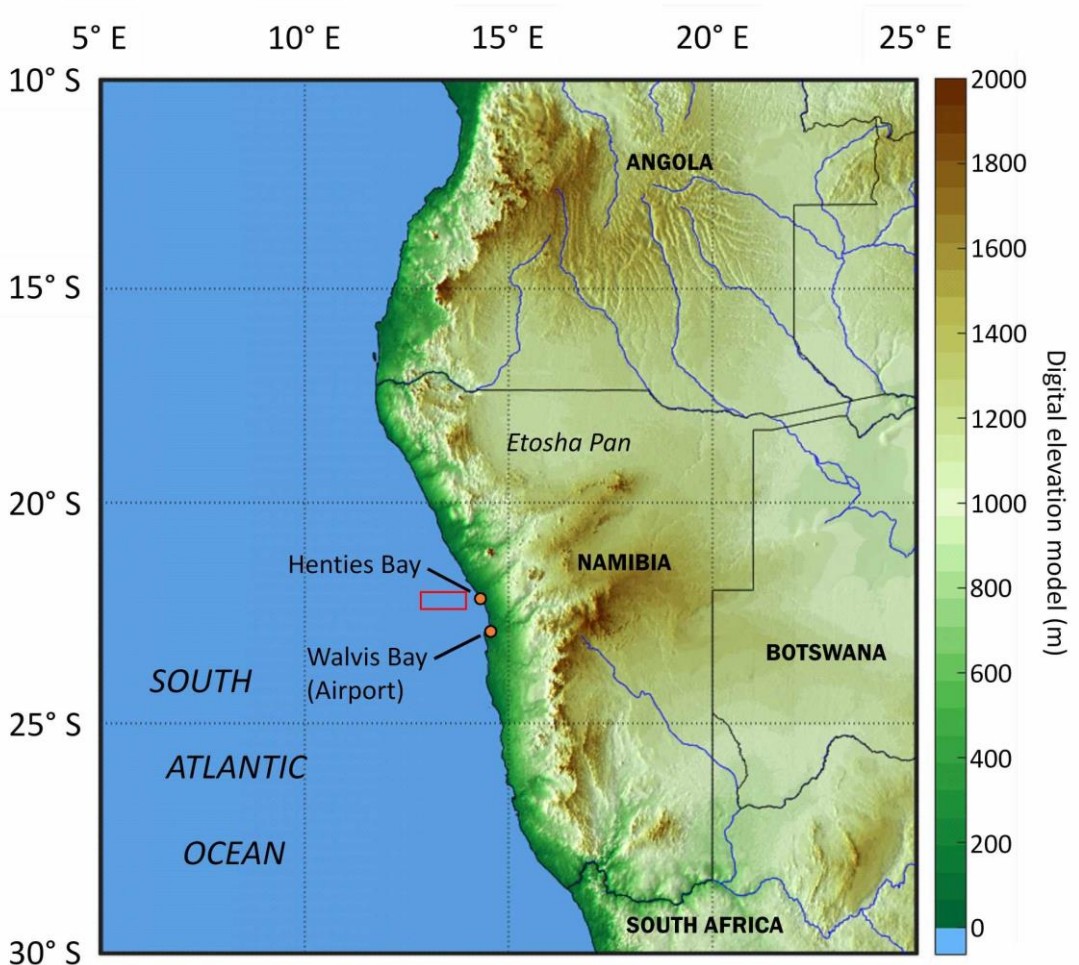

115

Figure 1: Location of the Henties Bay experimental site (in Namibia) on the west African coast. The Walvis Bay airport where the SAFIRE Falcon 20 aircraft operated during AEROCLO-sA is also indicated. The black rectangle surrounds the area chosen to average the MODIS-derived AOTs. The Henties Bay and Walvis Bay locations are marked by orange dots.

## 2.1 Ground-based lidar

The ALS lidar measurements were carried out continuously between 22 August and 13 September, 2017. The data coverage for aerosol study is low because of the quasi-ubiquitous presence of marine stratocumulus and fog during a large part of the observation days. The fog opacity was often such that the laser beam was fully attenuated after a few hundred meters. We therefore considered average profiles taken during periods when no low-level clouds or fog events are observed, i.e. between about 1 and 4 hours on a given day (see **Table 1**). The description of the lidar is given in Appendix A, together with the calibration and data inversion processing.

## 2.2 AERONET sun photometer

The site of Henties Bay was equipped with a sun and sky scanning spectral radiometer manufactured by CIMEL Inc (Paris, France) and belonging to the AERONET automatic and global network of sun photometers providing long-term and continuous monitoring of aerosol optical, microphysical and radiative properties (http://aeronet.gsfc.nasa.gov/). Eight spectral bands are generally used between 340 and 1020 nm. The aerosol optical thickness at the lidar wavelength of 355 nm ($AOT_{355}$) is assessed using the Ångström exponent (Ångström, 1964) and the sun photometer AOT at 380 and 440 nm (e.g. Hamonou et al., 1999). We use level 2.0 (cloud screened and quality-assured) aerosol optical thickness (AOT) data in the following. The total uncertainty on AOT is $<\pm0.01$ for $\lambda>440$ nm and $<\pm0.02$ for $\lambda<440$ nm (Holben et al., 1998). Nevertheless, additional bias may exist when thin clouds are present and not screened in the AERONET level-2 products (Chew et al., 2011). To limit this, ground-based lidar profiles are used to identify the presence of clouds when sun photometer observations are available.

## 2.3 Airborne measurements

In this study, we also analyse extinction coefficients over the Atlantic, and in the vicinity of Henties Bay, acquired with the LNG Lidar (Bruneau et al., 2015) flown on the SAFIRE Falcon 20 on 5 and 6 September. We only use the 532 nm channel because the high level of noise in the high spectral resolution 355 nm channel. Hence, the lidar was operated as a simple backscatter Rayleigh-Mie lidar. The Falcon 20 operated from Walvis Bay, on the western coast of Namibia, roughly 100 km south of Henties Bay where the AEROCLO-sA supersite was located. Details on the Falcon payload as well as the on the flight plans conducted during these two days can be found in Formenti et al. (2019). In addition to the LNG data, we also make use of dynamical and thermodynamical data acquired offshore of Namibia with the Vaisala dropsonde system.

During the first flight (flight #6 in the morning of 5 September 2017), the Falcon operated from 0736 to 1014 UTC. It flew mostly above the continent to monitor dust emissions over the Etosha pan (see Formenti et al., 2019). The later portion of the flight was conducted over the sea (from 0930 to 1014 UTC), and a dropsonde was launched from 13.78°E/ 21.69°S at 0952 UTC. For the second flight (flight #9 in the morning of 6 September 2017), the Falcon 20 operated from 0703 to 0927 UTC and flew over the ocean from 0820 to 0927 UTC. Two dropsondes were launched from 11.92°E / 19.87°S at 0843 UTC and from 13.41°E / 22.23°S at 0908 UTC.

The LNG data over the sea are inverted using the same procedure as for the ground-based ALS lidar (see Appendix A) and utilizing the same LR vertical distribution (see values retrieved in Henties Bay for the two days in Section 3).

**2.4    Spaceborne observations**
**2.4.1    CALIOP & CATS**
The Cloud-Aerosol LIdar with Orthogonal Polarization (CALIOP) has been flying onboard the Cloud-Aerosol
Lidar Pathfinder Satellite Observation (CALIPSO) since 2006 (https://www-calipso.larc.nasa.gov/products/).
Details on the CALIOP instrument, data acquisition, and science products are given by Winker et al. (2007). In
this work, we use CALIOP level-2 data, version 4.10 (Kim et al., 2018), which was corrected for aerosol typing,
as noted in Burton et al. (2012). The aerosol types identified in the free troposphere (FT) are typically polluted
dust and elevated smoke (see example in Appendix A).
The CATS lidar orbited between 375 and 435 km onboard the non-sun-synchronous International Space Station
(Yorks et al., 2016). It operated between January 2015 and October 2017 with the objective of measuring some
cloud and aerosols properties which are useful for climate study. CATS flew over Namibia at various times during
the AEROCLO-sA field campaign (**Table 1**). We mainly used the aerosol typing derived from CATS
measurements, which is similar to the one established for CALIOP. The correspondence between the aerosol
typing derived from CALIOP and CATS measurements are given in the **Table 2**. It should be noted that not all
the aerosol types are named exactly in the same way. An example of aerosol typing is given in Appendix A.

Table 2: Lidar ratio (LR) corresponding with the CATS- and CALIOP-derived aerosol typing.

| CALIOP/CATS Aerosol typing | Lidar ratio (sr) at 532 nm |
|---|---|
| Polluted continental or smoke/Polluted continental | 70/65 |
| Clean continental/Clean-background | 53/55 |
| Clean marine/Marine | 23/25 |
| Dust/Dust | 44/45 |
| Polluted dust/Dust mixture | 55/35 |
| Elevated smoke/Smoke | 70/70 |
| Dusty marine/Marine mixture | 37/45 |


**2.4.2    MODIS**
The MODIS instruments (King et al., 1992; Salmonson et al., 1989) are aboard the Aqua and Terra platforms
(http://modis-atmos.gsfc.nasa.gov). The polar orbit of Terra (http://terra.nasa.gov) passes over the equator from
north to south in the morning, whereas Aqua (http://aqua.nasa.gov) has its ascending node over the equator during
the afternoon. They provide a complete coverage of the Earth surface in one to two days with a resolution between
250 and 1000 m at ground level depending on the spectral band. We use the Terra and Aqua AOT at 550 nm from
the MODIS aerosol product level-2 data. Both products are given with a spatial resolution of $10 \times 10$ km$^2$ at nadir.
The uncertainty in the AOT retrieval (Remer et al., 2005) over land (ocean) is $0.15 \pm 0.05 AOT$ ($0.05 \pm 0.03 AOT$).
We will only use data over the sea because Henties Bay is a coastal site affected by the sea breeze and bordered
by a strong topography (Figure 1). This is associated with the lowest levels of uncertainty. The thermal anomalies
derived from the MODIS fire product (e.g. Ichoku et al., 2008) are also used
(https://modis.gsfc.nasa.gov/data/dataprod/mod14.php).
**2.5    Modelling**
The meteorological patterns are studied using Meteorological fields provided by the 6-hourly operational analyses
of the European Centre for Medium-Range Weather Forecasts (ECMWF, http://apps.ecmwf.int/datasets/, Dee et
al. (2011)). We also use the near real time analyses of atmospheric dynamics and aerosols from the Copernicus
Atmosphere Monitoring Service (CAMS, https://atmosphere.copernicus.eu/). The calculations for synoptic
analysis are computed on a 0.75-degree horizontal regular grid. Daily means are computed by averaging time steps
at 03:00, 09:00, 15:00 and 21:00 UTC of daily forecasts initialised at 00.00 UTC. For local analyses, the
meteorological wind fields are computed by using 1-h data on a 0.25-degree horizontal regular grid from the Fifth
ECMWF       Reanalysis     (ERA5,      https://www.ecmwf.int/en/forecasts/datasets/archive-datasets/reanalysis-
datasets/ERA5, Hoffmann et al., 2018). The back trajectories analyses are based on the Hybrid Single Particle
Lagrangian Integrated Trajectory (HYSPLIT) model (Draxler and Rolph, 2014; Stein et al., 2015). The wind fields
used as input from the HYSPLIT model are from GDAS (Global Data Assimilation System,
http://www.ncep.noaa.gov/) at 0.5° horizontal resolution. The isentropic ensemble mode with 24 individual back
trajectories is used to take into account the transport trajectory spread associated with the wind field variability
around the trajectories starting point. Using different modelling approaches also allows the consistency of results
to be verified.
**3    Temporal evolution of the aerosol properties and vertical distribution over Henties Bay**
**3.1    Identification of periods from the total AOT**
The temporal evolution of the AOT at 550 nm derived from passive remote sensing observations (MODIS and the
Henties Bay sun photometer) and 6-hourly CAMS fields between 22 August and 9 September 2017 are shown in
**Figure 2**a. For CAMS, both the AOT extracted from the grid cell centred on Henties Bay and the average AOT
calculated on a 3x3 grid-point box surrounding the site are shown. There are little differences between the two
CAMS-derived AOTs, which highlight the homogeneity of aerosol plumes overpassing Henties Bay according to
the model and during that period. The MODIS AOT at 550 nm plotted in Figure 2a is a daily synthesis of Terra
and Aqua products extracted over the sea only (see the black rectangle in Figure 1), to avoid mixing the effects of
coast, topography and surface albedo in the AOT retrievals. Overall, the AOTs from CAMS match within 0.1 the
ones derived from both MODIS and the sun photometer, except on 2 September and 7-8 September. These
discrepancies on AOT may be also explained by the coarse spatio-temporal sampling of the model, which is
insufficient to highlight the sharp variation in AOT due to a very localized aerosol features during these 3 days.
As a result, even small differences in the simulation of the weather conditions could lead to substantial differences
in AOT for specific locations, especially when AOT values are rather low. Note that no significant precipitation
event was recorded during the field campaign, so that we can exclude any CAMS misrepresentation of wet
deposition processes around Henties Bay. In addition, CAMS simulations show that the AOT is essentially due to
organic matter (i.e. biomass burning aerosols), the contribution from non-biomass aerosol can then be excluded as
well.  On 2 September a minimum in AOT is observed by the sun photometer which is not reproduced by CAMS
simulations (even though a local minimum in the CAMS AOT can be seen). During this day, the mid-tropospheric
circulation was characterised by a low-pressure system located offshore of Henties Bay, juxtaposed to a high-
pressure system over South Africa, resulting in a small river of smoke descending along the coast that CAMS is
simulating too far east over Henties Bay. On 7-8 September, the sun photometer- and MODIS-derived AOTs are
larger than the one computed from CAMS. This could be related to the presence of unscreened optically thin clouds
such as the ones observed in the ground-based lidar data on 8 September (Figure A2d) and/or to the heterogeneity
of the meteorological field. Indeed, on 7-8 September, an elongated high pressure dominating over the continent,
led to the channelling of the smoke from the north-west that is slightly mis-located in the CAMS analyses.
In **Figure 2**a, three distinct periods can be identified based on the temporal evolution of both the remote sensing
instruments and the CAMS-derived AOT. The optical and geometrical properties of the aerosol layers derived
from the remote sensing instruments over Henties Bay during the 3 periods are summarized in **Table 3**. The first
period $P_1$ (22-28 August 2017, see **Figure 2**a) is characterized by an averaged AOT of ~0.20 at 550 nm, while for
the second period $P_2$ (28 August – 1 September 2017, see **Figure 2**a) the AOT increases to ~0.4. During the third
period $P_3$ (3-11 September 2017), the average AOT is higher than during $P_2$ and around 0.55 at 550 nm (see **Figure
2**). 2 September can be considered as a transition period between $P_2$ and $P_3$. The variability of the CAMS-derived
AOT is much larger during $P_3$ than during $P_1$ and $P_2$ which may show greater variability in atmospheric transport
conditions. The sunphotometer derived Angstrom exponent (AE) evolves during the period of interest, with AE~1
during $P_1$ et AE~1.4 during $P_2$ and $P_3$ (see **Table 3**), suggesting the presence of larger aerosol in the atmospheric
column during $P_1$.

## 3.2 Aerosol vertical profiles

The AEC profiles shown in Figures 3 to 7 are obtained in cloud free conditions using a standard inversion
procedure detailed in Appendix A. Most AEC profiles show clear air with low particle concentrations between the
planetary boundary layer (PBL) and the elevated aerosol layer, with the notable exception of 2 September in the
afternoon, when aerosols are mainly observed in the PBL (Figure 5b). Figure 2b shows the AOTs at 355 nm
calculated from the lidar-derived AEC profiles between the surface and ~6.5 km AMSL, as well as partial column
AOTs in the FT for three different altitude ranges where aerosol loads can be highlighted: namely [1500-3000[,
[3000-5000[ and [5000 6000[ m (green, grey and red bars in Figure 2b, respectively). The temporal evolution of
the partial column AOTs corroborate the existence of the 3 periods. During $P_3$, we observe AOTs in excess of 0.1
between 5000 and 6000 m AMSL for at least 4 days (3, 6, 7 and 11 September) whereas partial AOTs in that
height range are negligible in the previous two periods. AOT values as high as 0.4 are observed on 6 September.
The increase in the lidar-derived column AOT (blue bars in Figure 2b) during $P_3$ is also well correlated to the
increase of the partial column AOT in the 1500-3000 m AMSL.
We note a significant increase in terms of the lidar-derived thickness of elevated aerosol layer between the 3
periods (~1-2.5 km during $P_1$, ~2.5-3 km during $P_2$ and ~2.5-5 km during $P_3$, **Table 3**) as well as in terms of
maximum AEC in the FT (~0.1 km$^{-1}$ during $P_1$, ~0.25 km$^{-1}$ during $P_2$ and ~0.3 km$^{-1}$ during $P_3$, **Table 3**) as seen in
the AEC profiles (compare **Figure 3** for $P_1$ with **Figure 4** for $P_2$). The height of the base of the elevated aerosol
layer also increases between $P_1$ and $P_2$, from ~1-1.5 km AMSL to more than 2 km AMSL (Table 3), but appears
more variable during P3 (from ~1 to 3 km AMSL, **Figure 6** and **Figure 7**). These changes in optical and
geometrical properties of the aerosols in the FT are related to the variability of long-range transport over the area,
as discussed in Section 5.
CALIPSO and CATS retrievals suggest differences in the FT aerosols between $P_1/P_2$ and $P_3$, with more occurrence
of polluted dust (55 sr) in $P_1/P_2$ and polluted continental or smoke (70 sr) in $P_3$. In the PBL, during $P_1/P_2$, the
retrieved low value of LR (i.e. 23 sr) required to reproduce the sunphotometer AOT is consistent with the presence
of clean marine aerosols in the PBL (e.g. Flamant et al., 1998). The retrieved higher LRs required in $P_3$ indicate
the presence of other aerosol types, which may include smoke (i.e.70 sr) or a mixture of smoke and terrigenous
aerosols (i.e. 55 sr). The latter LR value suggests the presence of terrigenous aerosols mixed with smoke,
corresponding to the aerosol typing "Polluted Dust". During $P_3$, aerosols in the FT are mainly identified as "smoke"
(based on the CALIOP and CATS typing). Very few sun photometer data are available for LR retrieval due to the
quasi permanent presence of a cloud cover over Henties Bay during the cycles of almucantar measurements.
Nevertheless, such a measurement could be obtained during $P_3$, on 3 September 2017 at ~14:10 UTC. A sun
photometer-derived LR of ~63 sr at 532 nm has been computed from the backscatter phase function and the single
scattering albedo (Dubovik et al., 2000). It was found to match the LR associated with the smoke type of CALIOP
and CATS (i.e. 65-70 sr at 532 nm).
The PDR is computed for each AEC profile given in Figures 3 to 7. The PBL is associated with the lower PDR
(i.e. < 2-3%), mainly during $P_1$ and $P_2$. This argues for the presence of hydrophilic spherical particles as marine
aerosols. Within the free troposphere the PDR is higher, mainly between 5 and 10% and may correspond to a
mixing of biomass burning and dust aerosols as often observed in biomass burning aerosol plume over others areas
(e.g. Chazette et al., 2015; Kim et al., 2009). This is consistent with the hypothesis of dust mobilization and mixing
by convection in biomass burning regions. Above the PBL larger PDR can be observed and may indicate a higher
relative presence of dust. This should be taken with caution as AEC values are low for these layers and uncertainties
are therefore higher.

Table 3. Properties of aerosol layers above the Henties Bay site as derived from the ground-based lidar, CALIOP, CATS, the
sun photometer and MODIS: lidar ratios for the free troposphere ($LR_{FT}$) and the planetary boundary layer $LR_{PBL}$ at 532 nm,
ground-based lidar (GBL)-derived $AOT_{GBL}$ at 355 nm and its uncertainty (detection noise and atmospheric variability),
sunphotometer-derived $AOT_{phot}$ at 355 nm and 550 nm, sunphotometer-derived Ångström exponent (AE), MODIS-derived
$AOT_{MODIS}$ in 0.5°x0.5° area over the sea close to Henties Bay, free troposphere aerosol layer (FTA) thickness and bottom
height and maximum of the aerosol extinction coefficient ($AEC_{max}$) in the UAL. $P_1$ and $P_2$ correspond to periods when the AFT
is mostly composed of "polluted dust", and $P_3$ corresponds to period when smoke aerosols dominate the composition of the
UAL.

| Date UTC | $LR_{FT}$ $LR_{PBL}$ (sr) | $AOT_{GBL}$ at 355 nm | $AOT_{phot}$ at 355 nm *at 550 nm* | AE | $AOT_{MODIS}$ 550 nm 0.5°x0.5° | FTA width (km) | FTA bottom height (km) | $AEC_{max}$ in the FTA (km$^{-1}$) |
|---|---|---|---|---|---|---|---|---|
| *Period $P_1$* | | | | | | | | |
| 22/08 1400-2300 | **55** 23 | 0.36±0.02 | 0.37±0.02 *0.22±0.01* | 1.15±0.15 | 0.26±0.03 | ~1 | ~1.5 | ~0.15 |
| 23/08 | **55** 23 | 0.31±0.03 | 0.34±0.01 *0.22±0.01* | 0.95±0.05 | 0.23±0.03 | ~1.5 | ~1 | ~0.1 |

| | | | | | | | | |
|---|---|---|---|---|---|---|---|---|
| 1645-2330 | | | | | | | | |
| 27/08 1545-1700 | **55** 23 | 0.32±0.01 | 0.33 *0.18* | 1.27 | Clouds | ~2.5 | ~1.5 | ~0.1 |
| *Period P₂* | | | | | | | | |
| 28/08 1030-1230 | **55** 23 | 0.63±0.03 | 0.59±0.04 *0.24±0.04* | 1.5±0.05 | 0.25±0.12 | ~3 | ~2 | ~0.2 |
| 29/08 1730-2250 | **55** 23 | 0.60±0.02 | - | - | Clouds | ~2 | ~3 | ~0.2 |
| 30/08 1800-2000 | **55** 23 | 0.82±0.04 | - | - | 0.30±0.05 | ~2.5 | ~2.3 | ~0.3 |
| 31/08 1430-2100 | **55** 23 | 0.83±0.01 | 0.85±0.02 *0.42±0.08* | 1.4±0.04 | 0.44±0.05 | ~2.5 | ~2.5 | ~0.3 |
| *Transition period* | | | | | | | | |
| 02/09 0930-1130 | **37** 18 | 0.32±0.02 | 0.28±0.03 *0.19±0.02* | 0.9±0.1 | Clouds | ~2 | ~2.5 | < 0.1 |
| 02/09 1715-1900 | **37** 18 | 0.16±0.01 | - | - | - | ~0.9 | ~0.5 | < 0.1 |
| *Period P₃* | | | | | | | | |
| 03/09 1400-1540 | **70** 70 | 1.19±0.05 | 1.21±0.02 *0.65±0.01* | 1.43±0.02 | Clouds | ~5 | ~1.2 | ~0.25 |
| 04/09 2330-2400 | **70** 70 | 0.84±0.02 | - | - | Clouds | ~3.5 | ~1.2 | ~0.25 |
| 05/09 1400-1500 | **70** 55 | 0.92±0.09 | - | - | Clouds | ~2.8 | ~1.8 | ~0.35 |
| 06/09 0830-1030 | **70** 55 | 1.33±0.12 | 1.34±0.06 *0.70±0.05* | 1.50±0.04 | 0.56±0.11 | ~3.2 | ~2.8 | ~0.4 |
| 07/09 | **70** 55 | 1.31±0.11 | 1.30±0.04 *0.68±0.02* | 1.46±0.01 | 0.74±0.03 | ~3.3 | ~2.5 | ~0.3 |

| 1600-1900 | | | | | | | | |
|---|---|---|---|---|---|---|---|---|
| 08/09 1300-1500 | **70** **70** | 0.94±0.10 | 1.87 *1.01* | 1.4 | 0.74±0.08 | ~3 | ~1.2 | ~0.25 |
| 09/09 0900-1200 | **70** 70 | 1.04±0.06 | 1.41±0.09 *0.75±0.01* | 1.44±0.01 | 0.69±0.12 | ~4 | ~1 | ~0.3 |
| 11/09 1040-1140 | **70** 70 | 0.70±0.12 | 0.86 *0.41* | 1.68 | Clouds | ~4.9 | ~0.8 | ~0.25 |


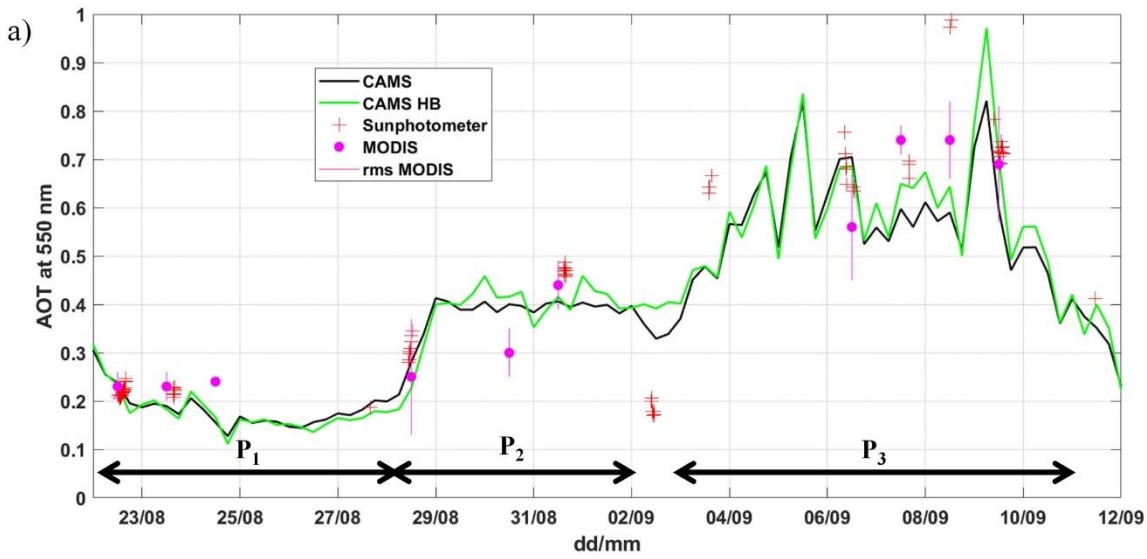

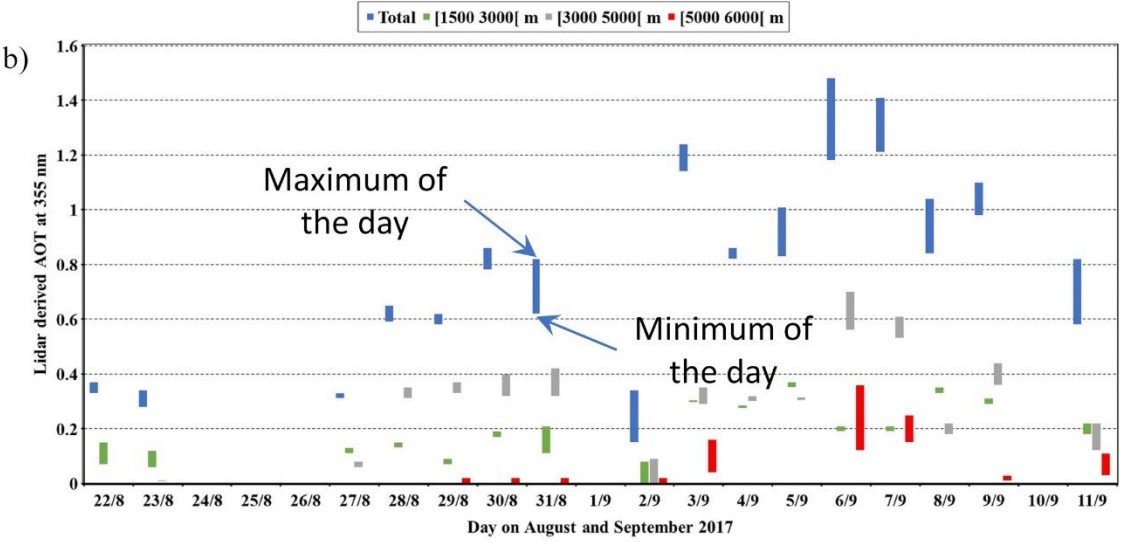


Figure 2: a) Temporal evolution of the AOT at 550 nm derived from CAMS (black and green solid lines), sun photometer (red
crosses) and MODIS (magenta dots) data. The green solid line shows CAMS AOT extracted on the grid cell centred on Henties
Bay. The black solid line shows the CAMS AOT averaged over 9 grid cells (a 3x3 grid box) centered on Henties Bay. The 3
periods highlighted by the AOT values ($P_1$, $P_2$ and $P_3$) are indicated. b) Temporal evolution of the lidar-derived AOT at 355
nm for the altitude ranges [1500 3000[ m in green, [3000 5000[ m in grey and [5000 6000[ m in red. The total AOT is given in
blue. The vertical bars delimit the daily extremes of AOT.

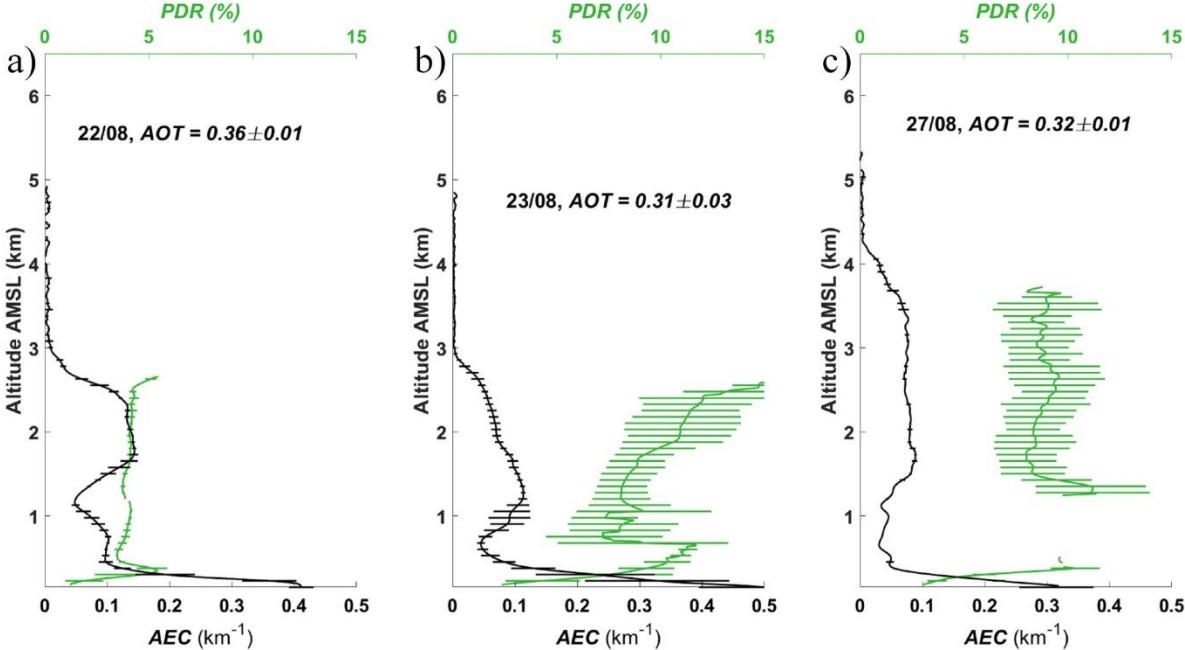


Figure 3: Vertical profiles of the aerosol extinction coefficient (AEC) and particle depolarization ratio (PDR) at 355 nm with
their uncertainties (horizontal bars) for Period $P_1$: on a) 22 (1400-2300 UTC), b) 23 (1645-2330 UTC) and c) 27 (1545-1700
UTC). The total aerosol optical thickness at 355 nm (AOT) is also given for each profile with its uncertainty.

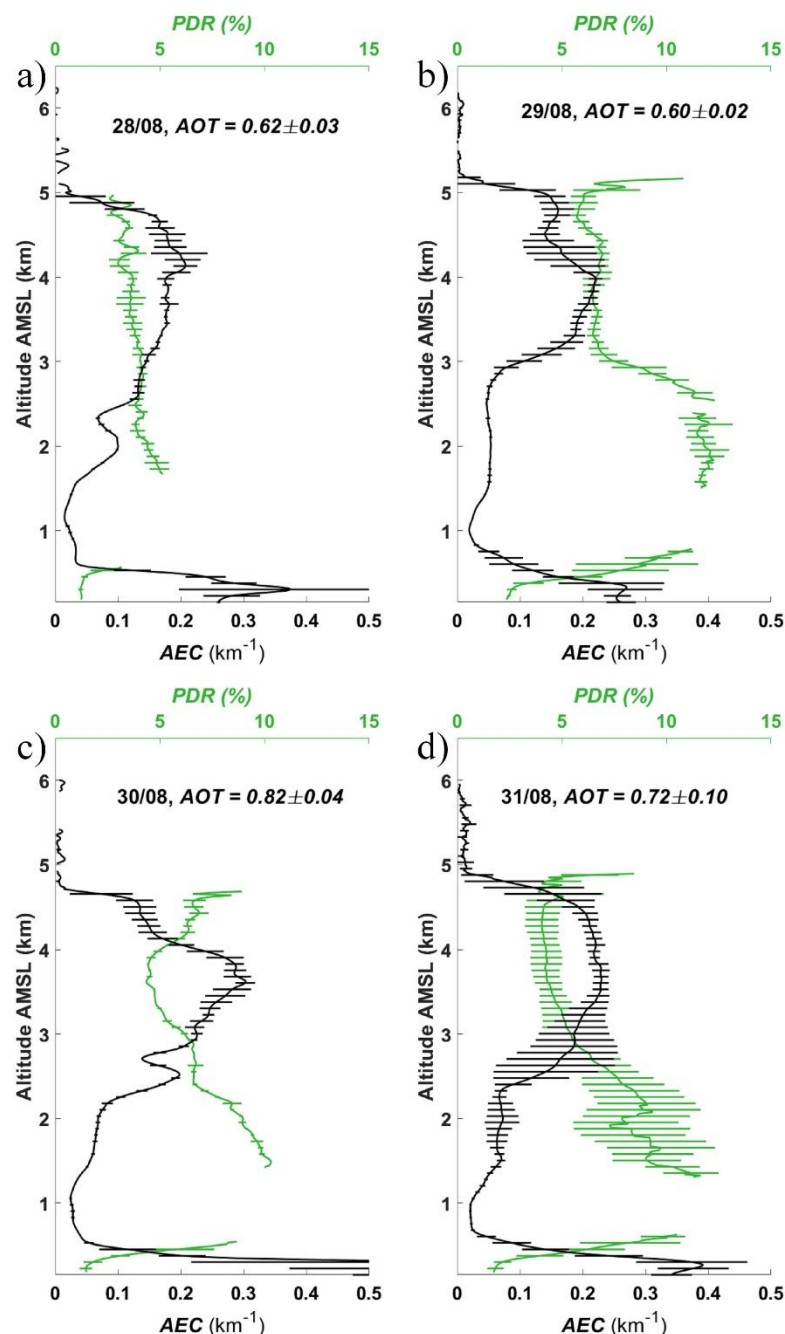


Figure 4: Vertical profiles of the aerosol extinction coefficient (AEC) and particle depolarization ratio (PDR) at 355 nm with
their uncertainties (horizontal bars) for Period P2: on a) 28 (1030-1230 UTC), b) 29 (1730-2250 UTC), c) 30 (1800-2000 UTC)
and d) 31 (1430-2100 UTC) August 2017. The total aerosol optical thickness at 355 nm (AOT) is also given for each profile
with its uncertainty.

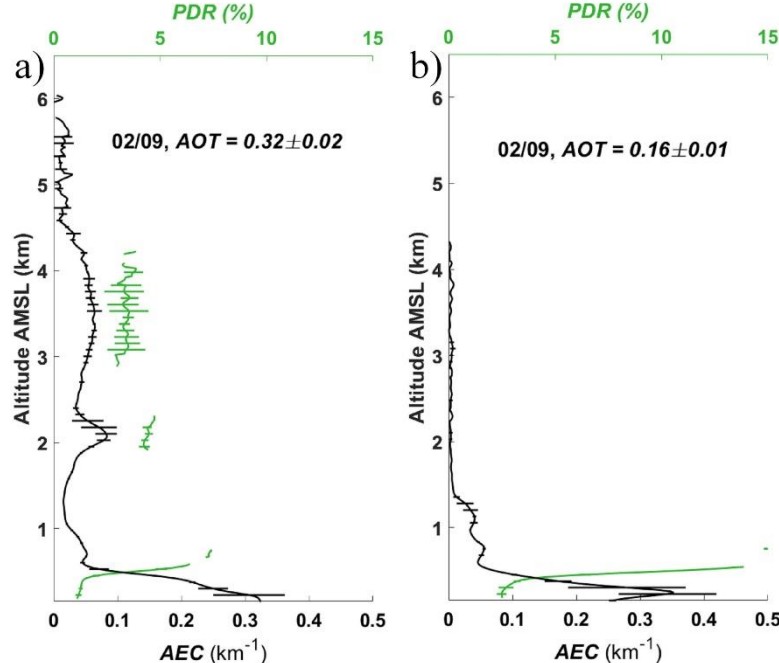


Figure 5: Vertical profiles of the aerosol extinction coefficient (AEC) and particle depolarization ratio (PDR) at 355 nm with
their uncertainties (horizontal bars) for the transition period on 2 September 2017 at a) 0930-1130 UTC and b) 1715-1900
UTC. The total aerosol optical thickness at 355 nm (AOT) is also given for each profile with its uncertainty.

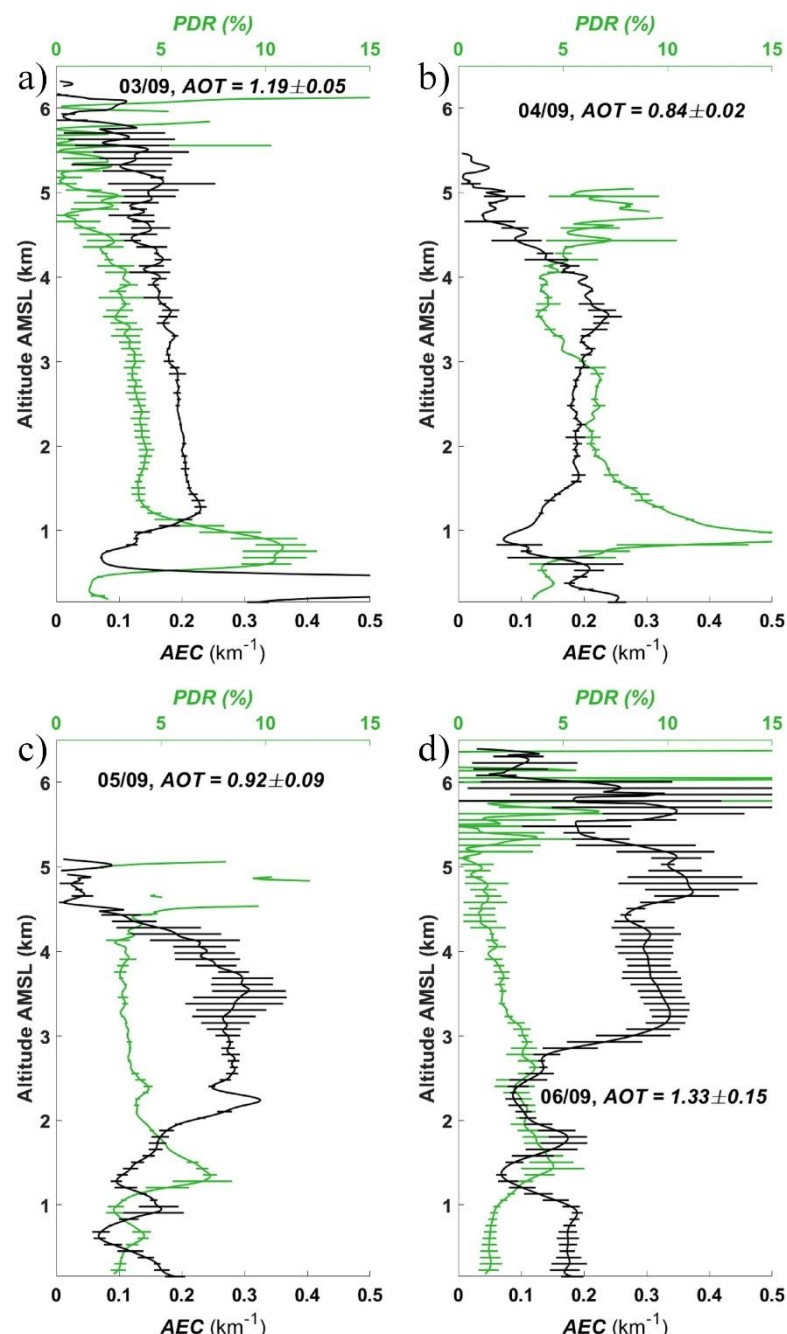



Figure 6: Vertical profiles of the aerosol extinction coefficient (AEC) and particle depolarization ratio (PDR) at 355 nm with
their uncertainties (horizontal bars) for Period P₃: on a) 3 (1400-1540 UTC), b) 4 (2330-2400 UTC), c) 5 (1400-1500 UTC)
and d) 6 (0830-1030 UTC) September 2017. The total aerosol optical thickness at 355 nm (AOT) is also given for each profile
with its uncertainty.

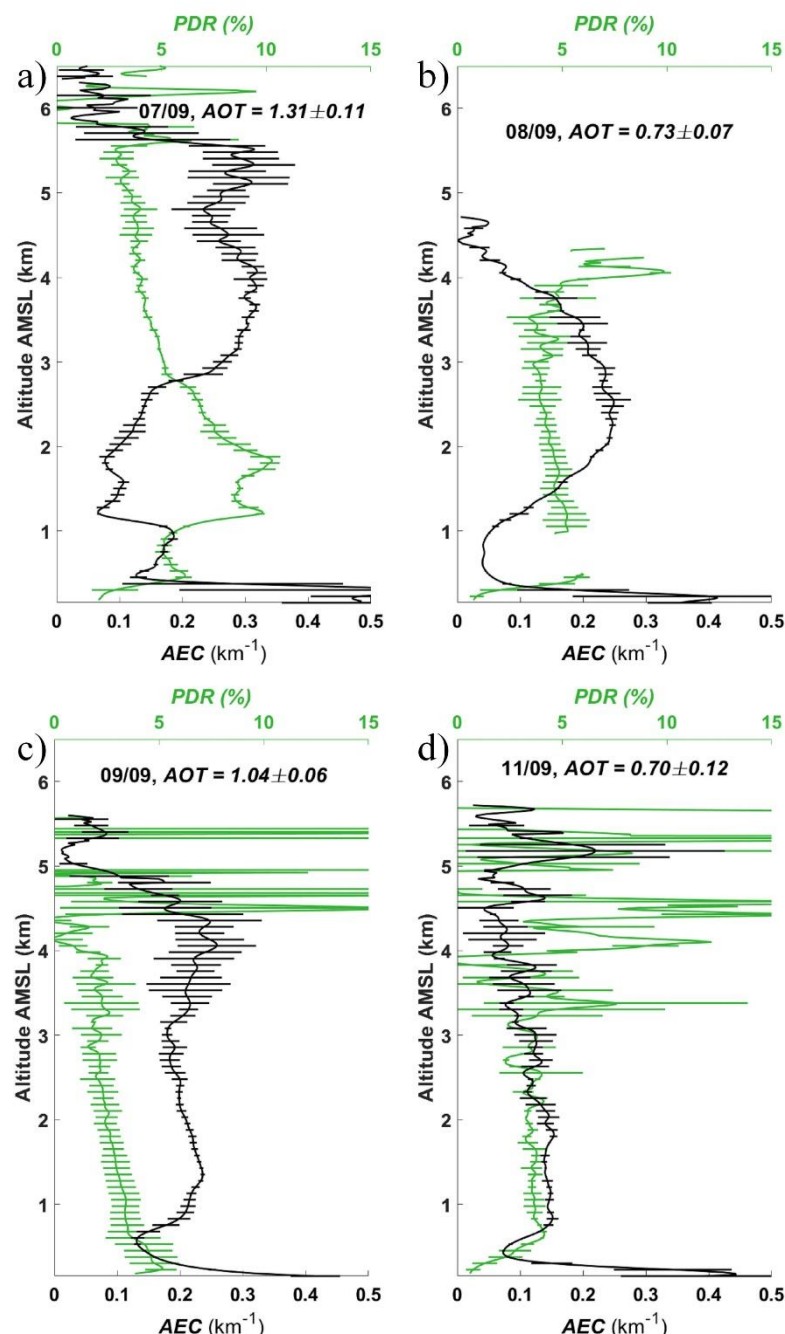


Figure 7: Vertical profiles of the aerosol extinction coefficient (AEC) and particle depolarization ratio (PDR) at 355 nm with
their uncertainties (horizontal bars) for Period $P_3$: on a) 7 (1600-1900 UTC), b) 8 (1300-1500 UTC), c) 9 (0900-1200 UTC)
and d) 11 (1040-1140 UTC) September 2017. The total aerosol optical thickness at 355 nm (AOT) is also given for each profile
with its uncertainty.

## 4    Vertical distribution from airborne observations

The purpose of this section is to highlight the spatial variability of the vertical structure of aerosols in the vicinity

of Henties Bay through an analysis of the airborne lidar observations acquired offshore during two flights, on 5

and 6 September 2017. Note that airborne observations during AEROCLO-sA were only made during period $P_3$

(Formenti et al., 2019).

## 4.1 Flight on 5 Septembre 2017

**Figure 8**a shows the time-space cross section of the LNG-derived apparent aerosol backscatter coefficient (ABC) profiles at 532 nm along the Falcon 20 flight track in the morning of 5 September 2017 following the methodology by Chazette and Totems (2017). LNG data highlight the presence of a widespread elevated BBA layer over the area of interest. The inversion of the LNG ABC data is performed using the same LRs as for the inversion of the ground-based lidar in Henties Bay (70 sr in the FT and 55 sr in the PBL, see Table 3). The average LNG-derived AEC profile shown in **Figure 8**b is obtained over the ocean between the two vertical dotted black lines in **Figure 8**a around 1000 UTC. **Figure 9** shows the comparison between the dropsonde profiles of temperature, wind and relative humidity (RH) located over the ocean in **Figure 8**a and their counterparts extracted from ERA5 at 1000 UTC in a 0.25° x 0.25° grid centred on the Henties Bay site. There is a very good agreement between the vertical wind profiles (intensity and direction), nonetheless the wind is a little stronger on the dropsonde vertical profile, especially around 2 km AMSL, above the marine PBL, where it is in excess of 20 m s$^{-1}$ (and less than 15 m s$^{-1}$ in ERA5). The dropsonde measurements provide evidence of a very sharp RH gradient at the top of the BBA layer (from 80% to nearly 1-2%, **Figure 9**b) at 6 km AMSL, this gradient being collocated with the large vertical gradient of AEC at 532 nm seen in the LNG data (**Figure 8**b). ). They also provide evidence of a minimum of RH above the PBL, around 2 km AMSL, roughly coinciding with the base of the BBA layer (~2.2 km AMSL, **Table 3**). The high RH values in the elevated BBA layer may be associated with the large amounts of water vapour released during combustion in wild fires (Clements et al., 2006; Deaconu et al., 2019; Parmar et al., 2008). The high RH may also be characteristic of continental air whereas low humidity air above may be associated with subsiding tropical or mid-latitude air that has been depleted of moisture via prior precipitation. The sharp RH gradient at the top of the BBA layer is not well represented in the ERA5 analysis. The depth of the marine PBL is also seen to be thicker in the observations than in the model (**Figure 9**b), possibly because the ERA5 profiles is partly over the Namibian coast. The airborne lidar data highlight the presence of stratocumulus over the ocean around 1 km AMSL (**Figure 8**b, the absence of lidar data below that height indicating that the laser beam is completely extinguished in the cloud), close to the maximum of RH observed with the dropsonde (**Figure 9**b).

When comparing the mean vertical distribution of aerosols from the LNG-derived AEC profile offshore and the ground-based lidar AEC profile in Henties Bay averaged between 1400 and 1500 UTC (**Figure 8**b, the two profiles being separated by ~100 km), we observe differences in terms of the altitude of the BBA layer top. Note that i) since the two lidars operate at different wavelengths, the AEC intensity is not directly comparable, but the vertical structure of AEC profiles is, and ii) there is a 4-hour difference between the aircraft profiles and the mean profile over Henties Bay. On the other hand, we see that the bottom of the BBA layer is located at roughly the same altitude (**Figure 8**b). Furthermore, ERA5 analyses also highlight the fact that the dynamical and thermodynamical structure of the lower troposphere over Henties Bay did not evolve significantly between 1000 and 1500 UTC (not shown), except for an increase of RH between 5 and 6 km AMSL (by 20%, coherent with the appearance of clouds as seen in Figure A2c) and of wind speed at 4.5 km AMSL (by 5 m s$^{-1}$). Rather, the difference may be explained by the regional scale circulation in the mid troposphere across the area. Over the ocean, ERA5 data indicates stronger northwest winds (~23 m s$^{-1}$) at the location of the airborne lidar AEC profile compared to the wind over Henties Bay (12 m s$^{-1}$) for the entire day on 5 September (not shown). The resulting horizontal wind shear between the Namibian coast and the ocean leads to differential advection within the BBA layer, and a different vertical structure of the aerosol layer between the coastline and over the ocean.

## 4.2    Flight on 6 Septembre 2017

During the flight on 6 September 2017 (**Figure 10**a), LNG observations were made further offshore than on the previous day. In **Figure 10**b, we compare the AEC profiles acquired with LNG to the west and the northwest of Henties Bay (marked '1' and '2', respectively in **Figure 10**a) at ~0830 and ~0900 UTC, with the average AEC profile obtained between 0700 and 0930 UTC from the ground-based lidar in Henties Bay. Differences in the structure of the BBA layer appear between the vertical profiles west of Henties Bay (profile '1' in **Figure 10**a) and the one further north (profile '2' in **Figure 10**a). The shape of the elevated BBA layer observed from the AEC profiles in '1' and in Henties Bay match the structure of the RH and wind speed profiles from the southernmost dropsonde (**Figure 11**b), with a top (base) altitude of 5 km (3 km) AMSL. The wind in the BBA layer is observed to be rather constant and equal to 17 m s$^{-1}$ on average as well as coming from the north. The maximum RH in the FT is ~55% and observed near the top of the BBA layer (**Figure 11**b), while small RH values (less than 10%) are seen above ~6 km AMSL. It is worth noting the presence of a slightly enhanced RH layer between 5.5 and 6 km AMSL, where enhanced lidar-derived AEC values are also observed in Henties Bay (**Figure 10**b). The elevated BBA layer is separated from the PBL by a rather dry layer with small AECs, characterized by a strong wind shear (**Figure 11**b).  The apparent height of the PBL observed in the AEC profile in Henties Bay agrees with the location of the gradient in RH.

The AEC profile '2' derived from LNG observations and obtained ~100 km north of profile '1' exhibits a different structure than that of Henties Bay. The top of the BBA layer is observed to be slightly higher (5.2 km AMSL) while the altitude of the base of the BBA layer is the same (~3 km AMSL). The wind speed in the BBA layer as seen from the northernmost dropsonde (**Figure 11**a) is weaker than when it is off Henties Bay (**Figure 11**b), while the RH is higher throughout the lower troposphere, especially below the elevated BBA layer. The LNG profile in '2' exhibits significant AEC values below 3 km AMSL corresponding to the base of the BBA layer observed further south, which may be partly related to the impact of RH on aerosol optical properties. A deep moist layer (including the PBL) is observed below the BBA layer.

In addition to the important variability in terms of vertical structure of the AEC profiles, it should be noted that the 550 nm AOT derived from the sun photometer in Henties Bay (0.70±0.05) is significantly higher than those determined from the airborne lidar data at 532 nm in '1' (0.37±0.06), but also significantly lower than that measured in '2' (1.13±0.10). This variability also is reflected in the vertical distribution of aerosols above 5 km AMSL, where non-negligible contributions to the AOT are observed in Henties Bay (with 0.15 < AOT < 0.35 at 355 nm, **Figure 2**b) and in '2' (with AOT ≳ 0.08 at 532 nm). Such a contribution was even more marked on the previous day in the LNG observations (see **Figure 10**b), with an AOT at 532 nm above 5 km AMSL in excess of ~0.05.

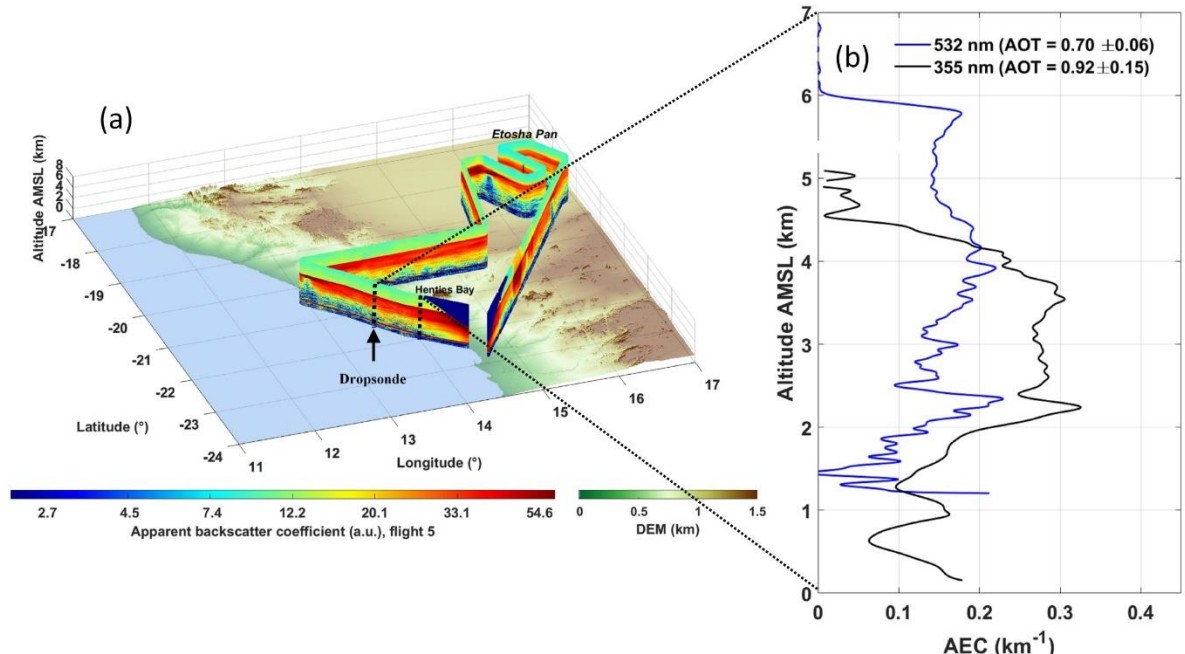


Figure 8: (a) Distance-height ("curtain-like") evolution of the LNG-derived apparent backscatter coefficient at 532 nm below
the SAFIRE Falcon 20 during the morning flight on 5 September 2017. The location of the dropsonde released over the ocean
is indicated as well as the location of the averaged LNG aerosol extinction coefficient (AEC) profile shown in (b) (between the
2 dotted vertical lines). (b) Vertical profiles of the AEC derived from the airborne lidar at 532 nm (~1000 UTC, blue solid line)
and from the ground-based lidar at 355 nm (~1400-1500 UTC, black solid line).

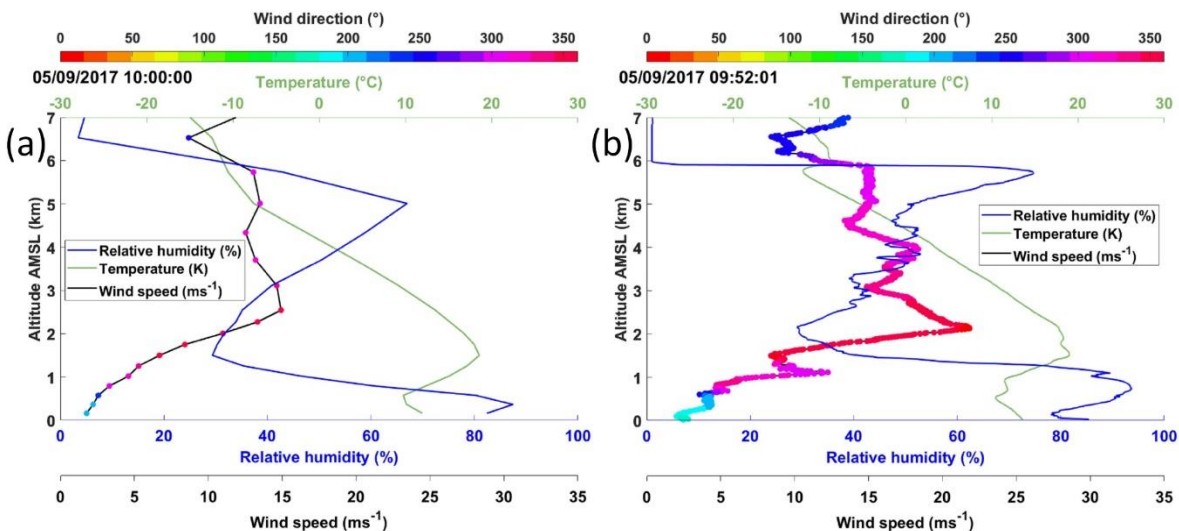


Figure 9: (a) Wind speed (black solid line), wind direction (coloured dots), RH (blue solid line) and temperature (green solid
line) profiles extracted from ERA5 at 1000 UTC above Henties Bay over a 0.25° by 0.25° grid. (b) Same as (a) but measured
by the dropsonde released over the ocean at 0952 UTC on 5 September 2017.

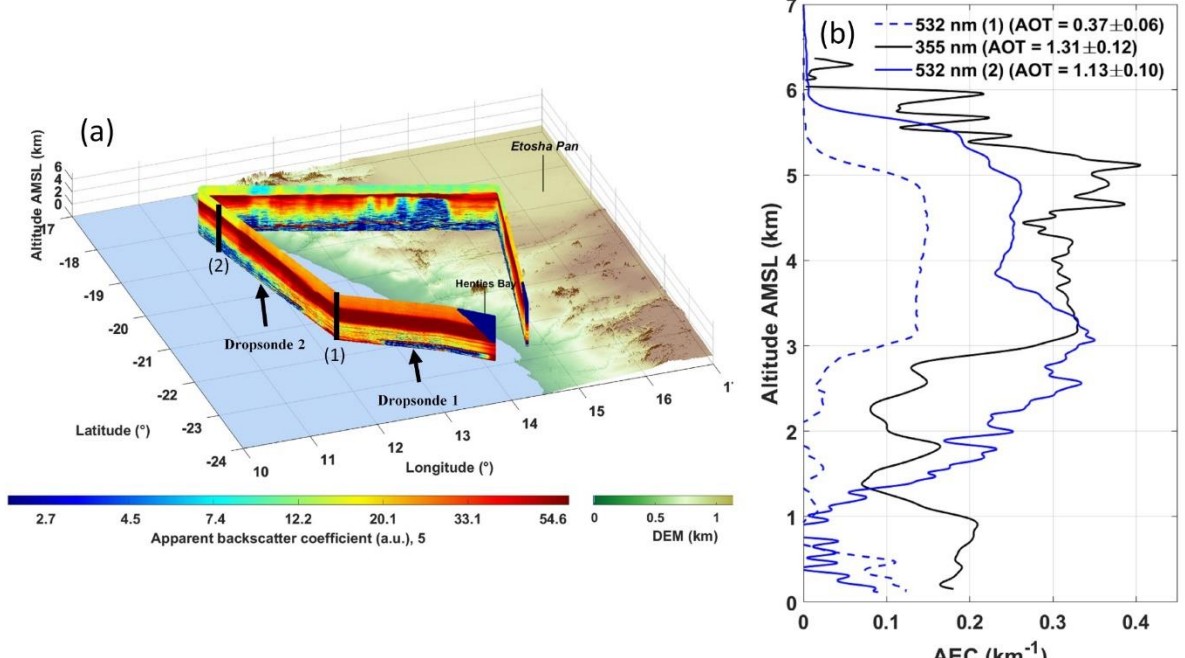


Figure 10: (a) Same as Figure 6a, but on 6 September 2017. The locations of the two launched dropsondes are also indicated
by arrows. The lidar AEC profile labelled '1' shown in (b) is obtained after inversion of the LNG observations averaged
between the two locations of the two dropsondes. The AEC profile labelled '2' is obtained after inversion of the lidar data
between the northern most dropsonde and the northern end of the Falcon leg. (b) Vertical profiles of the AEC derived from the
airborne lidar at 532 nm (~0830 and ~0900 UTC, for profile '2' (solid blue line) and '1' (dashed blue line), respectively) and
from the ground-based lidar at 355 nm (~0700-0930 UTC, black solid line).

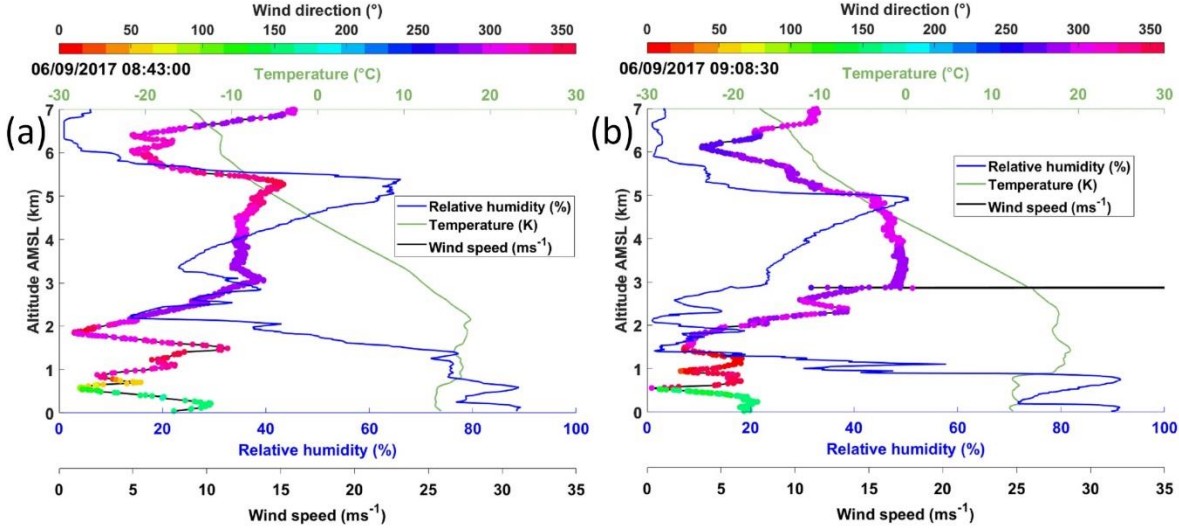

419

Figure 11: (a) & (b) Same as Figure 7b, but for the dropsondes released at 0843 UTC (to the northwest of Henties Bay,
Dropsonde 2 in Figure 10a) and at 0908 UTC (west of Henties Bay, Dropsonde 1 in Figure 10a).

## 5    Origin of elevated BBA layers over Henties Bay

### 5.1    RH as indicator of changing synoptic conditions

Figure 12 shows the time-height evolution of hourly RH profiles from ERA5 between 22 August and 9 September
2017 at Henties Bay. The 3 periods ($P_1$, $P_2$ and $P_3$) identified from the AOT (Figure 2) are seen to correspond to
distinct RH conditions in the mid troposphere, with rather dry conditions during $P_1$, then increased RH below 5 km
AMSL during $P_2$ and even more humid conditions below 6 km AMSL during $P_3$. For instance, the RH values
between 2.5 and 5 km AMSL increases from values below 10% to values in excess of 60% between $P_1$ and $P_2$,
which is most probably associated with the transport of BBA over Henties Bay. Likewise, the RH values between
5 and 6 km AMSL increases from 5% to ~70-80% between $P_2$ and $P_3$, which is an indication that the meteorology
has changed and that the origin of air masses may be different. Periods $P_2$ and $P_3$ are clearly separated by an episode
of very dry RH conditions on 2 September, the day also corresponding to a minimum of AOT over Henties Bay
(**Figure 2**). In general, the location of the elevated aerosol layer in the vertical corresponds to the highest RH as
previously observed from airborne measurements. In the following, we designed back trajectories analyses to
investigate the origin of the air masses in the FT.
**5.2    Air masses pathway change during the 3 periods**
A statistical study of the back trajectories of air masses originating from Henties Bay was designed to analyse the
circulations related to the 3 identified periods $P_1$, $P_2$ and $P_3$. Six-day back-trajectories are initialized at 1200 UTC
using the ensemble mode of the Lagrangian HYSPLIT model for which 27 isentropic trajectories are calculated
for each selected altitude point over Henties Bay. Altitudes are discretised every 250 m between the base height
(~1500 m AMSL) and the maximum top height (~6000 m AMSL) of the BBA layers. A composite of the back
trajectories is then made for the 3 different periods by calculating the probability of trajectories passing through
each grid point with a spatial resolution of 0.5°. This statistical approach makes it possible to consider the
dispersion of back trajectories that can be linked to complex atmospheric circulations. The altitude ranges selected
for releasing the back trajectories are derived from the structure of the elevated aerosol layer given in **Table 3** and
Figures 3-7. They are the same for the 3 periods in order to facilitate comparison: [1500 3000[ m AMSL, [3000
5000[ m AMSL and [5000 6000[ m AMSL. To visualize the results, we used the two-dimensional histograms
presented in Figures 13-15.
**5.2.1    Period $P_1$**
During $P_1$, the density of trajectories is highest to the north of Henties Bay, and particularly along the Angolan and
Namibian coastlines (**Figure 13**). The distribution of the trajectories suggests that the aerosols observed over
Henties Bay mainly originate from Angola and northern Namibia (close to the back trajectories starting point) and
are transported towards the observational super site. Considering the altitude of the back trajectories, plausible
injection heights over Angola are highly variable and may reach ~5 km AMSL to explain the vertical structures of
lidar profiles. There are also many trajectories coming from over the southern Atlantic Ocean. For the altitude
range [3000 5000[ m, some trajectories arriving on 25 August in Henties Bay are seen to originate from southern
Brazil 6 days earlier, a region where fires are detected by MODIS between 16 and 21 August. It should be noted
that BBA would have needed to be injected to heights between 5 and 7 km AMSL in order to be transported to
Henties Bay on 25 August. Nevertheless, no lidar measurements are available during this day to confirm this
possible alley of cross-Atlantic transport. For the altitude ranges [5000 6000[ m no significant aerosol layer is
observed by the ground-based lidar (**Figure 3**).

### 5.2.2 Period P₂

During P$_2$ (**Figure 14**), the density of trajectories is also high along the Namibia coastline north of Henties Bay between 1500 and 5000 m AMSL and over the ocean. The distribution of trajectories suggests that the BBA observed in Henties Bay mainly are advected within the altitude range [3000 5000[ m from central Angola and have travelled a few hundred kilometres over the ocean before being transported back towards the southern African coastline. This constitutes the main contribution of the lidar-derived AEC profiles, provided that the injection heights over Angola can reach 5 km AMSL, as suggested by the CALIOP and CATS observations (see Figure A3). As for P$_1$, we observed no significant aerosol contribution above 5 km AMSL (**Figure 4**). The contribution from South America are due to air masses arriving over Henties Bay on 30 and 31 August between 3 and 5 km AMSL. These air masses have the possibility to import biomass burning aerosols emitted 6 days before from northern Argentina and injected at altitudes close to 4 km AMSL according to back trajectories. Such injection heights are often observed via CALIOP over South America. The lidar observations over Henties Bay do not show any significant AEC features above 5 km AMSL, in spite of the possibility of cross-Atlantic transport highlighted by the back trajectories. This could be related to a lack of fires in the region overpassed by the trajectories, or injection heights in the biomass burning regions that are below the altitude of the transport associated with the trajectories. It may also be the case that BBA are subject to wet deposition along the trajectories as air masses experience precipitation associated with the weather systems over the Atlantic Ocean.

### 5.2.3 Period P₃

During P$_3$ for the 3 altitude ranges, the occurrence of trajectories (**Figure 15**) is highest along the northern Namibian coast, over the land. This suggests a more direct transport from the anthropogenic- and/or wild-fire areas in Angola than during P$_2$ and P$_1$, which may explain the highest AOTs for the third period. The occurrence of trajectories over the ocean just west of the southern African coast suggests that a significant part of the aerosols arriving in Henties Bay have travelled over the Atlantic ocean before being transported back towards the continent. This constitutes the main contribution of the lidar-derived AEC profiles below 5 km AMSL, provided that the injection heights over Angola can reach that height over the continent. Above 5 km AMSL, significant AEC features are observed with the lidar (Figure 6 and 7) that reliably contribute to the AOT ((~10-15%, Figure 2b). According to Figure 15c, such features could be related to transport from Angola, provided that BBA are injected sufficiently high over the biomass burning areas. Figure 15c also shows that a significant number of trajectories reaching Henties Bay come from South America. For instance, more trajectories originating from the South America burning zones are also seen over the southern Atlantic Ocean for the altitude range [5000 6000[m than during the two other periods. Several transport pathways from South America to southern Africa are observed for this altitude range: (i) two southern routes where trajectories go as far south as 48°S for the first one and 40°S for the second one before moving equatorward towards Namibia, (ii) a northern routes where trajectories first follow the eastern coast of Brazil before heading due east towards Namibia, and (iii) a more direct eastward route across the Atlantic before turning counter clockwise towards Henties Bay. Back trajectories suggest that air mass transport from South America along the last 3 more northern routes took 5 to 6 days to reach Henties Bay, whereas the transport along the more southern route only took 3-4 days.

### 5.3 Possible contribution to the AOT from South America during P₃

We now look specifically at the P₃ period during which a large number of trajectories coming from South America is seen compared with the two other periods. Some of the aerosol layers observed during P₃ between 5 and 6 km AMSL by the ground-based lidar, and in particular those associated with the highest AOTs on 6 and 7 September 2017 (Figure 2b), may be associated with biomass burning over Angola, but also with fires occurring on 1-4 September 2017 over southern Brazil, northern Argentina and Uruguay.

The back trajectories shown in Figures 13-15 are calculated assuming isentropic transport. However, this hypothesis is not necessarily verified during the studied period. Indeed, when trajectories cross the Atlantic Ocean, they encounter more a baroclinic fluid than a barotropic fluid due to the presence of strong low pressure centres such as the cut-off low. The potential temperature is therefore no longer necessarily a tracer of the air mass and isentropic trajectories can quickly diverge towards higher altitudes. This is shown in **Figure 16** on 6 September (the same is true on 7 September). Nevertheless, some trajectories pass under 5 km AMSL over northern Argentina. The same trajectory simulation conducted with an isobaric hypothesis on 6 and 7 September shows that all the back-trajectories come from Argentina for altitudes that remain in the range of biomass burning injection heights (~5 km AMSL). However, isobaric trajectories are not necessarily more representative than isentropic trajectories (Stohl, 1998).

MODIS-derived AOTs (**Figure 17**) highlight the existence of an aerosol plume over the ocean along the northern fringe of a large cloud band. The location of fires over South America are also indicated in Figure 17a on 3 September 2017. The BBAs seem to be advected across the Atlantic Ocean along two main routes also identified in the previous back trajectory analyses (Section 5.2.3). The northernmost one follows the coast of Brazil before heading straight towards Namibian coasts. The poleward one follows the strong winds at 500 hPa along the western flank of a high pressure centred over the eastern coast of Brazil (**Figure 17**a). A mid-tropospheric westerly jet then transports the aerosol plumes over the Atlantic Ocean where they are then advected northward around the eastern edge of the high-pressure system located over the Atlantic Ocean. The ubiquitous cloud cover along the southern and eastern fringes of the high-pressure system does not allow the retrieval of AOTs with MODIS, except offshore of the Rio de la Plata estuary and at the edge of cloud fields caught in the west-east circulation. The northward progression of the air masses transporting the BBA along the coast is further accelerated by the presence of a poleward moving cut-off low (centred at 40°S, 15°W) separating from the westerlies further south (**Figure 17**a). Over the following days, the cut-off low is seen to merge back with the westerlies while progressing eastward, and the high-pressure system at 500 hPa is observed to also move over the Atlantic Ocean and merge with the St Helena high on 5 September (**Figure 17**b). The mid-tropospheric westerly jet may transport the aerosols issued from biomass burning over South America along the southern fringe of the St Helena high, which is centred at ~25°S and ~20°W. The jet is seen to extend quite far east over the Atlantic Ocean and to almost reach the southern tip of southern Africa (**Figure 17**c). Some aerosols travelling along the southern route may be redirected towards Namibia by the strong northerly flow along the eastern flank of the St Helena high.

Furthermore, the temporal variability of BBA transport patterns from South America to southern Africa may be related to the variability of the Southern Annular Mode (SAM, i.e. the north-south movement of the westerly wind belt around Antarctica). Indeed, Trenberth (2002) show that the SAM is the main driver of extratropical circulation in the Southern Hemisphere on weekly to decennial time scales, which is also the main driver of climate variability, affecting anthropogenic- and/or wild-fire activities over South America (e.g. Holz et al., 2017). For instance, positive phases of the SAM (i.e. when a band of westerly winds contracts toward Antarctica) are associated

primarily with warm conditions in the forested areas of South America, thereby favouring biomass burning events.
On the other hand, negative phases lead to an expansion of the wind belt towards the lower latitudes, leading to
the possibility for BBA transported in the westerlies to reach southern Africa in the austral winter. Given the
possible short time scale of variability of the SAM, it is likely that the transport patterns to Henties Bay identified
during period $P_3$ are related to a negative SAM phase, while during $P_1$ they are related to a positive phase. On
longer time scales, climate modelling studies indicate a robust positive trend in the SAM for the end of this century
(Lim et al., 2016), so that climate conditions conducive to an impact of the widespread South American fire activity
in southern Africa will likely continue throughout the 21st century. . However, further studies are needed to support
this conclusion, which will have to be based on longer observation periods involving lidar technology.

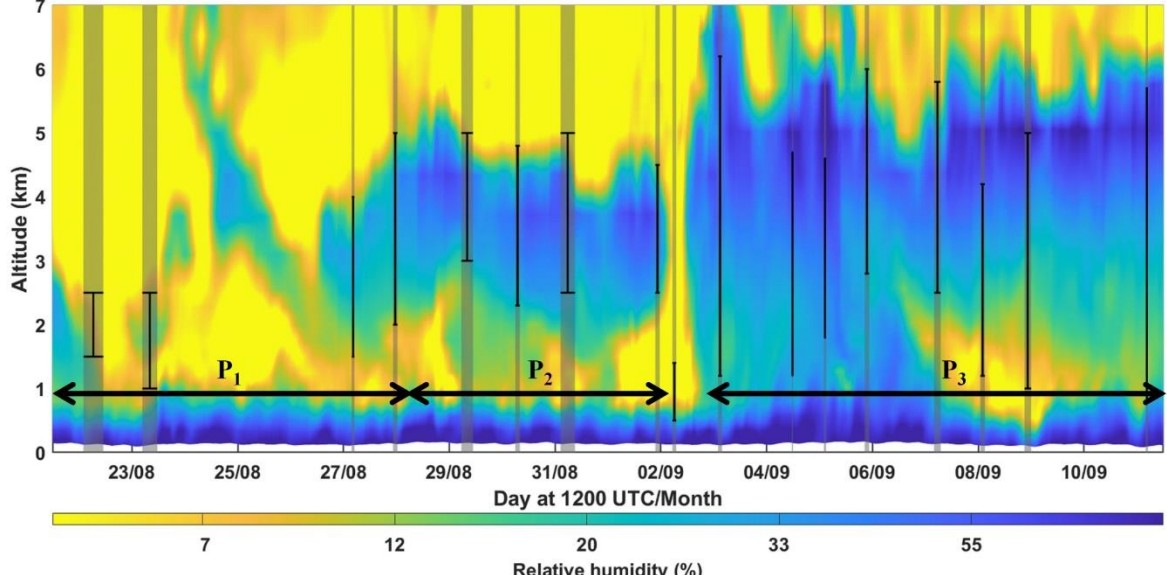


Figure 12: Time-height evolution of the relative humidity vertical profiles derived from ERA5 above Henties Bay. The grey
vertical lines indicate the time of the ground-based lidar profiles shown in Figure 3-7. The thickness of the grey lines depends
on the averaging period (the thicker the line, the longer the average). The 3 periods highlighted by the AOT values ($P_1$, $P_2$ and
$P_3$) are also indicated. The vertical black lines show the lidar-derived altitude location of the aerosol layer.

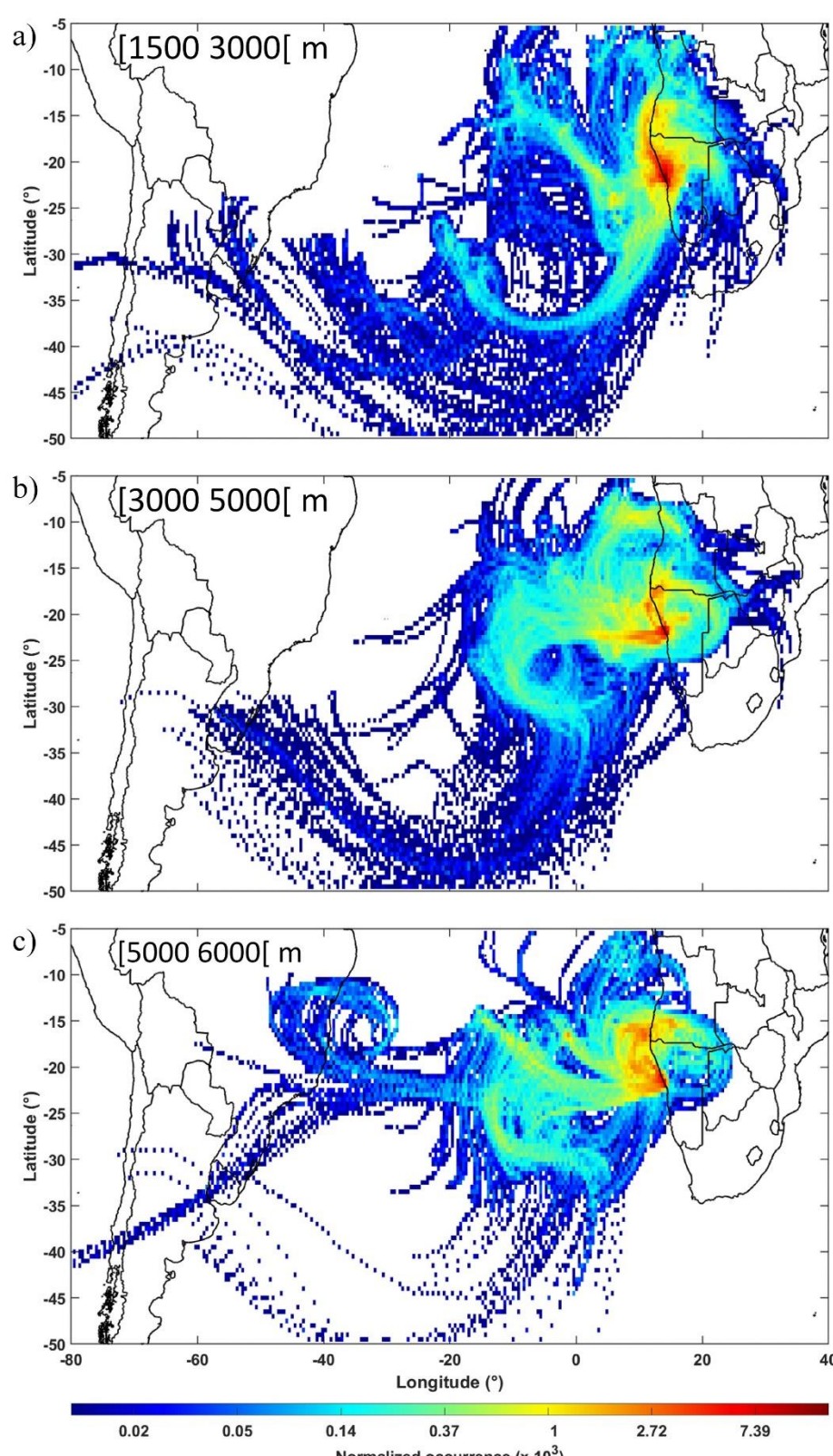


Figure 13: Normalized occurrence of the back trajectories starting over Henties Bay at 1200 UTC during periods $P_1$, from the altitude range [1500 3000[ (a) [3000 5000[ (b) and [5000 6000[ (c), m. The calculations have been made using 6-day isentropic back trajectories with the HYSPLIT model (courtesy of NOAA Air Resources Laboratory; http://www.arl.noaa.gov) in ensemble mode. The normaliation is performed with respect to the total number of pixels for a horizontal resolution of 0.5°.

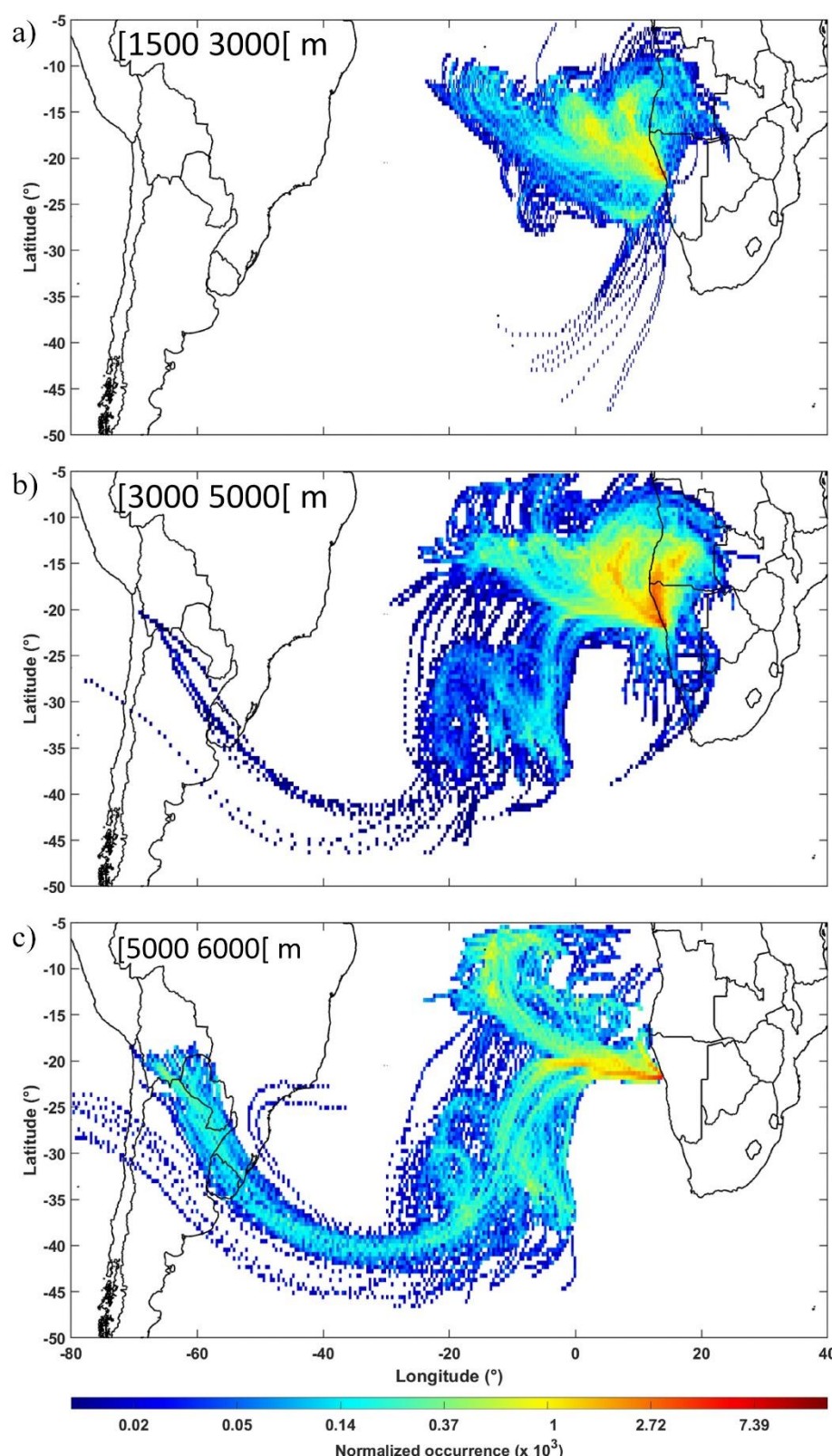

559

Figure 14: Normalized occurrence of the back trajectories starting over Henties Bay at 1200 UTC during periods P2, from the altitude range [1500 3000[ (a) [3000 5000[ (b) and [5000 6000[ (c), m. The calculations have been made using 6-day isentropic back trajectories with the HYSPLIT model (courtesy of NOAA Air Resources Laboratory; http://www.arl.noaa.gov) in ensemble mode. The normalization is performed with respect to the total number of pixels for a horizontal resolution of 0.5°.

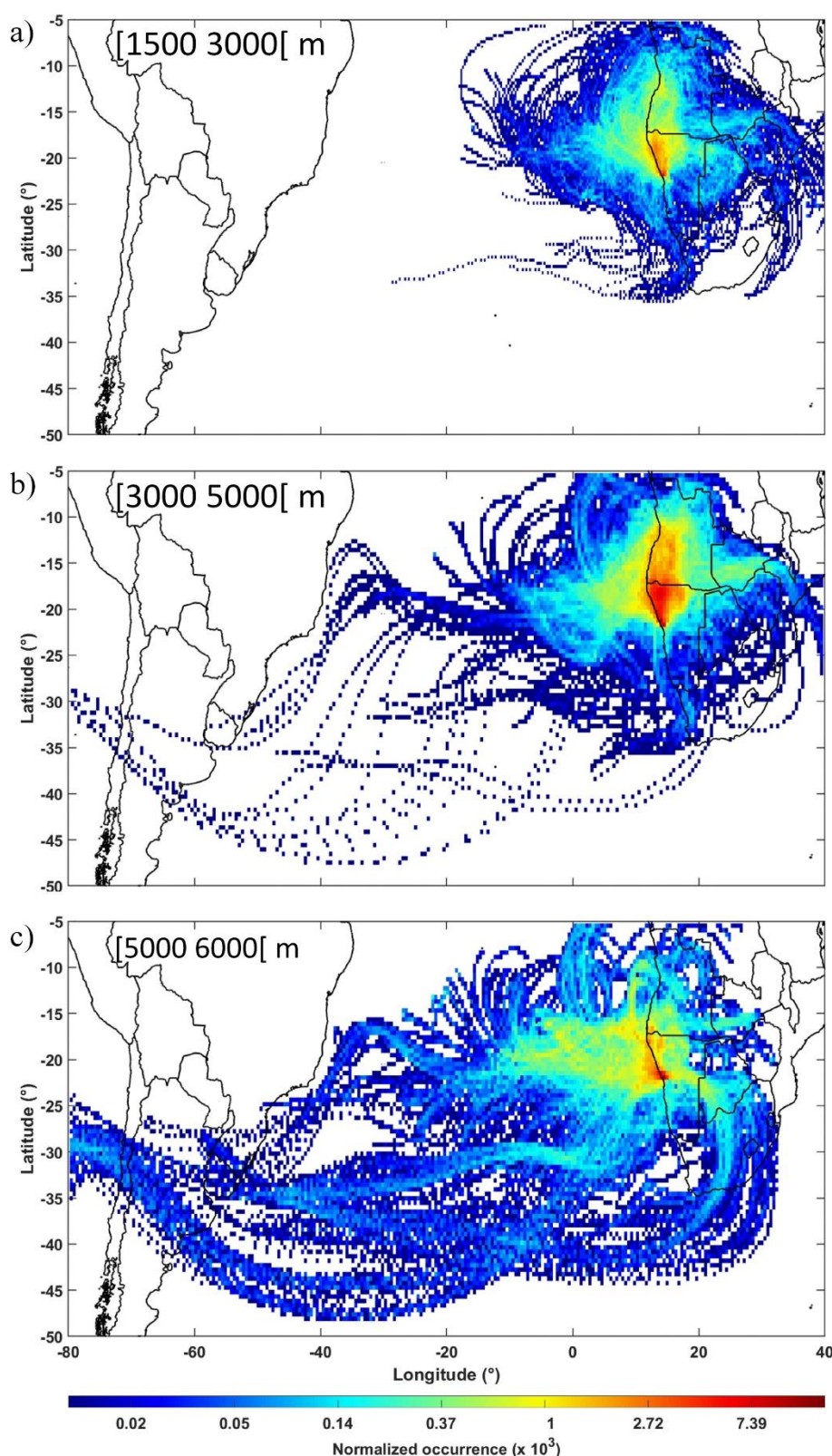

564

Figure 15: Normalized occurrence of the back trajectories starting over Henties Bay at 1200 UTC during periods $P_3$, from the altitude range [1500 3000[ (a) [3000 5000[ (b) and [5000 6000[ (c), m. The calculations have been made using 6-day isentropic back trajectories with the HYSPLIT model (courtesy of NOAA Air Resources Laboratory; http://www.arl.noaa.gov) in ensemble mode. The normalization is performed with respect to the total number of pixels for a horizontal resolution of 0.5°.

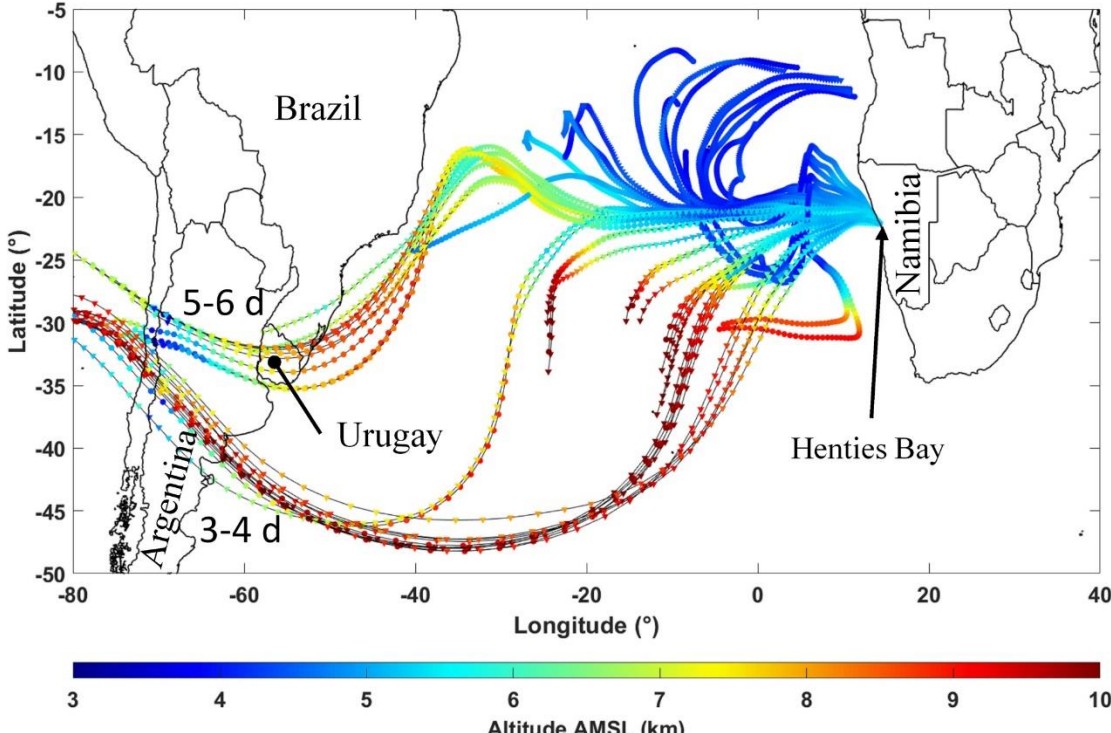


Figure 16: 6-days isentropic back trajectories starting over Henties Bay on 6 September at 1200 UTC. They are computed by
the HYSPLIT model (courtesy of NOAA Air Resources Laboratory; http://www.arl.noaa.gov) in ensemble mode. The time to
arrival above the South America is indicated. The altitude of back trajectories along the route is given by the colour bar.

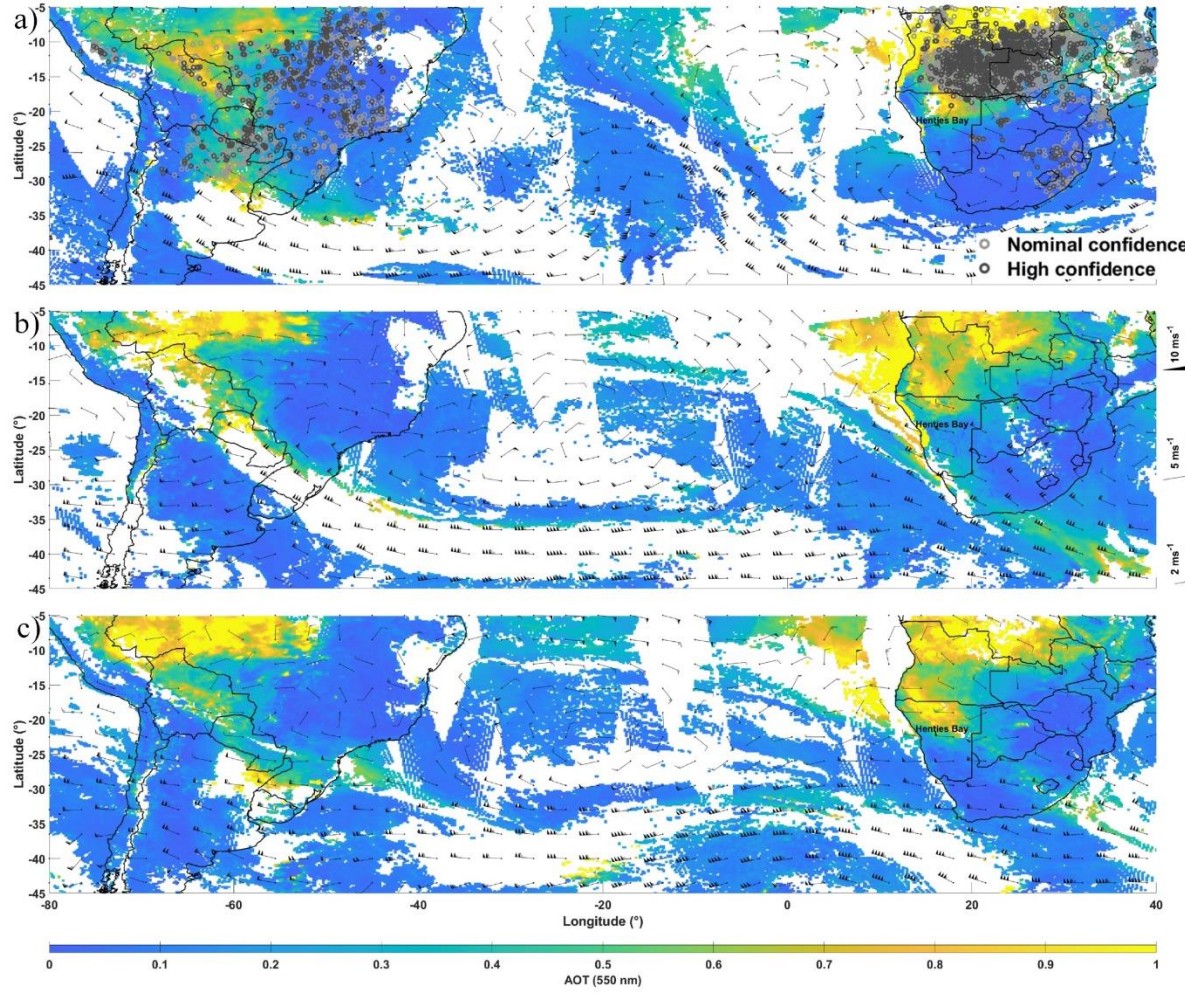


Figure 17: MODIS-derived AOT at 550 nm on (a) on 3 September 2017 with wild fire hotspots over both South Africa and
South America, (b) on 5 September 2017 and c) 6 September 2017. The ERA5 wind field at 500 hPa on each day have been
added in black.
**6    Conclusion**
During the intensive field campaign of the AEROCLO-sA project (22 August - 12 September 2017), the very
persistent cloud cover topping the marine boundary did not allow continuous ground-based monitoring of the
aerosol layers above the stratocumulus deck, in the mid-troposphere. Nevertheless, the available lidar observations
performed over the coastal site of Henties Bay allowed to highlight three contrasted periods of biomass burning
aerosol transport ($P_1$, $P_2$ and $P_3$). The inversion of the ground-based lidar profiles was carried out using the
constraints provided by the aerosol typing of the CALIOP and CATS space-borne instruments, but also the
photometric measurements from AERONET network. The latter showed an overall good agreement with the
MODIS AOT observations and the AOT outputs of the CAMS model. Differences were noted in the presence of
high aerosol contents (AOT at 355 nm > 0.8) between the lidar- and sun photometrer-derived AOTs, but those
were likely due to the presence of clouds that were not detected by the passive sensors.
Combining observations and back trajectory analyses, we highlight the existence of 3 periods with very different
transport modes towards Henties Bay during the field campaign. The lowest AOTs (<0.2 at 550 nm) of the first
period ($P_1$) are associated with air masses from Angola travelling along the Namibian and Angolan coasts.
Intermediate AOTs (~0.4 at 550 nm) of the second period (P$_2$) are associated with polluted dusts (i.e. dust mixed
with biomass burning aerosols from Angola), as well as dust from the Etosha Pan, which are recirculated above
the ocean. During the third period (P$_3$), the largest AOTs (~0.7 at 550 nm) are observed, mainly due to a more
direct transport from the Angola burning areas with an aerosol plume vertical extending between 1.5 and ~6 km
AMSL. The atmospheric composition in the free troposphere for this period is the most variable in the time. We
show a possible contribution of forest fire aerosols from South America (South of Brazil, Argentina and Uruguay)
with plumes transported to Henties Bay around 5000-6000 m AMSL and mainly observed on 6 and 7 September
with a contribution to the total AOT of ~10-15%. The aerosol plume from South America could be advected across
the Atlantic Ocean along a route following the strong westerlies of the southern fringes of the St Helena high
before heading north toward Namibia in connection with an equatorward moving cut-off low.
To the authors' knowledge, this is the first time that the evolution of the optical properties of aerosols in the FT
over coastal Namibia is characterized, in relation to different transport regimes. The main contribution of the BBA
from Angola and the arguably smaller contribution of the South American anthropogenic- and/or wild-fires to the
atmospheric aerosol composition over the Namibian coast were shown. The synergy between active and passive
remote sensing observations performed from ground-based and space-borne platforms together with back
trajectory analyses, was essential to provide these conclusions.

**Acknowledgments.** This work was supported by the French National Research Agency under grant agreement n°
ANR-15-CE01-0014-01, the French national program LEFE/INSU, the Programme national de Télédetection
Spatiale (PNTS, http://www.insu.cnrs.fr/pnts), grant n° PNTS-2016-14, the French National Agency for Space
Studies (CNES), and the South African National Research Foundation (NRF) under grant UID 105958. The
authors would also like to thank the AERIS data center for their support during the campaign and managing the
AEROCLO-sA database. The research leading to these results has received funding from the European Union's
7th Framework Programme (FP7/2014-2018) under EUFAR2 contract n°312609". Airborne data was obtained
using the aircraft managed by SAFIRE, the French facility for airborne research, an infrastructure of the French
National Center for Scientific Research (CNRS), Météo-France and the French National Center for Space Studies
(CNES). The authors would like to thank F. Blouzon and A. Abchiche (DT-INSU) as well as P. Genau and M. van
Haecke (LATMOS) for their support in operating and processing the LNG data.  The invaluable diplomatic
assistance of the French Embassy in Namibia, the administrative support of the Service Partnership and
Valorisation of the Regional Delegation of the Paris-Villejuif Region of the CNRS, and the cooperation of the
Namibian National Commission on Research, Science and Technology (NCRST) are sincerely acknowledged. The
long-term hosting and support of the SANUMARC, a research center of the University of Namibia in Henties Bay
have been essential through the years and are warmly appreciated. The authors acknowledge the MODIS science,
processing and data support teams for producing and providing MODIS data (at
https://modis.gsfc.nasa.gov/data/dataprod/) and the NASA Langley Research Center Atmospheric Sciences Data
Center for the data processing and distribution of CALIPSO products (level 4.10, at
https://eosweb.larc.nasa.gov/HORDERBIN/HTML_Start.cgi). The authors would like to thank the AERONET
network for sun photometer products (at https://aeronet.gsfc.nasa.gov/). Finally, the authors are grateful to Michael
Diamond and an anonymous reviewer for their comments that helped improve the overall quality of the paper.

**Data availability.** The aircraft and ground-based data used here can be accessed using the AEROCLO-sA database
at http://baobab.sedoo.fr/AEROCLO-sA/. An embargo period of 2 years after the upload applies. After that,
external users can access the data in the same way as AEROCLO-sA participants before that time. Before the end
of the embargo period, external users can request the release of individual datasets. It is planned for AEROCLO-
sA data to get DOIs, but this has not been carried out for all datasets yet. The back trajectories data can be obtained
upon request to the first author of the paper.
**Author contributions.** PC inverted the ground-based and airborne lidar data, analysed the data and wrote the
paper, with comments from all the co-authors; CF analysed the data and wrote the paper; JT aligned and validated
the ground-based lidar, MG participated to the study of atmospheric dynamic and to the paper editing, GS
participated to the back-trajectories computation, AB gathered the CATS lidar data and the wind fields, PF
coordinated the AEROCLO-sA project, XL participated in the pre- and post-field calibration and operation of the
lidar, KD and JFD maintained and operated the lidar during the field campaign.
**Competing interests.** The authors declare that they have no conflict of interest.
**Special issue statement.** This article is part of the special issue "New observations and related modeling studies
of the aerosol–cloud–climate system in the Southeast Atlantic and southern Africa regions" (ACP/AMT inter-
journal SI)". It is not associated with a conference.

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

**Appendix A: Ground-based lidar analysis – link with spaceborne lidar observations**
**A.1    Description of the ground-based lidar**
The ground-based lidar system used at the Henties Bay site is the ALS450® lidar manufactured by Leosphere and
initially developed by the Commissariat à l'Energie Atomique (CEA) and the Centre National de la Recherche
Scientifique (CNRS) (Royer et al., 2011a). The lidar emission is based on an Ultra® Nd:YAG laser manufactured
by Quantel, delivering 6 ns width pulses at the repetition rate of 20 Hz with a mean pulse energy of 16 mJ at a
wavelength of 355 nm. This system is particularly well-adapted to measure tropospheric aerosol profiles in the
lower and middle troposphere. Its high vertical resolution of ~15 m after filtering and temporal resolution (~1
minute) gives the advantage of being able to follow the fast vertical evolutions of the atmospheric scattering layers
and to accurately locate the aerosol layers within the troposphere. The lidar is composed of two receiver channels
dedicated to the measurement of the co-polar and cross-polar signals. The detection is carried out by
photomultiplier tubes and narrowband filters with a bandwidth of 0.5 nm. Its main characteristics are summarized
in Table A1 where we have added the features of the LNG lidar for comparison.

Table A1: Main characteristics of both the ALS and LNG lidars.

| | Ground-based lidar ALS | Airborne lidar LNG |
|---|---|---|
| Laser | Nd:YAG, flash-pumped, Q-switched Q-smart QUANTEL | Flashlamp-pumped Nd:YAG Q-switched oscillator (Quantel YG980) |
| Pulse duration | 6 ns | 6 ns @ 335 nm 7 ns @ 532 nm 8 ns @ 1064 nm |
| Reception channels | // 354.7 nm ⊥ 354.7 nm | // 355, 532 and 1064 nm ⊥ 355 nm |
| Emitted energy | 16 mJ | 50 mJ @ 335 nm 10 mJ @ 532 nm 50 mJ @ 1064 nm |
| Frequency | 20 Hz | 20 Hz |
| Reception diameter | 15 cm | 30 cm (Cassegrain telescope) |
| Field-of-view | ~2.3 mrad | 0.5 mrd @ 335 nm 6 mrd @ 532 nm 8 mrd @ 1064 nm |
| Filter bandwidth/transmission | 0.5 nm / 70% @ 335 nm // and ⊥ | 5 nm/ 25% @ 335 nm // and ⊥ 0.2 nm / 25% @ 532 nm 1 nm / 30% @ 1064 nm |
| Detector | Photomultiplier (PM) tubes | PM Hamamatsu H6780-04 @ 355 nm |

| | | PM Hamamatsu H6780-02 @ 532 nm APD Perkin-Elmer C30659-1060 @ 1064 nm |
|---|---|---|
| Post-processing vertical resolution | 15-30 m | 6 m |
| Post-processing Temporal resolution | Variable, see Table 1 | 1 minute |


### A.2 Overlap correction and rightness of lidar profiles

In order to derive aerosol extinction coefficient profiles (AEC), the lidar apparent backscatter coefficient (ABC)
in the aerosol-free portions of the vertical profiles must be assessed and must follow the slope of the molecular
backscattering. The ABC, also called the total attenuated backscatter coefficient (Royer et al., 2011a), correspond
to the raw lidar signal corrected for both the contribution of the sky background and the solid angle, as in the
Equation (3) of Royer et al. (2010).
Furthermore, close to the lidar emission source the overlap factor generated by the overlap defects of the laser
emission and telescope reception fields also needs to be assessed. The overlap factor is derived from measurements
acquired in the horizontal line of sight, with the hypothesis of a homogeneous atmosphere along the line of sight
between the emission and a distance of 1.5 km. The overlap factor and the associated standard deviation are shown
in Figure A1. It can be considered that the correction of the overlap factor induces a relative error lower than 15%
for an overlap factor between 0.8 and 1 (Chazette, 2003), corresponding to a distance of 150 m from the emitter.
The molecular contribution is obtained from the Era5 pressure and temperature data at the horizontal resolution of
0.25° using the Nicolet model (Nicolet, 1984). The error on the aerosol extinction coefficient due to uncertainty
on the molecular density remains below 2-3% (Chazette et al., 2012b). The main sources of uncertainty are the
shoot noise and the atmospheric variability during the measurement. Both are taken into account for each retrieved
profile.
A representative time-average lidar profiles of the ABC over the duration of the measurement field campaign is
shown in Figure A2. The dates were chosen to be representative of the dataset of lidar vertical profiles encountered
during the AEROCLO-sA campaign. The curves in black are the ABC profiles and those in red correspond to the
molecular backscatter coefficient computed using ERA5 data. We note that in the top of the profiles there is a very
good agreement that ensures that the lidar is well aligned. The area comprised between the black and red curves
corresponds to the contribution of atmospheric aerosols and, in the upper part of the profiles, to that of optically
thin clouds (Figure A2c and d). The aerosol content increases rapidly between 22 and 28 August, showing a
significant evolution of aerosol contributions in the free troposphere (FT), between 1 and 5 km above the mean
sea level (AMSL). It is notable that the vertical profiles of the ABC vary little during the averaging period, the
average profiles are therefore quite representative of the state of the atmosphere for all the considered periods.

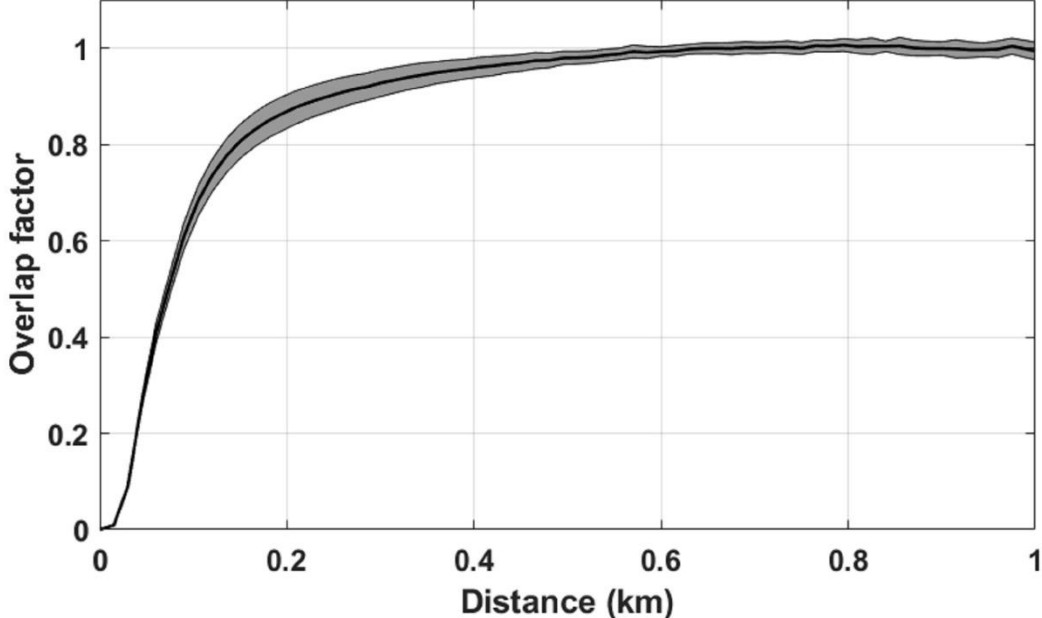


Figure A1: Overlap factor of the ALS (continuous black line) and its standard deviation (grey area).

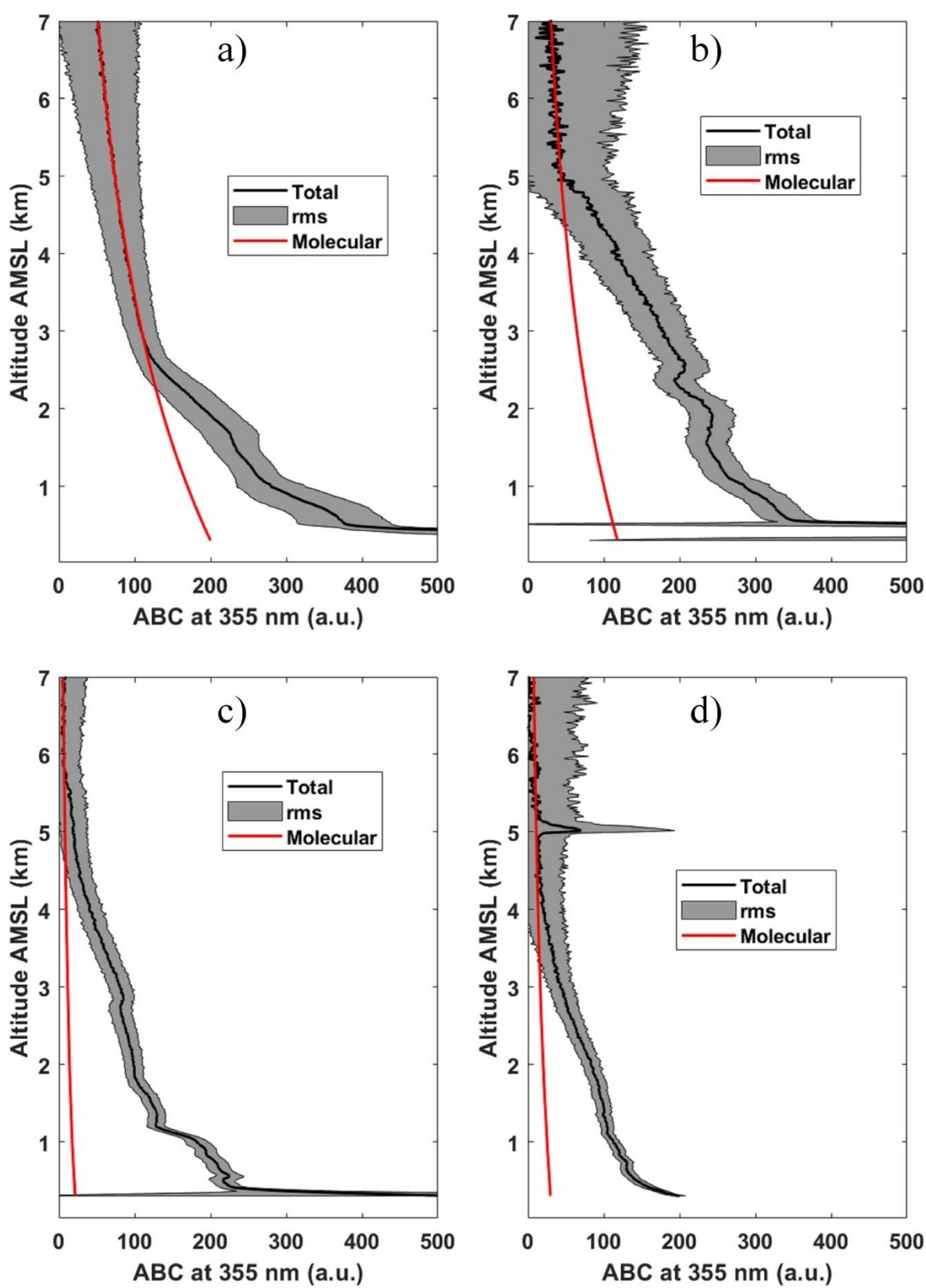


Figure A2: Apparent backscatter coefficient (black solid lines) profiles obtained from the ASL lidar in Henties Bay on: a) 22
August 2017 between 1400 and 2300 UTC, b) 28 August 2017 between 1030 and 1230 UTC, c) 7 September 2017 between
1600 and 1900 UTC, and d) 8 September 2017 between 1300 and 1500 UTC. The red lines correspond to the molecular
backscatter coefficient computed using ERA5 data. The grey area is the standard deviation linked with the statistical error (the
shoot noise and the atmospheric variability).

### A.3 Ground-based lidar data processing using external constraints

The inversion procedure to retrieve the aerosol optical properties from ALS is well documented in previous articles
where uncertainty sources are exhaustively quantified (e.g. Raut and Chazette, 2009; Royer et al., 2011b; Chazette
et al., 2012a). In the present case, where a simple elastic backscattering lidar is used, we use additional constraints
to the lidar equation using sun photometer-derived aerosol optical thickness (AOT) when available, but also the
aerosol typing determined from the CALIOP and CATS measurements for cases where the orbit allowed the
sampling of aerosols present in the FT. Figure A3 gives the example of the case of the geographical coincidence
between the night CALIOP (CATS) orbit on 28 (30) August 2017 and the lidar measurements above the Henties
Bay site. All available CALIOP and CATS orbits passing over Namibia were analysed and the results in terms of
aerosol typing are given in **Table 1** and Table 2. The correspondences in terms of LR are given in **Table 2** for both
instruments.
In the area of interest, aerosol properties are different in the planetary boundary layer (PBL), where the composition
is dominated by marine and coastal dust emissions, and in the FT where the composition is dominated by long-
range transport of BBA and dust emitted over the continental plateau. Therefore, we have used different values of
LR in the PBL and in the FT to perform the lidar inversion when lidar measurements were acquired concomitantly
with sun photometer AOT measurements. The LR in the FT is derived from the aerosol typing performed by the
space-borne lidars (see Table 2). When there is no CALIOP or CATS overpasses we take the value of LR of the
nearest day also considering the shape of the AEC profile and the origin of air masses using back trajectories.
Values of 65-70±25 sr and 55±25 sr at 532 nm are used for the two main aerosol types sampled, namely smoke
and polluted dust, respectively. The ground-based lidar in Henties Bay operates at 355 nm, the LR value is then
different. Müller et al. (2007) showed that LR values at 355 and 532 nm differ by about of 20% for forest fire
smoke and less than 10% for dust aerosols (see the Table 1 of their paper), widely included in the expected
uncertainty in LRs for spaceborne lidar. In the PBL, the LR values are selected from the discrete set of lidar ratios
shown in Table 2 via a minimization of the difference of AOT between the ground-based lidar and the sun
photometer: the LR in the PBL is adjusted so that the AOT calculated from the lidar AEC profile matches best the
AOT from the sun photometer at 355 nm. The LR values obtained during the field campaign are associated with
clean marine air aerosols (i.e. 20-23 sr) and polluted dust (i.e. 55 sr). This was done for all days listed in Table 3,
with the exception of 8 and 9 September 2017. On those days, the sun photometer AOT could not be used to
constrain the inversion of the lidar measurements. This is likely due to the presence of unscreened clouds in the
sun photometer inversion (as logged by the ground-based lidar on 8 September, Figure A2d). For those two days,
we have used a LR of 20 sr in the PBL to be able to invert the lidar data. Note that the use of a value of 55 sr in
the PBL on those days (i.e. the value retrieved for the previous days) leads to an unrealistically high lidar-derived
AOT. As a consequence, we observed an underestimation of the lidar-derived AOT when compared to the sun
photometer level 2 product.
Besides the determination of the AEC, we also evaluated the linear particle depolarization ratio (PDR) values using
an approach described in Chazette et al. (2012b). A detailed study of uncertainties for different aerosol types can
be found in Dieudonné et al. (2017). Statistical errors of 2% on the PDR can be expected due to statistical noise
but the biais linked to the uncertainty on the LR increases these errors.

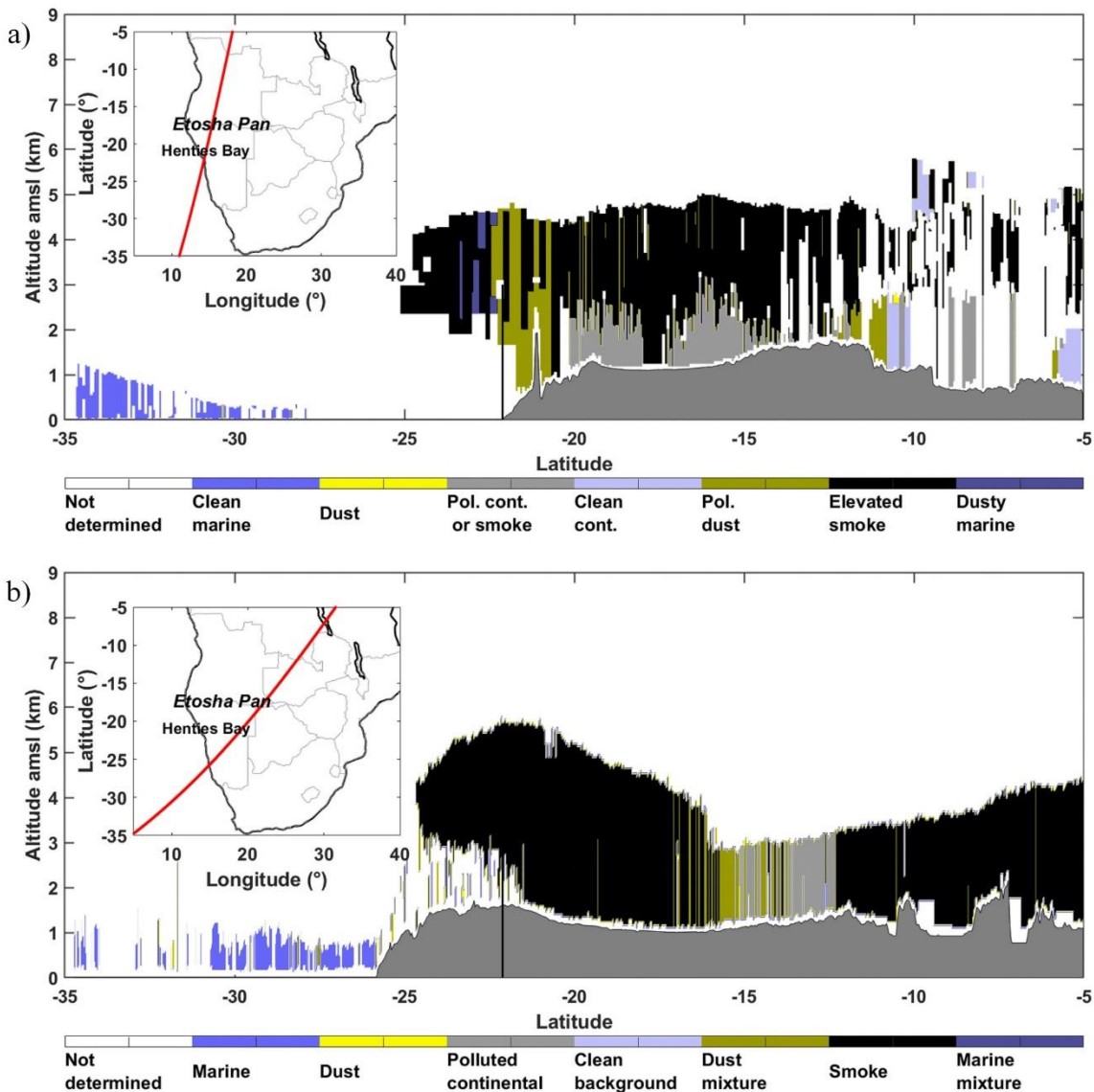


Figure A3: a) CALIOP-derived aerosol typing for the night time orbit (10.2017-08-28T00-08-17ZN) on 28 August 2017.  b) CATS-derived aerosol typing for the night time orbit (2017-08-30T00-32-37T01-18-13UT) on 30 August 2017. The latitudinal location of the Henties Bay site is given by the vertical black line. Inserted panels in a) and b) show the position of the space-borne lidar tracks over southern Africa and with respect to Henties Bay.

Figure A4 presents two vertical profiles on 22 August and 7 September 2017 which have been considered to illustrate the error due to the choice of the LR. The AEC is affected by less than 0.02 km⁻¹ except at the upper part of the profile on 7 September when the attenuation strongly decreases the signal to noise ratio. The AOTs at 355 nm are 0.36 on 22 August and 1.31 on 7 September. Accounting for the uncertainty on the LR of ±25 sr, the AOTs range from 0.34 to 0.39 and from 1.25 to 1.37 on 22 August and 7 September, respectively. The PDR can be more affected than the AEC, mainly when the AEC is smaller (< 0.1 km⁻¹). Nevertheless, in the aeorsol layers, the uncertainties due to the LR is smaller than 2-3%. All these uncertainty sources do not significantly impact the scientific findings.

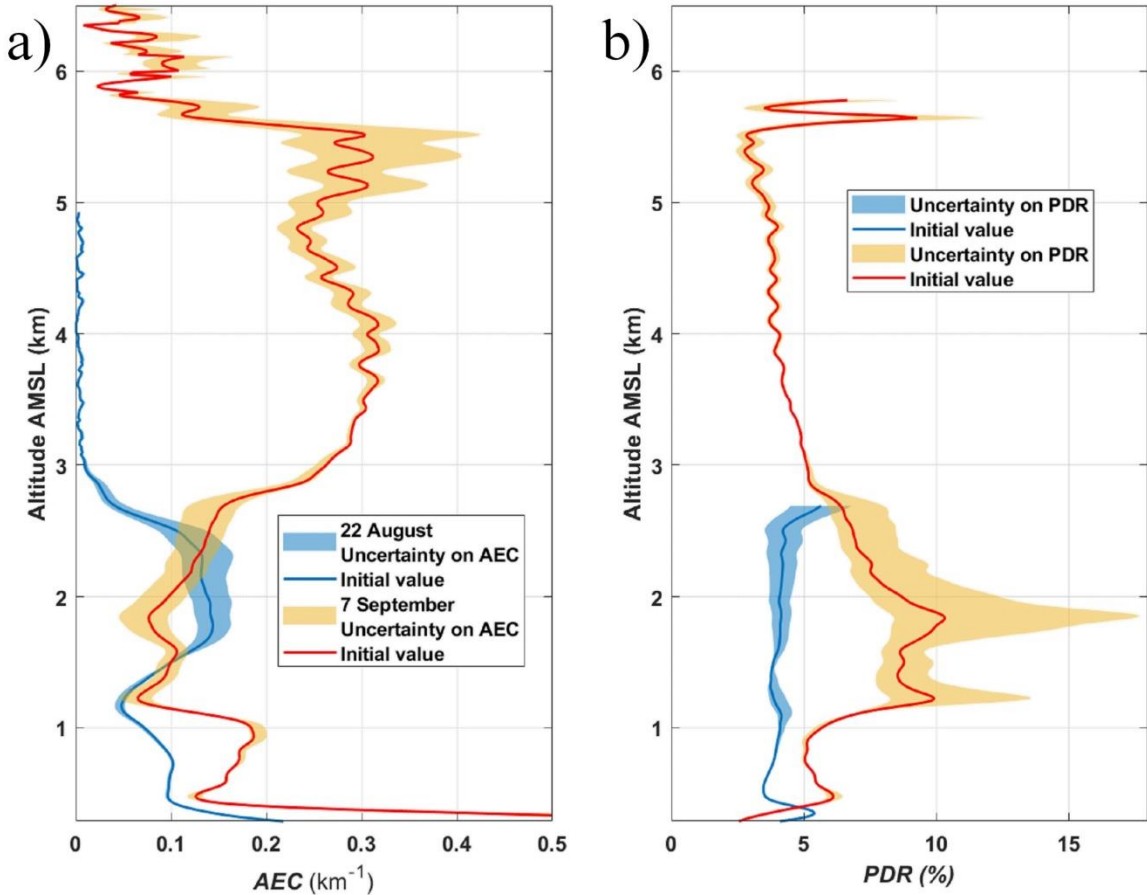


Figure A4: Vertical profiles of the aerosol extinction coefficient (AEC) and particle depolarization ratio (PDR) at 355 nm: on
a) 22 August 2017 and b) 7 September 2017. The shaded areas give the uncertainty linked to the one on the lidar ratio (LR) of
±25 sr as considered for the CALIOP operational algorithm.

 **Figure captions**

Figure 1: Location of the Henties Bay experimental site (in Namibia) on the west African coast. The Walvis Bay airport where
the SAFIRE Falcon 20 aircraft operated during AEROCLO-sA is also indicated. The black rectangle surrounds the area chosen
to average the MODIS-derived AOTs. The Henties Bay and Walvis Bay locations are marked by orange dots.

Figure 2: a) Temporal evolution of the AOT at 550 nm derived from CAMS (black and green solid lines), sun photometer (red
crosses) and MODIS (magenta dots) data. The green solid line shows CAMS AOT extracted on the grid cell centred on Henties
Bay. The black solid line shows the CAMS AOT averaged over 9 grid cells (a 3x3 grid box) centered on Henties Bay. The 3
periods highlighted by the AOT values ($P_1$, $P_2$ and $P_3$) are indicated. b) Temporal evolution of the lidar-derived AOT at 355
nm for the altitude ranges [1500 3000[ m in green, [3000 5000[ m in grey and [5000 6000[ m in red. The total AOT is given in
blue. The vertical bars delimit the daily extremes of AOT.

Figure 3: Vertical profiles of the aerosol extinction coefficient (AEC) and particle depolarization ratio (PDR) at 355 nm with
their uncertainties (horizontal bars) for Period $P_1$: on a) 22 (1400-2300 UTC), b) 23 (1645-2330 UTC) and c) 27 (1545-1700
UTC). The total aerosol optical thickness at 355 nm (AOT) is also given for each profile with its uncertainty.

Figure 4: Vertical profiles of the aerosol extinction coefficient (AEC) and particle depolarization ratio (PDR) at 355 nm with
their uncertainties (horizontal bars) for Period $P_2$: on a) 28 (1030-1230 UTC), b) 29 (1730-2250 UTC), c) 30 (1800-2000 UTC)
and d) 31 (1430-2100 UTC) August 2017. The total aerosol optical thickness at 355 nm (AOT) is also given for each profile
with its uncertainty.

Figure 5: Vertical profiles of the aerosol extinction coefficient (AEC) and particle depolarization ratio (PDR) at 355 nm with
their uncertainties (horizontal bars) for the transition period on 2 September 2017 at a) 0930-1130 UTC and b) 1715-1900
UTC. The total aerosol optical thickness at 355 nm (AOT) is also given for each profile with its uncertainty.

Figure 6: Vertical profiles of the aerosol extinction coefficient (AEC) and particle depolarization ratio (PDR) at 355 nm with
their uncertainties (horizontal bars) for Period $P_3$: on a) 3 (1400-1540 UTC), b) 4 (2330-2400 UTC), c) 5 (1400-1500 UTC)
and d) 6 (0830-1030 UTC) September 2017. The total aerosol optical thickness at 355 nm (AOT) is also given for each profile
with its uncertainty.

Figure 7: Vertical profiles of the aerosol extinction coefficient (AEC) and particle depolarization ratio (PDR) at 355 nm with
their uncertainties (horizontal bars) for Period $P_3$: on a) 7 (1600-1900 UTC), b) 8 (1300-1500 UTC), c) 9 (0900-1200 UTC)
and d) 11 (1040-1140 UTC) September 2017. The total aerosol optical thickness at 355 nm (AOT) is also given for each profile
with its uncertainty.

Figure 8: (a) Distance-height ("curtain-like") evolution of the LNG-derived apparent backscatter coefficient at 532 nm below
the SAFIRE Falcon 20 during the morning flight on 5 September 2017. The location of the dropsonde released over the ocean
is indicated as well as the location of the averaged LNG aerosol extinction coefficient (AEC) profile shown in (b) (between the
2 dotted vertical lines). (b) Vertical profiles of the AEC derived from the airborne lidar at 532 nm (~1000 UTC, blue solid line)
and from the ground-based lidar at 355 nm (~1400-1500 UTC, black solid line).

Figure 9: (a) Wind speed (black solid line), wind direction (coloured dots), RH (blue solid line) and temperature (green solid
line) profiles extracted from ERA5 at 1000 UTC above Henties Bay over a 0.25° by 0.25° grid. (b) Same as (a) but measured
by the dropsonde released over the ocean at 0952 UTC on 5 September 2017.

Figure 10: (a) Same as Figure 6a, but on 6 September 2017. The locations of the two launched dropsondes are also indicated
by arrows. The lidar AEC profile labelled '1' shown in (b) is obtained after inversion of the LNG observations averaged
between the two locations of the two dropsondes. The AEC profile labelled '2' is obtained after inversion of the lidar data
between the northern most dropsonde and the northern end of the Falcon leg. (b) Vertical profiles of the AEC derived from the
airborne lidar at 532 nm (~0830 and ~0900 UTC, for profile '2' (solid blue line) and '1' (dashed blue line), respectively) and
from the ground-based lidar at 355 nm (~0700-0930 UTC, black solid line).

Figure 11: (a) & (b) Same as Figure 7b, but for the dropsondes released at 0843 UTC (to the northwest of Henties Bay,
Dropsonde 2 in Figure 10a) and at 0908 UTC (west of Henties Bay, Dropsonde 1 in Figure 10a).

Figure 12: Time-height evolution of the relative humidity vertical profiles derived from ERA5 above Henties Bay. The grey
vertical lines indicate the time of the ground-based lidar profiles shown in Figure 3-7. The thickness of the grey lines depends
on the averaging period (the thicker the line, the longer the average). The 3 periods highlighted by the AOT values ($P_1$, $P_2$ and
$P_3$) are also indicated. The vertical black lines show the lidar-derived altitude location of the aerosol layer.

Figure 13: Normalized occurrence of the back trajectories starting over Henties Bay at 1200 UTC during periods $P_1$, from the
altitude range [1500 3000[ (a) [3000 5000[ (b) and [5000 6000[ (c), m. The calculations have been made using 6-day isentropic
back trajectories with the HYSPLIT model (courtesy of NOAA Air Resources Laboratory; http://www.arl.noaa.gov) in
ensemble mode. The normaliation is performed with respect to the total number of pixels for a horizontal resolution of 0.5°.

Figure 14: Normalized occurrence of the back trajectories starting over Henties Bay at 1200 UTC during periods $P_2$, from the
altitude range [1500 3000[ (a) [3000 5000[ (b) and [5000 6000[ (c), m. The calculations have been made using 6-day isentropic
back trajectories with the HYSPLIT model (courtesy of NOAA Air Resources Laboratory; http://www.arl.noaa.gov) in
ensemble mode. The normalization is performed with respect to the total number of pixels for a horizontal resolution of 0.5°.

Figure 15: Normalized occurrence of the back trajectories starting over Henties Bay at 1200 UTC during periods $P_3$, from the
altitude range [1500 3000[ (a) [3000 5000[ (b) and [5000 6000[ (c), m. The calculations have been made using 6-day isentropic
back trajectories with the HYSPLIT model (courtesy of NOAA Air Resources Laboratory; http://www.arl.noaa.gov) in
ensemble mode. The normalization is performed with respect to the total number of pixels for a horizontal resolution of 0.5°.

Figure 16: 6-days isentropic back trajectories starting over Henties Bay on 6 September at 1200 UTC. They are computed by
the HYSPLIT model (courtesy of NOAA Air Resources Laboratory; http://www.arl.noaa.gov) in ensemble mode. The time to
arrival above the South America is indicated. The altitude of back trajectories along the route is given by the colour bar.

Figure 17: MODIS-derived AOT at 550 nm on (a) on 3 September 2017 with wild fire hotspots over both South Africa and
South America, (b) on 5 September 2017 and c) 6 September 2017. The ERA5 wind field at 500 hPa on each day have been
added in black.

Figure A1: Overlap factor of the ALS (continuous black line) and its standard deviation (grey area).

Figure A2: Apparent backscatter coefficient (black solid lines) profiles obtained from the ASL lidar in Henties Bay on: a) 22
August 2017 between 1400 and 2300 UTC, b) 28 August 2017 between 1030 and 1230 UTC, c) 7 September 2017 between
1600 and 1900 UTC, and d) 8 September 2017 between 1300 and 1500 UTC. The red lines correspond to the molecular
backscatter coefficient computed using ERA5 data. The grey area is the standard deviation linked with the statistical error (the
shoot noise and the atmospheric variability).

Figure A3: a) CALIOP-derived aerosol typing for the night time orbit (10.2017-08-28T00-08-17ZN) on 28 August 2017. b)
CATS-derived aerosol typing for the night time orbit (2017-08-30T00-32-37T01-18-13UT) on 30 August 2017. The latitudinal
location of the Henties Bay site is given by the vertical black line. Inserted panels in a) and b) show the position of the space-
borne lidar tracks over southern Africa and with respect to Henties Bay.

Figure A4: Vertical profiles of the aerosol extinction coefficient (AEC) and particle depolarization ratio (PDR) at 355 nm: on
a) 22 August 2017 and b) 7 September 2017. The shaded areas give the uncertainty linked to the one on the lidar ratio (LR) of
$\pm 25$ sr as considered for the CALIOP operational algorithm.