# Peer review of "Evidence of the complexity of aerosol transport in the lower"

_Atmospheric Chemistry and Physics, 2019_

## Referee Comment (RC1) · Michael Diamond (Referee) · 30 Jun 2019

**Review of "Evidence of the complexity of aerosol transport in the lower troposphere on the Namibian coast during AEROCLO-sA" by Patrick Chazette et al.**

**Summary of manuscript:**

This paper uses ground-, aircraft-, and space-based lidar systems to evaluate the vertical structure of aerosol plumes around Henties Bay, Namibia, and the surrounding southeast Atlantic region during the time of the AEROCLO-sA campaign. The observations are divided into three periods with different column loadings and vertical distributions characteristic of each period. Back trajectories are run using the HYSPLIT model at the primary plume altitudes for each period to evaluate transport pathways. The authors interpret the trajectories from the highest altitudes in period 3 as evidence that smoke from South America contributes significantly to the column loading over the southeast Atlantic and argue this must be taken into account in regional studies of biomass burning aerosol radiative effects.

Unfortunately, the trajectory analysis does not directly compare the same altitudes between the three periods, making it difficult to determine whether the transport pathways truly are distinct between these three periods. In addition, the result that elevated smoke in period 3 is of South American origin may be a misinterpretation of the ensemble HYSPLIT trajectories.

**Recommendation:** This manuscript requires major revisions, particularly in the analysis of the HYSPLIT trajectories. It will, however, be a valuable addition to our knowledge of the complex aerosol vertical structure in the southeast Atlantic region and should be accepted upon adequate revision.

**Major issues:**

The major issue with the manuscript is the interpretation of the HYSPLIT trajectories, and in particular the interpretation of the result that elevated smoke from South America is of major importance for radiation over the southeast Atlantic.

Section 5 and Figure 11 cannot be interpreted as showing differences in transport pathways between periods, as the authors attempt to do, because the comparisons are "apples-to-oranges" in that the trajectories are initialized at very different altitudes. Even if the circulation had been perfectly steady throughout all three periods, vertical wind shear alone would lead to apparent differences using this methodology. The correct "apples-to-apples" comparison would be to compare trajectories at the same altitudes for all three periods.

One solution could be to divide Figure 11 into three figures: one with the trajectories from 1500-3000 m for all three periods, another from 3000-4500 m, and a third from 4000-6000 m. 2000 m is a fairly large vertical area to lump together, so I would further suggest subdividing from the original three altitude bins. For instance, binning by 1 km increments (1.5-2.5 km, 2.5-3.5 km, 3.5-4.5 km, 4.5-5.5 km, 5.5-6.5 km) could be more illustrative. These are all just suggestions: there are many different ways to expand the analysis to make more meaningful comparisons between the periods.

Another concern relates to the interpretation of the trajectory ensembles. Figure 11c) is currently being interpreted by the authors as showing that some of the elevated smoke is coming from South America. However, there appear to be a significant number of trajectories that are limited to re-circulations around the African continent. Are all of the re-circulating trajectories from below ~5 km? If not, then the ensemble is actually telling us that it is plausible that the air came from South America but equally plausible that it re-circulated locally. In a similar vein, Figure 12 shows that South America is a plausible source for the elevated smoke observed in period 3 but certainly does not prove that this must be the source.

Even if a more rigorous evaluation of the trajectories does confirm that there are meaningful circulation differences between the three periods (which is extremely likely) and that the elevated plume in period 3 is very likely of South American origin (which I am open to but more skeptical of), the significance of these results is either inflated or incompletely explained. In the conclusion, the idea that there may be some influence from elevated South American smoke is deemed "of paramount importance" for the region in the context of aerosol-radiation and aerosol-cloud interactions. However, even accepting the hypothesis that the elevated smoke is South American in origin, it still only comprised ~10% of the column loading of aerosol and is too high in altitude to plausibly influence low cloud microphysics or have relevant semi-direct effects (unless high clouds occur more frequently over Henties Bay than I assume). Given the pre-existing uncertainties in aerosol loading, vertical distribution, and optical properties over the region, I find it hard to argue that an occasional 10-15% contribution to the direct radiative effect is "fundamental."

Major revisions likely need to be made to the abstract and conclusion based on the updated trajectory analysis. Below I detail additional specific comments that are independent from the issues identified above.

**Specific comments:**

1. Page 2, Line 38: What metric are you using to determine that southern Africa is the "most important source" of biomass burning aerosol?
2. Page 2, Lines 49-51: There is now an accumulation of evidence from the LASIC (e.g., Zuidema et al. 2018, GRL) and ORACLES (e.g., Diamond et al. 2018, ACP) campaigns that the MBL in the southeast Atlantic often contains quite a bit of smoke.

In particular, the ORACLES campaign in September 2016 observed some very smoke-polluted MBLs not far from the Namibian coast. The MBL-FT dichotomy in this sentence is a bit oversimplified.

3. Page 2, Line 53: The papers from Costantino & Bréon do not show that clouds in this region are particularly sensitive to aerosol increases — indeed, their aerosol-cloud interaction parameter estimates are well within the range of the other literature they cite. They do show an apparent inverse relationship between cloud effective radius and aerosol index when smoke is near cloud tops, which is consistent with a widespread Twomey effect. If the point here is more that marine cloud radiative properties should be particularly sensitive to aerosol increases, the paper from Oreopoulos & Platnick (2008) cited below may be a more appropriate reference, among other suitable choices.

4. Page 2, Line 56-60: The vertical distribution is also incredibly important for indirect effects, not only semi-direct effects. Without contact between the plume bottom and cloud tops, smoke cannot entrain into the MBL and influence cloud microphysical properties. The Costantino & Bréon papers would be good to cite here. Diamond et al. (2018) and Painemal et al. (2014) may also be relevant.

5. Page 2, Line 72: What does "aerosol activation" mean in this context?

6. Page 3, Line 89: Namib desert, not "Namibia" desert

7. Page 3, Line 91: Please define "ALS."

8. Page 3, Line 93: Please define "LEANDRE."

9. Page 3, Table 1: Please explain what the "X" and "-" symbols mean in the table caption.

10. Page 5, Figure 1: In the caption mention that the Henties Bay and Walvis Bay locations are marked by orange dots.

11. Page 8, Line 200: How does the non-colocation of the MODIS area average and Henties Bay affect your results, if at all?

12. Page 8, Line 205: Move "only" to after "aerosols are."

13. Page 9, Lines 251-254: This period of disagreement between the observations and CAMS may be worth exploring further. What is the circulation like then? Does it seem like the FT air is being sourced from a non-biomass burning affected area, or is there perhaps loss of aerosol occurring (e.g., precipitation scavenging) that CAMS may not be capturing?

14. Page 9, Line 265: Why do you refer to the observations as biased with respect to CAMS? Couldn't the CAMS value be off? It might be helpful to explain why either estimate may be different than "truth."

15. Page 9, Table 3: It would be helpful to explain what the uncertainty range is in the caption. Also, for the profiles encompassing a long period of time, how much of the uncertainty in average value comes from remote sensing uncertainties versus real variability over the time period?

16. Page 10, Table 3: How are these profiles divided? For instance, why are the 22/08 profiles divided at 1607?

17. Pages 12-13, Figures 3-4: I don't understand why the figures are divided in this manner. Also, not all potential profiles from Table 3 appear to be plotted (e.g., it

seems like there is GBL data for 22/08 from 1608-2400 that is not included). It might make more sense to divide this into a number of separate figures for each period.

18. Page 14, Figure 5: The color scale here makes the figure very difficult to read. Perhaps using one that ranges from a very light color to a darker color for high AOT would both be more intuitive but also make it easier to read the wind markers.

19. Page 14, Line 294: What do you mean by "evolution"? That term may be misleading because you are not really showing changes over time but rather a time-space cross section following the aircraft.

20. Page 15, Lines 307-309: I cannot find evidence for this statement in the Haywood et al. (2003) paper cited. Parmar et al. (2008) does establish that water vapor is emitted at fire sources along with carbonaceous species but does not claim that this could humidity an entire well-mixed continental boundary layer. Why would the high relative humidity not simply be characteristic of continental air whereas low humidity air be from subsiding tropical or midlatitude air that has been depleted of moisture via prior precipitation?

21. Page 16, Line 342: In what way does elevated RH suggest the aerosol layer must be distinct?

22. Page 16, Line 362: In what way does the existence of non-negligible AEC values suggest that the aerosol has a different origin than the other aerosols observed?

23. Page 18, Figure 9: It would be helpful to make clear which profile corresponds to which dropsonde in Figure 8.

24. Page 19, Line 395: Do you mean "back-trajectory" instead of "retro-trajectory"?

25. Page 20, Line 447: What do you mean by "nebulosity"?

26. Page 22, Figure 12: I would suggest using whichever color scale you choose for Figure 5 here as well.

27. Page 23, Lines 482-483: Couldn't high observed AOT also be due to aerosol humidification at high RH, versus thin clouds?

28. Page 23, Lines 497-498: What do you mean by "trapped in the FT"?

29. Page 23, last paragraph in its entirety: This is very interesting, but not appropriate to introduce and discuss only in the conclusion. This could potentially be a great addition to the paper as a separate section with some new analysis of the meteorological fields to provide evidence for the claims made.

**References:**

Diamond, M. S., Dobracki, A., Freitag, S., Small Griswold, J. D., Heikkila, A., Howell, S. G., . . . Wood, R. (2018). Time-dependent entrainment of smoke presents an observational challenge for assessing aerosol–cloud interactions over the southeast Atlantic Ocean. *Atmospheric Chemistry and Physics, 18*(19), 14623-14636. doi:10.5194/acp-18-14623-2018

Haywood, J. M., Osborne, S. R., Francis, P. N., Keil, A., Formenti, P., Andreae, M. O., & Kaye, P. H. (2003). The mean physical and optical properties of regional haze dominated by biomass burning aerosol measured from the C-130 aircraft during

SAFARI 2000. *Journal of Geophysical Research: Atmospheres, 108*(D13). doi:10.1029/2002jd002226

Oreopoulos, L., & Platnick, S. (2008). Radiative susceptibility of cloudy atmospheres to droplet number perturbations: 2. Global analysis from MODIS. *Journal of Geophysical Research, 113*(D14). doi:10.1029/2007jd009655

Painemal, D., Kato, S., & Minnis, P. (2014). Boundary layer regulation in the southeast Atlantic cloud microphysics during the biomass burning season as seen by the A-train satellite constellation. *Journal of Geophysical Research: Atmospheres, 119*, 11288-11302. doi:10.1002/2014JD022182

Parmar, R. S., Welling, M., Andreae, M. O., & Helas, G. (2008). Water vapor release from biomass combustion. *Atmos. Chem. Phys., 8*(20), 6147-6153. doi:10.5194/acp-8-6147-2008

Zuidema, P., Sedlacek III, A. J., Flynn, C., Springston, S., Delgadillo, R., Zhang, J., . . . Muradyan, P. (2018). The Ascension Island boundary layer in the remote southeast Atlantic is often smoky. *Geophysical Research Letters, 45*, 4456–4465. doi:10.1002/2017gl076926

---

## Referee Comment (RC2) · Anonymous Referee #2 · 8 Jul 2019

The manuscript describes observations from ground-based and airborne lidar, sunphotometer, and dropsondes made during the AEROCLO-sA field campaign on and near the Namibian coast. Data analysis is aided by coincident satellite observations, HYSPLIT back-trajectory analysis and two additional models. These data and analysis are used to illustrate the vertical distribution and the spatial and temporal variability of the aerosol amount during the campaign. A finding of the paper is that while most of the aerosol is of African origin, a portion has been transported long distances from South America.

General comments

[Figure]

The manuscript is interesting and generally easy to read, although the organization of sections 3 and 4 could be made to flow better. The clear organization in the introduction and the conclusions sections were particularly appreciated. The paper includes appropriate acknowledgement of prior work.

On the other hand, I find that some of the conclusions, both major and minor, are over-interpreted, that is, drawn from slight evidence that is not well supported or should be understood to have high uncertainty.

The argument that the South American aerosol layer is "distinct" is unconvincing and the conclusion that it is of significant importance for climate modeling is premature. Yet, aerosol transported from South America to Africa is still interesting enough to highlight, even without this. Better to be very straightforward about what is and is not known about it, to inform and motivate future study.

Additionally, some lack of clarity makes it difficult to understand the details of the lidar retrieval; in particular the derivation of the lidar ratio, and the associated uncertainties. The lidar retrieval is subject to large uncertainties because of the underdetermined nature of the problem, but there is a lack of useful information about the size of the uncertainties or their impact on the conclusions.

Specific comments

119 "between about 1 and 4 hours". Actually, several of these averages are 7 or 8 hours long, so this should be indicated correctly. Even 4 hours seems like quite a long time, considering the variability of aerosol in the region. Was there any need for filtering or any analysis to check for stationarity?

Section 2.3 The Bruneau et al. 2015 paper referenced at 135 is missing from the bibliography. I think I know which paper you mean, though, and it says this instrument has HSRL capability at 355 nm. If so, why does it require the retrieval procedure that's used for the ground-based elastic backscatter lidar? Even if I'm wrong about the HSRL,

the appendix says that the lidar has a 355 nm channel. So then why is the 355 nm data not shown in figures 6b and 8b? (if it's available, it would be very helpful there!)

Section 3. I found the organization of section 3 hard to follow. It goes back and forth multiple times between different subjects. Please consider reorganizing to make one paragraph about the overall AOT trend, one comparing P1 to P2, and another separate paragraph comparing P2 to P3.

215-218 statement that lidar ratios in the free troposphere "suggest the presence of terrigenous aerosols mixed with smoke. This is coherent with the polluted dust type inferred from the CALIOP observations." Is this circular reasoning? The Appendix suggests that only the lidar ratio in the PBL is retrieved and that the lidar ratio in the free troposphere is taken directly from the inferred lidar ratio in CALIPSO or CATS.

The presence of dust in the smoke layer isn't very consistent with the AEROCLO-Sa ground-based lidar measurements in Figure 3, which shows quite small values of the particle depolarization ratio within the smoke plume for all profiles.

Even if I'm misunderstanding how the constrained retrieval works on the ground based lidar, I believe it's likely that the retrieved lidar ratio in the free troposphere must have a large degree of uncertainty which probably precludes making a distinction between smoke and polluted dust, whereas the uncertainty on the particle depolarization ratio is likely to be quite reasonable. Does this conform to your analysis of systematic errors and uncertainties?

220 "being remobilized by pyroconvection and mixed with BBA before being transported aloft". How do you know these details about the origin of the dust? What evidence is there of pyroconvection?

217 What does the citation Flamant et al. 1998 refer to here? Is this a typo? (Also, the paper is missing from the bibliography).

232 "LR values in the FT evolving from 55 to 70 sr". I believe the lidar ratio in the

free troposphere is assigned based on aerosol type that is somehow inferred from the CALIPSO and CATS aerosol type inferences. It would be better, then, to directly refer to a change in inferred aerosol type rather than an evolution of the lidar ratio, which implies more quantitative detail than is actually available.

238 "LR values observed in the PBL". Please be specific about which retrieval you mean. The combined lidar-sunphotometer retrieval or the sunphotometer only retrieval from the previous sentence?

239 Again the presence of dust would be expected to be accompanied by an elevated particle depolarization ratio which is not evident.

245-247 The relatively high particle depolarization ratio in the altitude region between the smoke plume and the PBL seems unlikely to be significant compared to its uncertainty. The systematic error of particle depolarization ratio increases dramatically for low backscatter ratios (see, e.g. Freudenthaler et al. 2016, Burton et al. 2015). If you have done a calculation of the expected systematic uncertainty and believe these to be significant, then that should be included to support your interpretation.

Table 3 What is the number after the plus-and-minus sign?

Table 3 Why is the lidar ratio consistently only a few different values? In the PBL, the retrieval is described as a constrained retrieval, so I expected it to be able to take a continuous range of values.

Figure 3 Add error bars? I expect significant amount of variability due to the very long averaging and also a significant amount of systematic uncertainty. Even if only a coarse estimate of systematic uncertainty can be included, this would be valuable for interpreting the significance of features in the profiles.

316ff. Somewhere in this paragraph please state the distance between the profiles.

318 "operate at different wavelengths". Table A1 indicates that LNG also operates at 355 nm. Can you show 355 nm retrievals from LNG as well? It might help establish

that there really is correspondence between the airborne and ground-based retrieved extinction and give better support that the differences are not due to the wavelength.

324 "the difference can be explained". There are multiple factors affecting this difference, including the approximately four hour time difference between the profiles, which has not been addressed. Any comment about the role of the time difference? In any case, soften the statmeent. Maybe "More important is the regional scale circulation"

328-329. Does the CAMS model also show this difference between the aerosols over the land and ocean? You previously made an apparently conflicting point that CAMS indicates homogeneity in the aerosol around Henties Bay.

335-336 "the structure of [profile 2] being coherent with the ground-based AEC profile". Not convincingly so. Also, this statement is apparently contradicted at line 347. Is this a typo?

340-341 I'm confused by the indiation of 20% RH above the BBA layer. First, is this referring to 9b or 9a? Either way, I see that it doesn't drop dramatically to zero like 8b, but I do not see 20% RH for any part of the profile above the strongest gradient (top of the BBA) and even the highest value I do see (more like 10%?) is only for a few hundred meters.

341-342 "suggests the presence of a distinct aerosol layer". Is the key point that there is aerosol present? In which case, I think the lidar profile itself (assuming the signature of the aerosol is significant above the retrieval uncertainty and not just an artifact of the retrieval) is evidence of aerosol and the RH is mostly irrelevant. Or is the key here that the aerosol is "distinct". I don't understand how either the extinction profile or the relative humidity indicate that the aerosol is physically distinct from the African BBA layer.

344 I don't see that there is much agreement in the "structure" of the PBL. You might just say more simply that the apparent height of the PBL in the aerosol agrees with the

location of the gradient in RH.

353-354 Not clear what "distinct aerosol layer above the main BBA layer" means. Please specify the altitudes you're talking about. To me it looks like the main BBA layer extends to higher altitude in the Henties Bay uplooking lidar profile and in profile (1), while (2) shows a lower top for that layer. I don't see it as a separate, secondary layer. I can easily agree if you say "additional aerosol above 5 km" which is more unambiguous.

360 "related to ... hygroscopic growth ... particularly below the BBA layer where RH is high in '2'". Again, I'm having some trouble following the roundabout wording here, but I take "below the BBA layer where RH is high" to mean "below 1 km". Yet there is no enhancement of extinction in profile '2' below 1 km. So, either this statement is incorrect, or the wording is so confusing that the meaning has been lost for at least one reader.

361 How robust are the estimates of the AOT of the layer above 5 km to the uncertainties in the lidar retrieval?

362 "suggest that the aerosol may have a different origin". Again I do not understand what evidence supports this statement.

392-393 "RH values ... may be an indication of the transport of BBA from a different origin." Again, I don't understand why the RH values should be taken as evidence of a different origin. I believe you have stronger evidence in the back-trajectories. It would be better not to state this conclusion until it has actually been supported. At most, for this sentence, I would say something like that the difference in RH is an indicator that the meterology has changed (if indeed that is what you believe) and that that will be shown in the next section.

416 "are transported very rapidly" I don't understand how to interpret the figure to conclude that the aerosols are transported rapidly. Since the trajectories are more

tightly distributed around the point of origin here than in the other panels, and yet all 3 panels show back-trajectories for the same 6-day duration, I infer that in this panel they moved more slowly.

423 "They correspond to air masses arriving above 5000 m AMSL over Henties Bay". Please show this by splitting 11(c) into panels above and below 5000 m. Since you treat this point as important, it should be shown explicitly.

Figure 11 caption. What does "normalized occurrence" mean (normalized to what)? Does this imply that the scale does not translate to the same number of back-trajectory points in each of the three panels?

502 "of paramount importance" should be toned down. "Highlighting" the transported aerosol is interesting and worthwhile but not of "paramount importance" until it is shown to be climatically significant and not already captured in climate models.

750 Define "apparent backscatter coefficient", preferably using an equation. Not knowing for sure what this is is making it hard for me to follow the rest of the section.

751-752 "must follow the slope of the molecular bacskcattering". This is not well explained. Won't there be attenuation of the signal by air molecules and aerosol at lower levels? (wheras the molecular backscattering from ERA5 is not affected)?

759-760 "the error remains below 2-3%" Which error remains below 2-3%? The error in the molecular backscatter, or the aerosol backscatter derived from it, or the aerosol extinction derived from it?

779 "uncertainty sources are exhaustively quantified". Given that you have access to a methodology for quantifying the uncertainty, it should definitely be quantified for the data presented in this paper. Depending on how big the retrieval uncertainites are, some of your conclusions can be affected, as discussed elsewhere in this review, so it's not just an academic exercise.

781-782 "using aerosol typing determined from the CALIOP and CATS measurements". How to you obtain the free-troposphere aerosol type and lidar ratio for cases where there is no satellite overpass in table 1? How do you get them for cases where CALIPSO or CATS or both infer multiple types in different pixels in the region (like the 31 August case, for instance).

795-798 It seems like a stretch to infer from just a few individual cases of transported smoke from a totally different source as presented by Muller et al. 2007 that the lidar ratios for dust and smoke at 355 nm are the same as 532 nm. Is this the only relevant paper?

800 "matches best the AOT from the sun photometer". At the lidar wavelength?

812 "uncertainties of 2% on the PDR". Do these uncertainties include sources of systematic error? Doesn't the uncertainty level depend strongly on the amount of aerosol? So, is there a minimum aerosol amount to get the quoted 2% error?

Minor comments

52 "as also mixes". Should this be "and also mixes"?

91 says the ground-based lidar is an ALS 300, but in the appendix it says ALS 450. Is there a difference between these two designations?

93 please indicate what type of lidar the LNG lidar is.

98 UTC is "coordinated universal time"

154-155 a better reference for CALIOP 4.10 typing would be Kim et al. 2018

175-176 "The standard deviation on the AOT" should be "the uncertainty in the AOT".

205 change word order "when aerosols are only observed"

209 refers to lines highlighted in green in the table. I don't see highlighting. Does this need to be reworded?

210 "averaged AOT of ∼0.15" Indicate which instrument or model this refers to.

[Figure]

213 reference to Angstrom exponent. Which instrument or model?

227-228 "between the 2 periods". Between P1 and P2?

234 add reference for sun photometer retrievals of lidar ratio

250 "match perfectly" is an overstatement. Please reword. It's better to be quantitative anyway.

256 "positively biased" reword to avoid "biased" and use a more neutral phrase like "larger than". You've given reasons why either the model or the observations (or both) could be incorrect in this case.

Table 3 caption, UAL is used here where elsewhere it is described by the label FT. Is the difference signficant? If not, please pick just one.

Table 3 column "AE" please indicate which instrument this is from.

Figure 2 It would be easier to understand the x-axis if tick marks were an integer number of days instead of 1.5 days.

292 "where" should be "were"

294 I believe this is probably attenuated backscatter coefficient rather than aerosol backscatter coefficient. This is an important distinction.

308 remove "wood". I think it's more likely grasses and crops here than wood, but either way it's speculation. "Combustion" is sufficient.

323 "apparition" should be "appearance"

334 "offshore airborne lidar measurements from Henties Bay". Replace with "the profile west of Henties Bay" for clarity.

351 "below the base of the BBA layer observed further south". Please simply state the altitude for easier reading.

352 "The RH above the top of the BBA layer". Again, it would be helpful to be more specific about what altitude and which RH profile you are referring to, for easier reading.

Figure 6 caption "apparent bacskcatter coefficient". Is this the same as "attenuated backscatter coefficient"? Please add a reference for where this quantity is defined.

386 add "at Henties Bay" for clarity

408 "Illustrating" not "corroborating" (this isn't independent evidence)

412 "between the 3 regimes" perhaps rather than "periods" since the different regimes include different choice of ending altitude as well as choice of ending date.

412 similarly "For the layers observed during P1" for the start of the next sentence, since you are only showing back-trajectories corresponding to observed aerosol layers.

414 "consistent" not "coherent"

484 "illustrate" or "show", not "evidence"

497 probably delete "trapped" or reword to make the meaning clearer

498 I don't understand the meaning of "in link with the dominant transport patterns of lifted aerosols". Please reword.

504 replace "fundamental" with "necessary". Fundamental has a connotation of primary importance in addition to meaning "necessary" and therefore sounds like an overstatement.

Table A1 add post-processing temporal resolution

749 Replace section title with "Overlap correction" or something similar, since this is only one of many aspects of calibration but the only one discussed.

761 "average lidar profiles".

766 replace "perfect coincidence" with "very good agreement"

777 replace "exogeneous" with "external"

798 replace "standard deviation of LRs derived from spaceborne lidars" with "expected uncertainty in LRs for spaceborne lidars". Lidar ratio is not dervied from CALIOP and CATS; they do not measure or retrieve lidar ratio. It's an input, not an output, of the retrieval.

802 replace "coherent" with "consistent"

807 replace "inverse" with "invert"

Kim, M. H., Omar, A. H., Tackett, J. L., Vaughan, M. A., Winker, D. M., Trepte, C. R., Hu, Y., Liu, Z., Poole, L. R., Pitts, M. C., Kar, J., and Magill, B. E.: The CALIPSO version 4 automated aerosol classification and lidar ratio selection algorithm, Atmos. Meas. Tech., 11, 6107-6135, 10.5194/amt-11-6107-2018, 2018.

Freudenthaler, V.: About the effects of polarising optics on lidar signals and the ?90Ă-calibration, Atmos. Meas. Tech., 9, 4181-4255, 10.5194/amt-9-4181-2016, 2016.

Burton, S. P., Hair, J. W., Kahnert, M., Ferrare, R. A., Hostetler, C. A., Cook, A. L., Harper, D. B., Berkoff, T. A., Seaman, S. T., Collins, J. E., Fenn, M. A., and Rogers, R. R.: Observations of the spectral dependence of linear particle depolarization ratio of aerosols using NASA Langley airborne High Spectral Resolution Lidar, Atmos. Chem. Phys., 15, 13453-13473, 10.5194/acp-15-13453-2015, 2015.
* * *

---

## Author Comment (AC1) · 17 Sep 2019

**Reviewer #1**

Unfortunately, the trajectory analysis does not directly compare the same altitudes between the three periods, making it difficult to determine whether the transport pathways truly are distinct between these three periods. In addition, the result that elevated smoke in period 3 is of South American origin may be a misinterpretation of the ensemble HYSPLIT trajectories.

Back trajectories were calculated based on the layers of aerosols observed by lidar, which are different from one period to another, the objective being to identify their origin. For this reason, back trajectory levels change from one period to the next. In response to the reviewer's remark, we have introduced the back trajectories associated with the 3 layers of aerosols observed by lidar in the free troposphere for the 3 periods (see the new Figure 13 where we show 9 sub-figures (3 height ranges x 3 periods) instead of 3 in the original version of the manuscript).

The major issue with the manuscript is the interpretation of the HYSPLIT trajectories, and in particular the interpretation of the result that elevated smoke from South America is of major importance for radiation over the southeast Atlantic.

This is indeed one of the important points of the article, but the general characterization of aerosol plumes and their high variability over time is also important. However, we agree that the part concerning the importance for radiation of the southeast Atlantic needs to be backed with a bit more evidences. Therefore, we have put more nuances in our conclusions.

Section 5 and Figure 11 cannot be interpreted as showing differences in transport pathways between periods, as the authors attempt to do, because the comparisons are "apples-to-oranges" in that the trajectories are initialized at very different altitudes. Even if the circulation had been perfectly steady throughout all three periods, vertical wind shear alone would lead to apparent differences using this methodology. The correct "apples-to-apples" comparison would be to compare trajectories at the same altitudes for all three periods.

One solution could be to divide Figure 11 into three figures: one with the trajectories from 1500-3000 m for all three periods, another from 3000-4500 m, and a third from 4000-6000 m. 2000 m is a fairly large vertical area to lump together, so I would further suggest subdividing from the original three altitude bins. For instance, binning by 1 km increments (1.5-2.5 km, 2.5-3.5 km, 3.5-4.5 km, 4.5-5.5 km, 5.5-6.5 km) could be more illustrative. These are all just suggestions: there are many different ways to expand the analysis to make more meaningful comparisons between the periods.

As mentioned above, we decided to calculate the back trajectories for the aerosol layers observed by the lidar. The calculations were done every 500 m on the vertical over Henties Bay in the original manuscript and are now done every 250 m in the revised version of the paper, for the sake of capture all the possible transport pathways. Figure 11 is representative in the sense that it uses the probability of air mass passages at each point and that this calculation is performed over each period for more than 500 back trajectories. This is the choice we had to make in order to synthesize the information and not present too many back trajectories. This aspect is better explained in Section 5. The reviewer will find in the new version the results over the 3 periods for the same altitude ranges: [1500 3000[, [3000 5000[ and [5000 6000[ m

AMSL. Original Figure 11 (now Figure 13) was therefore effectively divided into 9 sub-figures. The text has been revised accordingly.

Another concern relates to the interpretation of the trajectory ensembles. Figure 11c) iscurrently being interpreted by the authors as showing that some of the elevated smoke is coming from South America. However, there appear to be a significant number of trajectories that are limited to re-circulations around the African continent. Are all of the re-circulating trajectories from below ~5 km? If not, then the ensemble is actually telling us that it is plausible that the air came from South America but equally plausible that it re-circulated locally. In a similar vein, Figure 12 shows that South America is a plausible source for the elevated smoke observed in period 3 but certainly does not prove that this must be the source.

The reviewer is right, this aspect is poorly explained in the text and is not visible in old Figure 11. This aspect is now better described, mainly for the period P$_3$ when we have separated what happens under 6 km AMSL and above.

Even if a more rigorous evaluation of the trajectories does confirm that there are meaningful circulation differences between the three periods (which is extremely likely) and that the elevated plume in period 3 is very likely of South American origin (which I am open to but more skeptical of), the significance of these results is either inflated or incompletely explained. In the conclusion, the idea that there may be some influence from elevated South American smoke is deemed "of paramount importance" for the region in the context of aerosol-radiation and aerosol-cloud interactions. However, even accepting the hypothesis that the elevated smoke is South American in origin, it still only comprised ~10% of the column loading of aerosol and is too high in altitude to plausibly influence low cloud microphysics or have relevant semi-direct effects (unless high clouds occur more frequently over Henties Bay than I assume). Given the pre-existing uncertainties in aerosol loading, vertical distribution, and optical properties over the region, I find it hard to argue that an occasional 10-15% contribution to the direct radiative effect is "fundamental."

Yes, the contribution of aerosols from South America should not represent more than 10-15% of the total aerosol load for mainly 2 days, on 6 and 7 September. We agree that they will not contribute significantly to interactions with low-level clouds and we have not presented it in this way. This aspect is now better specified in the text where Figure 2b was introduced, which gives the temporal evolution of AOTs at 355 nm per layer. The aerosols in the layer [5000 6000[ m will nevertheless likely have a radiative impact that has yet to be correctly estimated, but this is not the subject of this paper. It is also necessary to see how often this kind of phenomenon can occur, in relationship with cut-off lows. In order to comply with the referee's comment, we have agreed to be more cautious in the conclusions regarding the "fundamental" aspect of the direct radiative effect related to aerosols from South America.

Page 2, Line 38: What metric are you using to determine that southern Africa is the "most important source" of biomass burning aerosol?

Our metric is the AOT. This has been now specified.

Page 2, Lines 49-51: There is now an accumulation of evidence from the LASIC (e.g., Zuidema et al. 2018, GRL) and ORACLES (e.g., Diamond et al. 2018, ACP) campaigns that the MBL

in the southeast Atlantic often contains quite a bit of smoke. In particular, the ORACLES campaign in September 2016 observed some very smoke-polluted MBLs not far from the Namibian coast. The MBL-FT dichotomy in this sentence is a bit oversimplified.

The references have been added and we have added the possible presence of BBA within the MBL.

Page 2, Line 53: The papers from Costantino & Bréon do not show that clouds in this region are particularly sensitive to aerosol increases — indeed, their aerosol cloud interaction parameter estimates are well within the range of the other literature they cite. They do show an apparent inverse relationship between cloud effective radius and aerosol index when smoke is near cloud tops, which is consistent with a widespread Twomey effect. If the point here is more that marine cloud radiative properties should be particularly sensitive to aerosol increases, the paper from Oreopoulos & Platnick (2008) cited below may be a more appropriate reference, among other suitable choices.

The reference has been changed. Thank you for suggesting this.

Page 2, Line 56-60: The vertical distribution is also incredibly important for indirect effects, not only semi-direct effects. Without contact between the plume bottom and cloud tops, smoke cannot entrain into the MBL and influence cloud microphysical properties. The Costantino & Bréon papers would be good to cite here. Diamond et al. (2018) and Painemal et al. (2014) may also be relevant.

Agree. We have added this information. Thank you for suggesting this.

Page 2, Line 72: What does "aerosol activation" mean in this context?

It is not the "aerosol activation" but the activation of their sources. The term "activation" has been replaced by "emissions".

Page 3, Line 89: Namib desert, not "Namibia" desert

The correction has been done.

Page 3, Line 91: Please define "ALS."

The information has been done.

Page 3, Line 93: Please define "LEANDRE."

LEANDRE is now defined.

Page 3, Table 1: Please explain what the "X" and "-" symbols mean in the table

caption.

X was meant to indicate when coincident AERONET and lidar were available. X has been replaced by "yes" or "No".

Page 5, Figure 1: In the caption mention that the Henties Bay and Walvis Bay

locations are marked by orange dots.

Thanks for picking this up. This has been added.

Page 8, Line 200: How does the non-colocation of the MODIS area average and

Henties Bay affect your results, if at all?

The accuracy of MODIS data is much better over the sea, we have insisted on this in both Section 2.4.2 and Section 3.1. Henties Bay is a coastal site influenced by sea breeze, so we have considered it more representative to take an average of the AOTs offshore the site. Local aerosol production can affect the measurements, especially in the event of dusts being lifted. Nevertheless, from observations of depolarization lidar, we have seen very few such situations.

Page 8, Line 205: Move "only" to after "aerosols are."

The correction has been done, the section 3 has been modified.

Page 9, Lines 251-254: This period of disagreement between the observations and CAMS may be worth exploring further. What is the circulation like then? Does it seem like the FT air is being sourced from a non-biomass burning affected area, or is there perhaps loss of aerosol occurring (e.g., precipitation scavenging) that CAMS may not be capturing?

First, we highlight that no precipitation event was recorded during the field campaign, so that we can exclude any CAMS misrepresentation of wet deposition processes. Similarly, the contribution from non-biomass aerosol can be excluded as well, because CAMS simulates very low dust AOT during the campaign, with only a peak on 3-5 Sep. We highlight that CAMS total AOT is essentially organic matter AOT. We also highlight that mid-tropospheric circulation was characterised by: 1) on 2 Sep, a low pressure system localised off-shore of Henties Bay, juxtaposed to a high pressure system localised over South Africa, resulting in a small river of smoke descending along the coast (see figure); 2) on 7-8 Sep, an elongated high pressure dominating over the continent, resulting in a channelling of the smoke from north-west. Therefore, we point out that, given the features of the smoke transport over the Henties Bay region, even small differences in the simulation of the weather conditions could lead to substantial differences in AOT for specific locations, especially when AOT values are rather low. These different aspects have been added in Section 3.1.

Page 9, Line 265: Why do you refer to the observations as biased with respect to CAMS? Couldn't the CAMS value be off? It might be helpful to explain why either estimate may be different than "truth."

Thank you for pointing this out. The sentence is misleading in this form, and it has been rephrased. We also expand the discussion on the discrepancies between CAMS and observations, so that now the paragraph reads:

" These discrepancies may be also explained by the coarse spatio-temporal sampling of the model, which is insufficient to highlight the sharp variation in AOT due to a very localized channelization during these 3 days. Note that no significant precipitation event was recorded during the field campaign, so that we can exclude any CAMS misrepresentation of wet deposition processes. Otherwise, CAMS simulations show that the AOT is essentially due to organic matter (i.e. biomass burning aerosols), the contribution from non-biomass aerosol can then be excluded as well. Hence, given the features of the smoke transport over the Henties Bay region, even small differences in the simulation of the weather conditions could lead to

substantial differences in AOT for specific locations, especially when AOT values are rather low.  On 2 September a minimum in AOT is observed by the sun photometer which is not reproduced by CAMS simulations (even though a local minimum in the CAMS AOT can be seen). During this day, mid-tropospheric circulation was characterised by a low-pressure system localised offshore Henties Bay, juxtaposed to a high-pressure system localised over South Africa, resulting in a small river of smoke descending along the coast. On 7-8 September, the sun photometer- and MODIS-derived AOTs are larger than the one computed from CAMS. This could be related to the presence of unscreened optically thin clouds such as the ones observed in the ground-based lidar data on 8 September (Figure A2d) and/or to the heterogeneity of the meteorological field. Indeed, on 7-8 September, an elongated high pressure dominating over the continent, resulting in a channelling of the smoke from north-west.".

Page 9, Table 3: It would be helpful to explain what the uncertainty range is in the caption. Also, for the profiles encompassing a long period of time, how much of the uncertainty in average value comes from remote sensing uncertainties versus real variability over the time period?

This discussion has been added in the Appendix A. The uncertainty includes both the detection noise and the natural atmospheric variability.

Page 10, Table 3: How are these profiles divided? For instance, why are the 22/08 profiles divided at 1607?

As explained in the text, we average the profiles on the time laps without cloud cover. Hence, they are not systematically located at the same times. We try to keep as much information as possible for each day.

Pages 12-13, Figures 3-4: I don't understand why the figures are divided in this manner. Also, not all potential profiles from Table 3 appear to be plotted (e.g., it seems like there is GBL data for 22/08 from 1608-2400 that is not included). It might make more sense to divide this into a number of separate figures for each period.

Following the reviewer's remark, we have divided the figures by period in order to improve visibility (see Figure 3-7). We have now included all mean available profiles available from Henties Bay.

Page 14, Figure 5: The color scale here makes the figure very difficult to read. Perhaps using one that ranges from a very light color to a darker color for high AOT would both be more intuitive but also make it easier to read the wind markers.

The correction has been done.

Page 14, Line 294: What do you mean by "evolution"? That term may be misleading because you are not really showing changes over time but rather a time-space cross section following the aircraft.

Agreed. We have changed the sentence to "**Erreur ! Source du renvoi introuvable.**a shows the time-space cross section of the LNG-derived apparent aerosol backscatter coefficient (ABC) profiles at 532 nm along the Falcon 20 flight track in the morning…"

Page 15, Lines 307-309: I cannot find evidence for this statement in the Haywood et al. (2003) paper cited. Parmar et al. (2008) does establish that water vapor is emitted at fire sources along with carbonaceous species but does not claim that this could humidity an entire well-mixed continental boundary layer. Why would the high relative humidity not simply be characteristic of continental air whereas low humidity air be from subsiding tropical or midlatitude air that has been depleted of moisture via prior precipitation?

Agree, the reference to Haywood et al. (2003) has been removed.

We also agree that the RH values could be related to the characteristics of continental air masses. Nevertheless, large correlations between RH and aerosol loads between 850 and 700 hPa have been observed in this area and for the same time period (e.g. Daeconu et al., 2019). Nevertheless, we have softened our statement so to include the suggestion of the referee.

We have added 2 references:

Deaconu, L. T., Ferlay, N., Waquet, F., Peers, F., Thieuleux, F., and Goloub, P., 2019: Satellite inference of water vapor and aerosol-above-cloud combined effect on radiative budget and cloud top processes in the Southeast Atlantic Ocean, Atmos. Chem. Phys., accepted.

Clements, C. B., Potter, B. E. and Zhong, S.: In situ measurements of water vapor, heat, and $CO_2$ fluxes within a prescribed grass fire, Int. J. Wildl. Fire, 15(3), 299–306, doi:10.1071/WF05101, 2006

Page 16, Line 342: In what way does elevated RH suggest the aerosol layer must be

distinct?

A high variability of RH in the atmosphere generally means that we have different pathways of air masses. When combined with signatures on the vertical aerosol profile, it is a favourable element for the presence of different layers against altitude. However, back trajectories are required to confirm this. We agree that this is a bit of an overstatement at this stage. We have deleted the end of the sentence, i.e. "[…] which together with the AEC profile in Henties Bay, suggests the presence of a distinct aerosol layer above the main BBA layer, that is not seen in the AEC profile '1' (Figure 8b)".

Page 16, Line 362: In what way does the existence of non-negligible AEC values

suggest that the aerosol has a different origin than the other aerosols observed?

Agreed. At this stage, there is not enough information for this to be concluded. Mention to the "different origin" has been removed.

Page 18, Figure 9: It would be helpful to make clear which profile corresponds to

which dropsonde in Figure 8.

The correction has been done.

Page 19, Line 395: Do you mean "back-trajectory" instead of "retro-trajectory"?

Agreed, it is a mistake. The correction has been done.

Page 20, Line 447: What do you mean by "nebulosity"?

The nebulosity corresponds to the cloud cover or cloudiness. The correction has been done.

Page 22, Figure 12: I would suggest using whichever color scale you choose for Figure 5 here as well.

The colour palette has been changed.

Page 23, Lines 482-483: Couldn't high observed AOT also be due to aerosol humidification at high RH, versus thin clouds?

If there were wet aerosols, the lidar would have detected them as well. We think that they are more clouds that are not properly identified by passive detection.

Page 23, Lines 497-498: What do you mean by "trapped in the FT"?

We have changed the sentence: "…in relationship with the main transport regimes across the Atlantic Ocean".

Page 23, last paragraph in its entirety: This is very interesting, but not appropriate to introduce and discuss only in the conclusion. This could potentially be a great addition to the paper as a separate section with some new analysis of the meteorological fields to provide evidence for the claims made.

Agreed, this paragraph has been moved at the end of Section 5.

---

## Author Comment (AC2) · 17 Sep 2019

**Reviewer #2**

The argument that the South American aerosol layer is "distinct" is unconvincing and the conclusion that it is of significant importance for climate modeling is premature. Yet, aerosol transported from South America to Africa is still interesting enough to highlight, even without this. Better to be very straightforward about what is and is not known about it, to inform and motivate future study.

Agreed, we have tamed down some of our previous conclusions to account for this. For instance, the back trajectories analysis has been revised for the sake of clarity and we have also added Figure 2b to highlight the likely contribution of the South American aerosol layer.

Additionally, some lack of clarity makes it difficult to understand the details of the lidar retrieval; in particular the derivation of the lidar ratio, and the associated uncertainties. The lidar retrieval is subject to large uncertainties because of the underdetermined nature of the problem, but there is a lack of useful information about the size of the uncertainties or their impact on the conclusions.

The processing of the mean lidar profiles has been clarified in Appendix A. The main uncertainty sources have been presented and assessed: the statistical noise due to the shot noise and the atmospheric variability during the average time, and the uncertainty linked with the lidar ratio.

119 "between about 1 and 4 hours". Actually, several of these averages are 7 or 8 hours long, so this should be indicated correctly. Even 4 hours seems like quite a long time, considering the variability of aerosol in the region. Was there any need for filtering or any analysis to check for stationarity?

There is a low signal-to-noise ratio for individual profiles, especially during the day. For this reason, but also because the vertical structure of the atmosphere does not vary significantly, we have performed some averages over several hours. These averages were very constrained by the presence of clouds as explained in the text. The temporal variability of lidar profiles is now illustrated in Figure A2.

Section 2.3 The Bruneau et al. 2015 paper referenced at 135 is missing from the bibliography. I think I know which paper you mean, though, and it says this instrument has HSRL capability at 355 nm. If so, why does it require the retrieval procedure that's used for the ground-based elastic backscatter lidar? Even if I'm wrong about the HSRL, the appendix says that the lidar has a 355 nm channel. So then why is the 355 nm data not shown in figures 6b and 8b? (if it's available, it would be very helpful there!)

Yes, LNG indeed has HSR capability at 355 nm. Nevertheless, this channel did not work during the campaign and the 355 nm channel was very noisy and unusable. We use only the 532 channel. We have added the sentence in Section 2.3: "We only use the 532 nm channel because the high level of noise of the 355 nm channel". Hence, the lidar operated has a simple backscatter Rayleigh-Mie lidar. The reference to Bruneau et al. (2015) has been added.

Section 3. I found the organization of section 3 hard to follow. It goes back and forth multiple times between different subjects. Please consider reorganizing to make one paragraph about the overall AOT trend, one comparing P1 to P2, and another separate paragraph comparing P2 to P3.

We agree that this section is difficult to read because the elements are not grouped together. We have reviewed its structure and divided it into two parts for clarity: "3.1 Identification of periods from the total AOT" and "3.2 Aerosol vertical profiles". We have removed the MODIS figure that did not belong in this paragraph. However, we have added Figure 2b, which allows us to better monitor the temporal evolution of AOTs at 355 nm and better highlight the correspondences with AOTs at 550 nm.

215-218 statement that lidar ratios in the free troposphere "suggest the presence of terrigenous aerosols mixed with smoke. This is coherent with the polluted dust type inferred from the CALIOP observations." Is this circular reasoning? The Appendix suggests that only the lidar ratio in the PBL is retrieved and that the lidar ratio in the free troposphere is taken directly from the inferred lidar ratio in CALIPSO or CATS.

Agree, the sentence is poorly formulated and has been removed.

The presence of dust in the smoke layer isn't very consistent with the AEROCLO-Sa ground-based lidar measurements in Figure 3, which shows quite small values of the particle depolarization ratio within the smoke plume for all profiles.

It's not so small. In the presence of pollution aerosols, the PDR is rather between 1 and 2% because they are generally hydrophilic. For biomass burning aerosols, they are often mixed with dust that may have been raised by pyro-convection process. Values of ~ 5 to 10% are therefore likely for aerosol mixtures of biomass burning and dust aerosols.

Even if I'm misunderstanding how the constrained retrieval works on the ground based lidar, I believe it's likely that the retrieved lidar ratio in the free troposphere must have a large degree of uncertainty which probably precludes making a distinction between smoke and polluted dust, whereas the uncertainty on the particle depolarization ratio is likely to be quite reasonable. Does this conform to your analysis of systematic errors and uncertainties?

We have had a discussion on this uncertainty in the Appendix and now calculated the error due to the LR on the aerosol extinction coefficient (and AOT) and the particle depolarisation ratio (PDR). In addition, statistical noises have been added to the lidar profiles in Figures 3 to 7.

220 "being remobilized by pyroconvection and mixed with BBA before being transported aloft". How do you know these details about the origin of the dust? What evidence is there of pyroconvection?

We observe well mixed layers whose PDR remains substantially constant and above what is expected from pure biomass burning aerosols. One explanation is the simultaneous injection of dust during intense fires. This is only a hypothesis at his stage.

217 What does the citation Flamant et al. 1998 refer to here? Is this a typo? (Also, the paper is missing from the bibliography).

This refer to the LR value for marine aerosols. It is now made clearer in the text. The reference has been added.

232 "LR values in the FT evolving from 55 to 70 sr". I believe the lidar ratio in the free troposphere is assigned based on aerosol type that is somehow inferred from the CALIPSO and CATS aerosol type inferences. It would be better, then, to directly refer to a change in inferred aerosol type rather than an evolution of the lidar ratio, which implies more quantitative detail than is actually available.

Agreed, as explained in the text the LRs are derived from the CALIPSO and CATS aerosol typing. The aerosol typing is clearly identified in the text with the LR.

238 "LR values observed in the PBL". Please be specific about which retrieval you mean. The combined lidar-sunphotometer retrieval or the sunphotometer only retrieval from the previous sentence?

This is now better specified in the text. We have added the aerosol typing in brackets after the LR value. The aerosol typing is also given further in the text.

239 Again the presence of dust would be expected to be accompanied by an elevated particle depolarization ratio which is not evident.

As we explain, it is not pure dust particles, but a mix with other types of aerosols. The LR is then weighted against the ration between dust and other particles.

245-247 The relatively high particle depolarization ratio in the altitude region between the smoke plume and the PBL seems unlikely to be significant compared to its uncertainty. The systematic error of particle depolarization ratio increases dramatically for low backscatter ratios (see, e.g. Freudenthaler et al. 2016, Burton et al. 2015). If you have done a calculation of the expected systematic uncertainty and believe these to be significant, then that should be included to support your interpretation.

The uncertainties have been calculated and added on the profiles. As expected, they are generally more important when the AEC is low, but this is also a function of the signal to noise ratio. We reviewed the discussions in the Appendix and in the Section 3.2, which were indeed not enough clear.

Table 3 What is the number after the plus-and-minus sign?

This is the statistical noise due to the detection and the natural variability of the atmosphere. This is now indicated.

Table 3 Why is the lidar ratio consistently only a few different values? In the PBL, the retrieval is described as a constrained retrieval, so I expected it to be able to take a continuous range of values.

Our choice was to remain within the range of values proposed for CALIOP and CATS and these values allow us to find with enough precision the AOTs derived from passive instruments. With the uncertainties about the LR, little more can be expected.

Figure 3 Add error bars? I expect significant amount of variability due to the very long averaging and also a significant amount of systematic uncertainty. Even if only a coarse estimate of systematic uncertainty can be included, this would be valuable for interpreting the significance of features in the profiles.

The error bars have been added in Figures 3-7. They show that our discussion is appropriate.

316ff. Somewhere in this paragraph please state the distance between the profiles.

The distance between the 2 profiles (~100 km) is now mentioned in the text.

318 "operate at different wavelengths". Table A1 indicates that LNG also operates at 355 nm. Can you show 355 nm retrievals from LNG as well? It might help establish that there really is correspondence between the airborne and ground-based retrieved extinction and give better support that the differences are not due to the wavelength.

The LNG detection did not work properly at 355 nm during AEROCLO-sA. Hence, we have not used the data. Nevertheless, we agree that this is not mentioned in the manuscript, Table A1 just providing an overview of the LNG system. In the revised version of the manuscript, we are now mentioning that the LNG detection at 355 nm was not working nominally (Section 2.3) and hence the data not included in the analysis.

324 "the difference can be explained". There are multiple factors affecting this difference, including the approximately four hour time difference between the profiles, which has not been addressed. Any comment about the role of the time difference? In any case, soften the statmeent. Maybe "More important is the regional scale circulation"

We have added this information together with that on the wavelength difference, in the form: "there is a 4-hour difference between the aircraft profiles and the mean profile over Henties Bay". We also have softened the statement by using "may' instead of "can".

328-329. Does the CAMS model also show this difference between the aerosols over the land and ocean? You previously made an apparently conflicting point that CAMS indicates homogeneity in the aerosol around Henties Bay.

No, the CAMS model does not show this difference in the structure of the aerosol profiles between the land and the coastline.

335-336 "the structure of [profile 2] being coherent with the ground-based AEC profile". Not convincingly so. Also, this statement is apparently contradicted at line 347. Is this a typo?

Thanks for picking this up. Indeed, this is a typo. The sentence has been corrected.

340-341 I'm confused by the indiation of 20% RH above the BBA layer. First, is this referring to 9b or 9a? Either way, I see that it doesn't drop dramatically to zero like 8b, but I do not see 20% RH for any part of the profile above the strongest gradient (top of the BBA) and even the highest value I do see (more like 10%?) is only for a few hundred meters.

Agreed. This is a mistake, the RH above the BBA does not reach 20%, and indeed more like 10% (at most) for a few hundred meters. We are referring to Figure 9b, i.e. the profiles from the dropsonde released closest to Henties Bay. The sentence now reads:

"The maximum RH in the FT is ~55% and observed near the top of the BBA layer (Figure 11b), while small RH values (less than 10%) are seen above ~6 km AMSL It is worth noting the presence of a slightly enhanced RH layer between 5.5 and 6 km AMSL.".

341-342 "suggests the presence of a distinct aerosol layer". Is the key point that there is aerosol present? In which case, I think the lidar profile itself (assuming the signature of the aerosol is significant above the retrieval uncertainty and not just an artifact of the retrieval) is evidence of aerosol and the RH is mostly irrelevant. Or is the key here that the aerosol is "distinct". I don't understand how either the extinction profile or the relative humidity indicate that the aerosol is physically distinct from the African BBA layer.

Agreed. This is a bit of an overstatement. We have deleted the end of the sentence, i.e. "[…] which together with the AEC profile in Henties Bay, suggests the presence of a distinct aerosol layer above the main BBA layer, that is not seen in the AEC profile '1' (Figure 8b)".

344 I don't see that there is much agreement in the "structure" of the PBL. You might just say more simply that the apparent height of the PBL in the aerosol agrees with the location of the gradient in RH.

Agreed. The sentence has been modified accordingly.

353-354 Not clear what "distinct aerosol layer above the main BBA layer" means. Please specify the altitudes you're talking about. To me it looks like the main BBA layer extends to higher altitude in the Henties Bay uplooking lidar profile and in profile (1), while (2) shows a lower top for that layer. I don't see it as a separate, secondary layer. I can easily agree if you say "additional aerosol above 5 km" which is more unambiguous.

Agreed. This part was removed to avoid repetition as the information about the additional aerosols above 5 km has been added earlier in the section where the features in the upper part of the BBA layer are discussed with respect to the profile in Henties Bay.

360 "related to ... hygroscopic growth ... particularly below the BBA layer where RH is high in '2'". Again, I'm having some trouble following the roundabout wording here, but I take "below the BBA layer where RH is high" to mean "below 1 km". Yet there is no enhancement of extinction in profile '2' below 1 km. So, either this statement is incorrect, or the wording is so confusing that the meaning has been lost for at least one reader.

Agreed. This part is rather unclear and has been removed.

361 How robust are the estimates of the AOT of the layer above 5 km to the uncertainties in the lidar retrieval?

The temporal evolution of the AOT between 5 and 6 km AMSL is now plotted in Figure 2b with the error bars. We have also added an assessment of the uncertainty due to the knowledge of the lidar ratio in Appendix A.

362 "suggest that the aerosol may have a different origin". Again I do not understand what evidence supports this statement.

Agreed. Mention to the "different origin" has been removed.

392-393 "RH values ... may be an indication of the transport of BBA from a different origin." Again, I don't understand why the RH values should be taken as evidence of a different origin. I believe you have stronger evidence in the back-trajectories. It would be better not to state this conclusion until it has actually been supported. At most, for this sentence, I would say something like that the difference in RH is an indicator that the meterology has changed (if indeed that is what you believe) and that that will be shown in the next section.

We agree and we have modified the sentence: "…which is an indication that the meteorology has changed and by this way that the origin of air masses may be different.".

416 "are transported very rapidly" I don't understand how to interpret the figure to conclude that the aerosols are transported rapidly. Since the trajectories are more tightly distributed around the point of origin here than in the other panels, and yet all 3 panels show back-trajectories for the same 6-day duration, I infer that in this panel they moved more slowly.

Yes, we have removed "very rapidly".

423 "They correspond to air masses arriving above 5000 m AMSL over Henties Bay". Please show this by splitting 11(c) into panels above and below 5000 m. Since you treat this point as important, it should be shown explicitly.

We have now added the back trajectories for the 3 altitude intervals and each period: [1500 3000[, [3000 5000[ and [5000 6000[ to better highlight the differences.

Figure 11 caption. What does "normalized occurrence" mean (normalized to what)? Does this imply that the scale does not translate to the same number of back-trajectory points in each of the three panels?

The back trajectories have been calculated with a vertical step of 250 m. Hence, the number of back trajectories is different against the altitude intervals but stays relevant (> 500 back trajectories). The normalization is performed with respect to the total number of pixels for a horizontal resolution of 0.5-1° (this information has been added in the figure captions). The use of normalized variables is that they are comparable provided that the number of samples is large enough, which is the case here.

502 "of paramount importance" should be toned down. "Highlighting" the transported aerosol is interesting and worthwhile but not of "paramount importance" until it is shown to be climatically significant and not already captured in climate models.

Agree, we have reviewed the sentence: "Highlighting the transport of BBA from South America and its likely advection on top of the BBA layers originating from Angola and northeast Namibia may be climatically significant but not already captured un climate models in this region of the globe, where the feedback of aerosols and clouds on the radiative balance of the Earth system is still poorly known.".

750 Define "apparent backscatter coefficient", preferably using an equation. Not knowing for sure what this is is making it hard for me to follow the rest of the section.

This is like the total attenuated backscatter coefficient and the equation is always given in Royer et al. (2010). We have added both the definition and reference.

751-752 "must follow the slope of the molecular bacskcattering". This is not well explained. Won't there be attenuation of the signal by air molecules and aerosol at lower levels? (wheras the molecular backscattering from ERA5 is not affected)?

Agree, there is obviously an attenuation, but the molecular slope is not affected when there are no aerosols. It is therefore a strong element, often used by instrument designers, to check the alignment of lidar far-field. This is what we use regularly to check the alignments of the lidars that we develop. The use of Era5 is interesting because there is little error on the temperature and pressure profiles with these reanalyses which assimilate the IASI radiances. These profiles are not affected by atmospheric transmission.

759-760 "the error remains below 2-3%" Which error remains below 2-3%? The error in the molecular backscatter, or the aerosol backscatter derived from it, or the aerosol extinction derived from it?

This is only the error dur to the variability of the molecular density. This point has been better explained.

779 "uncertainty sources are exhaustively quantified". Given that you have access to a methodology for quantifying the uncertainty, it should definitely be quantified for the data presented in this paper. Depending on how big the retrieval uncertainites are, some of your conclusions can be affected, as discussed elsewhere in this review, so it's not just an academic exercise.

The uncertainties due to the statistical noise have been added on each vertical profile in the main part of the paper. The bias linked to the LR have been computed and illustrated by two representative cases in the Appendix.

781-782 "using aerosol typing determined from the CALIOP and CATS measure ments". How to you obtain the free-troposphere aerosol type and lidar ratio for cases where there is no satellite overpass in table 1? How do you get them for cases where CALIPSO or CATS or both infer multiple types in different pixels in the region (like the 31 August case, for instance).

When there is no CALIOP or CATS overpasses, we take the LR values of the nearest day. We have added the sentence: When there is no CALIOP or CATS overpasses we take the value of LR of the nearest day also considering the shape of the AEC profile and the origin of air masses using back trajectories. Note that the CALIOP and CATS typing is very much in agreement.

795-798 It seems like a stretch to infer from just a few individual cases of transported smoke from a totally different source as presented by Muller et al. 2007 that the lidar ratios for dust and smoke at 355 nm are the same as 532 nm. Is this the only relevant paper?

To our knowledge, this is the only one article that uses a multi-wavelength lidar over a long enough period to deduce statistically convincing results.

800 "matches best the AOT from the sun photometer". At the lidar wavelength?

Yes, we have specified the wavelength.

812 "uncertainties of 2% on the PDR". Do these uncertainties include sources of systematic error? Doesn't the uncertainty level depend strongly on the amount of aerosol? So, is there a minimum aerosol amount to get the quoted 2% error?

Yes, the uncertainty on the PDR depends on the value of the AEC. We have reviewed this part by taking two representative examples in the Appendix.

*Minor comments*

52 "as also mixes". Should this be "and also mixes"?

The correction has been done.

91 says the ground-based lidar is an ALS 300, but in the appendix it says ALS 450. Is there a difference between these two designations?

No, they are very similar. The correction has been done.

93 please indicate what type of lidar the LNG lidar is.

The LNG usage mode has been specified.

98 UTC is "coordinated universal time"

It is the Universal Time Count as explained in the text. It is like the definition of the reviewer.

154-155 a better reference for CALIOP 4.10 typing would be Kim et al. 2018

The reference has been added.

175-176 "The standard deviation on the AOT" should be "the uncertainty in the AOT".

The correction has been done.

205 change word order "when aerosols are only observed"

The correction has been done.

209 refers to lines highlighted in green in the table. I don't see highlighting. Does this need to be reworded?

It is a mistake, the correction has been done.

210 "averaged AOT of ~0.15" Indicate which instrument or model this refers to.

This referred to Figure 2, the correction has been done.

213 reference to Angstrom exponent. Which instrument or model?

This is from the sunphotometer, the information has been added.

227-228 "between the 2 periods". Between P1 and P2?

Yes, the correction has been done.

234 add reference for sun photometer retrievals of lidar ratio

The reference to Dubovik et al. (2000) has been added.

250 "match perfectly" is an overstatement. Please reword. It's better to be quantitative anyway.

The correction has been done.

256 "positively biased" reword to avoid "biased" and use a more neutral phrase like "larger than". You've given reasons why either the model or the observations (or both) could be incorrect in this case.

Right, the correction has been done: "the sun photometer- and MODIS-derived AOTs are larger than the one computed from CAMS".

Table 3 caption, UAL is used here where elsewhere it is described by the label FT. Is the difference signficant? If not, please pick just one.

Yes, it is different. The free troposphere is not only the aerosol layer. Nevertheless to avoid confusion we now used AFT for Aerosol layer within the free troposphere.

Table 3 column "AE" please indicate which instrument this is from.

The information is already given in the table caption.

Figure 2 It would be easier to understand the x-axis if tick marks were an integer number

of days instead of 1.5 days.

Yes, the correction has been done. We also have added the temporal evolution of the AOT for each altitude interval in the free troposphere.

292 "where" should be "were"

The correction has been done.

294 I believe this is probably attenuated backscatter coefficient rather than aerosol backscatter coefficient. This is an important distinction.

Yes, the correction has been done.

308 remove "wood". I think it's more likely grasses and crops here than wood, but either way it's speculation. "Combustion" is sufficient.

Ok, the correction has been done.

323 "apparition" should be "appearance"

The correction has been done.

334 "offshore airborne lidar measurements from Henties Bay". Replace with "the profile west of Henties Bay" for clarity.

The correction has been done.

351 "below the base of the BBA layer observed further south". Please simply state the altitude for easier reading.

The altitude has been added.

352 "The RH above the top of the BBA layer". Again, it would be helpful to be more specific about what altitude and which RH profile you are referring to, for easier reading. Figure 6

caption "apparent bacskcatter coefficient". Is this the same as "attenuated backscatter coefficient"? Please add a reference for where this quantity is defined.

Yes, we have rephrased this part in a sake of clarity: "Figure 9, the RH sharply increases close to the BBA layer, which together with the AEC profile in Henties Bay suggests the presence of a distinct aerosol layer above 4.5-5 km AMSL, that is not seen Figure 8b in the AEC profile '1', but seen over Henties Bay."

386 add "at Henties Bay" for clarity

We added it.

408 "Illustrating" not "corroborating" (this isn't independent evidence)

The correction has been done.

412 "between the 3 regimes" perhaps rather than "periods" since the different regimes include different choice of ending altitude as well as choice of ending date.

"Period" is better adapted because the transport regimes are very variable in time and what explains the differences on the lidar profiles involves the transport regime but also the origin of the air masses. We have revised our explanation in considering the evolution of the back trajectories.

412 similarly "For the layers observed during P1" for the start of the next sentence, since you are only showing back-trajectories corresponding to observed aerosol layers.

The same answer as the previous one.

414 "consistent" not "coherent"

The correction has been done.

484 "illustrate" or "show", not "evidence"

The correction has been done.

497 probably delete "trapped" or reword to make the meaning clearer

"trapped" has been deleted.

498 I don't understand the meaning of "in link with the dominant transport patterns of lifted aerosols". Please reword.

We have reworded the sentence: "in relation to the main transport regimes across the Atlantic Ocean".

504 replace "fundamental" with "necessary". Fundamental has a connotation of primary importance in addition to meaning "necessary" and therefore sounds like an overstatement.

The correction has been done.

Table A1 add post-processing temporal resolution

The information has been added.

749 Replace section title with "Overlap correction" or something similar, since this is only one of many aspects of calibration but the only one discussed.

Yes, the main correction is about the lidar overlap and we have added "Overlap correction" in the title of the section: "Overlap correction and rightness of lidar profiles".

761 "average lidar profiles".

The correction has been done.

766 replace "perfect coincidence" with "very good agreement"

The correction has been done.

777 replace "exogeneous" with "external"

The correction has been done.

798 replace "standard deviation of LRs derived from spaceborne lidars" with "expected uncertainty in LRs for spaceborne lidars". Lidar ratio is not dervied from CALIOP and CATS; they do not measure or retrieve lidar ratio. It's an input, not an output, of the retrieval.

The correction has been done.

802 replace "coherent" with "consistent"

The correction has been done.

807 replace "inverse" with "invert"

The correction has been done.

---

## Referee Report (RR1)

**Review of revision of "Evidence of the complexity of aerosol transport in the lower troposphere on the Namibian coast during AEROCLO-sA" by Patrick Chazette et al.**

**Recommendation:** This manuscript still requires major revisions, particularly in the analysis of the HYSPLIT trajectories. Importantly, the evidence presented that smoke from South America could play a major role in the region remains unconvincing.

**Major issues:**

The major issue with the manuscript remains the presentation and interpretation of the HYSPLIT trajectories and circulation patterns more generally.

Showing vertical ranges for all three periods together is a major improvement but does not obviously support the interpretations the authors prefer. In particular, looking at the 5000-6000 m range, it is not obvious that South American contributions should be expected in P3 but not in P2 or even P1.

One potential problem is that many dates and vertical levels are convolved in the presentation in addition to the ensemble "uncertainty" around each trajectory. Thus, it is difficult to know if certain regions of the cloud of probability are very certain to have been the source at certain time periods but not others or whether the meteorology was fairly steady but there is great uncertainty about the path taken.

In addition, it is impossible to tell the vertical level of the trajectories through time in this presentation, which could be particularly important for the South America assertions. Running some ensemble back trajectories on the online HYSPLIT portal (see below), I see that a few ensemble members dip down to ~4 km over South America before reaching Henties Bay on September 7th but the majority of ensemble members remain above 7 km. It would be surprising if the South American smoke were lofted that high.

The authors do not adequately explain how they are reaching certain conclusions from the evidence presented. For example, an assertion is made that during P1 trajectories from the 3000-5000 m range show more air masses coming from the southern Atlantic Ocean, and indeed South America, as compared to the other vertical ranges, but this is does not appear to be supported by Figure 13a-c. Similarly, the direct link between the circulation and aerosol transport in Figure 14 is either not obvious or directly contradicted by the figure (in that the coherent area of high AOTs seeming to come from South America stays far south of Henties Bay).

One potential remedy would be to restructure the transport section as separate case studies of emblematic days during the three periods. This should simplify things down enough to show more clearly whether the source is particularly uncertain or if there's just variation in source within the periods and, importantly, the vertically-resolved and

time-resolved trajectory data that would aid in interpretation. This could be provided in addition to or in lieu of Figure 13 now, with a short, added discussion of how the days not highlighted compare. I'm open to other possibilities as well, but feel that the analysis as-is does not merit publication.

A final major issue has to do with figure quality. Many of the scales are all but illegible based on their small size and low resolution (see, e.g., the color bar in Figure 13 and the wind barbs in Figure 14). One or preferably both issues should be addressed prior to publication.

**Specific comments:**

1. Page 2, Line 54: Rather than claiming that stratocumulus are the most effective at reflecting sunlight (deep convective clouds have higher albedo but smaller net radiative effect due to compensating longwave heating), it would be more accurate to say something along the lines of: "marine stratocumulus are particularly sensitive to aerosol perturbations due to relatively low background aerosol concentrations (Oreopoulos and Platnick, 2008)"

2. Page 8, Lines 180-181: It makes sense that the AOD retrievals over the ocean surface will be more certain than those over land, but this still doesn't address any issues relating to the lack of co-location. Unless the sea breeze is acting uniformly from the surface to ~5 km, it is plausible that the AOD over the ocean may differ somewhat from that at Henties Bay. I would think the effect is small, but it may be worth mentioning as a source of uncertainty regardless.

3. Page 8, lines 213-215: Although no precipitation was observed at Henties Bay (which is unsurprising), it is possible that wet scavenging could have occurred closer to the source of the emissions, depleting the aerosol plume before that air was transported to the Henties Bay area.

4. Page 10, lines 313-314: I'm not sure that "pyro-convection" accurately describes the strength of the primarily anthropogenic plumes in the region. Also, is there significant burning in the Etosha Pan itself?

5. Page 22, line 444: Which altitudes? You're referring to 3000-5000 m here, right? As written, it sounds like you're referring to the full 1500-6000 m column.

6. Page 22, line 450: It is very hard to tell from the figures that the trajectories are turning counterclockwise. Perhaps some kind of composite trajectory would be useful? I'm thinking of the analysis in Adebiyi & Zuidema (2016), Figure 17, as inspiration here. Then you could address the altitude of the trajectories as well, which is not possible to do in the current format and could be important.

7. Page 22, lines 454-455: I don't see how you're concluding that the 3000-5000 m level is "mainly" influenced by air from over the southeast Atlantic as compared to the other two vertical ranges.

8. Page 22, lines 455-456: Some of the ensemble members show the starting location as southern Brazil, but others appear to disagree. It's plausible that the origin was around Brazil, but the trajectory analysis doesn't show that Brazil *was* the origin.

Especially when you've gotten out to 6 days, the HYSPLIT trajectories need to be taken with a generous helping of salt.

9. Page 22, line 456: The aerosol plume is located between 1500-3000 m during P1 according to all the other plots... why are you saying there is no aerosol in that range now? There certainly does not appear to be a plume between 3-5 km during P1, which would be the implication of this section...

10. Page 23, line 462: Most of the fires are anthropogenic and set for agricultural purposes, not "wildfire." "Biomass burning regions" may be better phrasing.

11. Page 23, lines 464-646: For a given start time, the HYSPLIT ensemble provides an estimate of meteorological uncertainty, and while it may be helpful to think of some mixing of airmasses on the way to Henties Bay, a more straightforward interpretation of the ensemble may be that more oceanic and more land-based transport pathways are both plausible. It's problematic to assume that all the trajectory ensemble paths were actually followed, however. This analysis is more appropriate for a plume dispersion analysis, which can be done with HYSPLIT or with another program like FLEXPART. However, if in analyzing individual days within the period you see some with different sources, and that is the source of the wide range in the cloud of probability, that is worth reporting. If true, however, this casts doubt upon the ability to tell a coherent story about three discrete periods, as the paper currently attempts.

12. Page 23, line 473-474: It would be really helpful to somehow indicate times on a figure. Again, see the note about a composite trajectory above. You could even group ensemble members with similar paths together to make things clearer on the map.

13. Page 23, line 476: Certainly most trajectories aren't coming from the south below 5000 m. Also, it's not clear that the amount of trajectories coming from South America differs substantially between P2 and P3, with P2 perhaps seeing even more from South America.

14. Page 23, line 478: The analysis presented does not establish that the highest AOTs on these days are associated with biomass burning from South America.

15. Page 23, line 484: Why is the cloud cover "important"?

16. Page 23, lines 496-497: What is the evidence that the temporal variability of South American transport "appears" linked to the SAM? Are you simply saying this is a plausible explanation because the SAM is generally important, or is the analysis above evidence in support?

17. Page 24, lines 521-523: The relationship between transport patterns and the SAM is plausible, and appears to be very easily testable given datasets like ERA already discussed by the authors. The authors should test this claim (that different SAM phases correspond to their periods P1-P3) if they want to report it. If the authors are unable or unwilling to provide further support for their linkages between aerosol transport and the SAM, it would be best not to include this section at all or merely mention it as an avenue for future research.

18. Page 25, Figure 13: It would be immensely helpful to the reader to have better labeling here, perhaps for both altitude and period (labeling columns/rows would be

fine). It could also be helpful to include an indication of whether smoke was present at the given altitude for each period.

19. Page 26, Figure 14: I don't see how the AOT in this figure supports your conclusions. If anything, it appears to show the South American-linked AOT stays south of 30 S.

20. Page 26, lines 548-549: It doesn't really make sense to say that this is the first time biomass burning aerosols were characterized by lidar at Henties "during the different periods of transport" — it's the first time, period, that (ground) lidar-based characterization was possible. That there were three transport periods is a separate idea (and the division into three periods is an interpretation of the data, not a direct observation).

21. Page 27, line 570: No evidence is presented that the transport regimes the authors associate with periods P1-P3 are the "main transport regimes across the Atlantic Ocean."

22. Page 27, line 577: I still think it is overstating the case to consider a 10-15% contribution to column loading seen on two days out of three weeks of observation as "necessary" for realistic simulation of the region. Or is this meant to refer to better constraints on the aerosol column more generally? In context, it appears to refer specifically to the South America-related results.

**References:**

Adebiyi, A. A., & Zuidema, P. (2016). The role of the southern African easterly jet in modifying the southeast Atlantic aerosol and cloud environments. *Quarterly Journal of the Royal Meteorological Society, 142*, 1574-1589.

**Figure:**

---

## Author Response (AR2)

Recommendation: This manuscript still requires major revisions, particularly in the analysis of the HYSPLIT trajectories. Importantly, the evidence presented that smoke from South America could play a major role in the region remains unconvincing.

**We have strengthened our arguments and reviewed the presentation of the back trajectories (see below).**

Major issues:

The major issue with the manuscript remains the presentation and interpretation of the HYSPLIT trajectories and circulation patterns more generally.

**We have taken into account the reviewer's concerns and clarified the discussion where needed.**

Showing vertical ranges for all three periods together is a major improvement but does not obviously support the interpretations the authors prefer. In particular, looking at the 5000-6000 m range, it is not obvious that South American contributions should be expected in P3 but not in P2 or even P1.

**The new presentation should be clearer. We are considering the possibility of transport, but with more caution, as rightly so the reviewer had initially requested.**

One potential problem is that many dates and vertical levels are convolved in the presentation in addition to the ensemble "uncertainty" around each trajectory. Thus, it is difficult to know if certain regions of the cloud of probability are very certain to have been the source at certain time periods but not others or whether the meteorology was fairly steady but there is great uncertainty about the path taken.

**Indeed, the meteorology is fast evolving in the region of interest in relationship with the anticyclone belt over the Atlantic and the modulation of the westerlies poleward of the it, as well as the numerous eastward travelling disturbances. Hence, isentropic back trajectories must be used with caution, but it is still informative. We opted for a statistical study rather than simple back trajectories to take into account the uncertainties associated with transport.**

In addition, it is impossible to tell the vertical level of the trajectories through time in this presentation, which could be particularly important for the South America assertions.

Running some ensemble back trajectories on the online HYSPLIT portal (see below), I see that a few ensemble members dip down to ~4 km over South America before reaching Henties Bay on September 7th but the majority of ensemble members remain above 7 km. It would be surprising if the South American smoke were lofted that high.

**Yes, the majority of isentropic back trajectories remains above 7 km over South America, but some are below 5 km AMSL and may catch biomass burning aerosols. The lofting of aerosols up to 7 km is possible, but difficult to verify from space-borne observations because of the large cloud cover over South America during the event. We do see evidence of biomass burning aerosols being transported from South America to southern Africa in**

**the CAMS analyses as well as in MODIS imagery. In the CAMS simulations the cross-Atlantic aerosol transport from South America occurs as high 500 hPa.**

The authors do not adequately explain how they are reaching certain conclusions from the evidence presented. For example, an assertion is made that during P1 trajectories from the 3000-5000 m range show more air masses coming from the southern Atlantic Ocean, and indeed South America, as compared to the other vertical ranges, but this is does not appear to be supported by Figure 13a-c.

**This point has been corrected.**

Similarly, the direct link between the circulation and aerosol transport in Figure 14 is either not obvious or directly contradicted by the figure (in that the coherent area of high AOTs seeming to come from South America stays far south of Henties Bay).

**An explanation for this is provided below.**

One potential remedy would be to restructure the transport section as separate case studies of emblematic days during the three periods. This should simplify things down enough to show more clearly whether the source is particularly uncertain or if there's just variation in source within the periods and, importantly, the vertically-resolved and time-resolved trajectory data that would aid in interpretation. This could be provided in addition to or in lieu of Figure 13 now, with a short, added discussion of how the days not highlighted compare. I'm open to other possibilities as well, but feel that the analysis as-is does not merit publication.

A final major issue has to do with figure quality. Many of the scales are all but illegible based on their small size and low resolution (see, e.g., the color bar in Figure 13 and the wind barbs in Figure 14). One or preferably both issues should be addressed prior to publication.

**We have reviewed the structure of Section 5 following the reviewer's remarks. An example (Figure 16) was given to show the altitude of the trajectories. We chose the day with the largest AOT between 5 and 6 km AMSL.**

*Specific comments:*

1. Page 2, Line 54: Rather than claiming that stratocumulus are the most effective at reflecting sunlight (deep convective clouds have higher albedo but smaller net radiative effect due to compensating longwave heating), it would be more accurate to say something along the lines of: "marine stratocumulus are particularly sensitive to aerosol perturbations due to relatively low background aerosol concentrations (Oreopoulos and Platnick, 2008)"

**We have modified the sentence including the reference proposed by the reviewer.**

2. Page 8, Lines 180-181: It makes sense that the AOD retrievals over the ocean surface will be more certain than those over land, but this still doesn't address any issues relating to the lack of co location. Unless the sea breeze is acting uniformly from the surface to ~5 km, it is plausible that the AOD over the ocean may differ somewhat from that at Henties Bay. I would think the effect is small, but it may be worth mentioning as a source of uncertainty regardless.

**It is difficult to check the level of homogeneity based on AOD observations only, but this is easier to do using CAMS reanalyses. In Figure 2a, CAMS analyses evidence that the AOD is homogeneous at a scale of ~75 km in the area of Henties Bay. The effect of sea**

breeze is included in CAMS. This scale is coherent with the size of the selected area for the MODIS retrieval analysis and gives us confidence that uncertainty associated with the non-co location is small.

3. Page 8, lines 213-215: Although no precipitation was observed at Henties Bay (which is unsurprising), it is possible that wet scavenging could have occurred closer to the source of the emissions, depleting the aerosol plume before that air was transported to the Henties Bay area.

**Agreed. This point is now better emphasized.**

4. Page 10, lines 313-314: I'm not sure that "pyro-convection" accurately describes the strength of the primarily anthropogenic plumes in the region. Also, is there significant burning in the Etosha Pan itself?

**Yes, there is a significant number of fires in the Etosha region, and more generally in the north-eastern part. Fires can be observed on Figure 14a. We now used "convection" instead to be more general.**

5. Page 22, line 444: Which altitudes? You're referring to 3000-5000 m here, right? As written, it sounds like you're referring to the full 1500-6000 m column. 6. Page 22, line 450: It is very hard to tell from the figures that the trajectories are turning counterclockwise. Perhaps some kind of composite trajectory would be useful? I'm thinking of the analysis in Adebiyi & Zuidema (2016), Figure 17, as inspiration here. Then you could address the altitude of the trajectories as well, which is not possible to do in the current format and could be important.

**The direction of rotation can still be seen on the figures. What is missing in the text is the origin of the air masses and this has been added. To make the text clearer, the sentence is removed. In order to make the figures more readable they have been enlarged and split into 3 panels. For the altitude of the retro trajectories above the source areas in the figures, it will not be readable enough with the number of trajectories considered. We fully agree that this is an important element and we have added this information to the text.**

7. Page 22, lines 454-455: I don't see how you're concluding that the 3000-5000 m level is "mainly" influenced by air from over the southeast Atlantic as compared to the other two vertical ranges.

**Agree. The correction has been done.**

8. Page 22, lines 455-456: Some of the ensemble members show the starting location as southern Brazil, but others appear to disagree. It's plausible that the origin was around Brazil, but the trajectory analysis doesn't show that Brazil *was* the origin. Especially when you've gotten out to 6 days, the HYSPLIT trajectories need to be taken with a generous helping of salt.

**It is true that back trajectories over 6 days may be associated with significant positioning errors. For this reason, the "ensemble" mode is used, and several altitude levels are considered in our study. We have chosen to analyse the problem statistically in order to have a more synthetic view. The meteorological situation is complex and fast evolving in the studied area, resulting in myriad of different transport routes. The sentence « This statistical approach, which uses the "ensemble" mode, makes it possible to consider the dispersion of back trajectories that can be linked to complex atmospheric circulations." has been added in Section 5.2.**

9. Page 22, line 456: The aerosol plume is located between 1500-3000 m during P1 according to all the other plots… why are you saying there is no aerosol in that range now? There certainly does not appear to be a plume between 3-5 km during P1, which would be the implication of this section…

**Agree. It is a mistake; the correction has been done.**

**"For the altitude ranges [5000 6000[ m no significant aerosol layer is observed by the ground-based lidar (Figure 3)."**

10. Page 23, line 462: Most of the fires are anthropogenic and set for agricultural purposes, not "wildfire." "Biomass burning regions" may be better phrasing.

**Agree, we have added "anthropogenic" and modified the text.**

11. Page 23, lines 464-646: For a given start time, the HYSPLIT ensemble provides an estimate of meteorological uncertainty, and while it may be helpful to think of some mixing of airmasses on the way to Henties Bay, a more straightforward interpretation of the ensemble may be that more oceanic and more land-based transport pathways are both plausible. It's problematic to assume that all the trajectory ensemble paths were actually followed, however. This analysis is more appropriate for a plume dispersion analysis, which can be done with HYSPLIT or with another program like FLEXPART. However, if in analyzing individual days within the period you see some with different sources, and that is the source of the wide range in the cloud of probability, that is worth reporting. If true, however, this casts doubt upon the ability to tell a coherent story about three discrete periods, as the paper currently attempts.

**The 3 periods are identified based on measurements and numerical simulation of aerosol optical property (AOT, vertical distribution). Their existence is therefore beyond doubt. Nevertheless, during each period, the meteorological situation complexity may induce a high variability in air mass trajectories, especially for the ones coming from South America. However, the contribution of biomass burning from Angola is seen to be much larger than the contribution of biomass burning South America, and to increase from $P_1$ to $P_3$, thereby providing backbone information for the definition of the 3 periods. We have insisted more on this point.**

12. Page 23, line 473-474: It would be really helpful to somehow indicate times on a figure. Again, see the note about a composite trajectory above. You could even group ensemble members with similar paths together to make things clearer on the map.

**It is very difficult to draw the location, time and altitude on the same figure of back trajectories. With the number of trajectories needed to limit uncertainties and explore all possibilities, this will quickly become unreadable. Moreover, drawing back trajectories instead of making a histogram also leads to unreadable figures and therefore difficult to interpret. Arrival times are given in the text and have been revised to better integrate penetration into South America.**

13. Page 23, line 476: Certainly most trajectories aren't coming from the south below 5000 m. Also, it's not clear that the amount of trajectories coming from South America differs substantially between P2 and P3, with P2 perhaps seeing even more from South America.

**When comparing Figures 13cfi, there are more trajectories from the south for the period 3 [5000 6000[ m. The new figures are 13c, 14c and 15c. Perhaps it was the incorrect labelling of the initial Figure 13 that caused the confusion.**

14. Page 23, line 478: The analysis presented does not establish that the highest AOTs on these days are associated with biomass burning from South America.

**Maybe the idea was not so well expressed in English. We have rephrased the sentence (see Section 5.3).**

15. Page 23, line 484: Why is the cloud cover "important"?

**We meant ubiquitous. We have replaced "important" by "ubiquitous".**

16. Page 23, lines 496-497: What is the evidence that the temporal variability of South American transport "appears" linked to the SAM? Are you simply saying this is a plausible explanation because the SAM is generally important, or is the analysis above evidence in support?

**Here, we give a hypothesis to explain the mode of transport from South America considering previous works. The reference is given in the following sentence.**

17. Page 24, lines 521-523: The relationship between transport patterns and the SAM is plausible, and appears to be very easily testable given datasets like ERA already discussed by the authors. The authors should test this claim (that different SAM phases correspond to their periods P1-P3) if they want to report it. If the authors are unable or unwilling to provide further support for their linkages between aerosol transport and the SAM, it would be best not to include this section at all or merely mention it as an avenue for future research.

**The transport pattern appears in Figure 16 where the wind field is also given. Our objective is not to provide a definitive conclusion because the measurement campaign took place over a very limited time period. Longer-term measurements should be continued, knowing that lidar profiles are necessary to properly verify the presence of aerosols at higher altitude. We have added the sentence "However, further studies are needed to support this conclusion, which will certainly have to be based on longer observation periods involving lidar technology.".**

18. Page 25, Figure 13: It would be immensely helpful to the reader to have better labeling here, perhaps for both altitude and period (labeling columns/rows would be fine). It could also be helpful to include an indication of whether smoke was present at the given altitude for each period.

**The revised Figure 13 has been split in 3 full-blown figures (Fig. 13, 14 and 15, one for each period) to comply with the reviewer's request. The altitude at which trajectories overpass the biomass burning areas is indicated in the text and Figure 16 has been added to illustrate the possible source of South America.**

19. Page 26, Figure 14: I don't see how the AOT in this figure supports your conclusions. If anything, it appears to show the South American-linked AOT stays south of 30 S.

**In our conclusions, we make the assumption, which we consider reasonable, that at least some of the aerosols observed above 5 km AMSL may originate from South America. In**

**Henties Bay, using the lidar measurements, we show that the AOTs concerned are quite low compared to the total AOT, and only seen during P3 (see Figure 2b), the largest part of the observed AOT over Henties Bay being associated with aerosols from the African continent. In Figure 2b, there are maximum values of AOT between ~0.2 and 0.3 at 355 nm. This corresponds to MODIS-derived AOT of the order of 0.15 at 550 nm. Given the color scale in Figure 14 (now Figure 17 in the revised MS), we would therefore be in the bluish part of the figure and the aerosols transported from South America to southern Africa are not so easy to distinguish without considering the wind direction.**

20. Page 26, lines 548-549: It doesn't really make sense to say that this is the first time biomass burning aerosols were characterized by lidar at Henties "during the different periods of transport" — it's the first time, period, that (ground) lidar-based characterization was possible. That there were three transport periods is a separate idea (and the division into three periods is an interpretation of the data, not a direct observation).

**Agree. We have revised the sentence. Note that the division into 3 periods is a direct interpretation that comes from observations, and even from modelling.**

21. Page 27, line 570: No evidence is presented that the transport regimes the authors associate with periods P1-P3 are the "main transport regimes across the Atlantic Ocean."

**We have changed "main" to "different…".**

22. Page 27, line 577: I still think it is overstating the case to consider a 10-15% contribution to column loading seen on two days out of three weeks of observation as "necessary" for realistic simulation of the region. Or is this meant to refer to better constraints on the aerosol column more generally? In context, it appears to refer specifically to the South America-related results.

**We have removed this sentence.**

**Anonymous Referee #2**

I do have a lingering concern that some of the statements interpreting the lidar results, in particular, the lidar ratio, are over-interpreted. If my understanding of the retrieval technique is correct, the attenuated backscatter profiles are retrieved to extinction using (1) in the free troposphere, the lidar ratio that was assumed by CALIOP or CATS for their retrievals and (2) in the PBL, a lidar ratio selected from the small set of lidar ratio models of CALIOP or CATS, whichever one creates the best match to the sun photometer AOD. Since the lidar ratio in the free troposphere is merely assumed, it is not appropriate to discuss changes in the lidar ratio as if it is an observation. (Of course, the CALIOP and CATS algorithms do not choose it arbitrarily; it's based on other evidence, but it is not a direct observation. If some comment is to be made about it, it should be made about the evidence that was used in the lidar ratio selection, which is probably primarily the altitude of the aerosol layers and the satellite-observed attenuated depolarization.) In the PBL, since only a few lidar ratios are attempted, the precision or resolution of the lidar ratio is extremely coarse, and so the authors should likewise be very careful not to over interpret the results. If the authors would consent to make the following changes, I would appreciate it. Other than this, I would be happy to see this manuscript published.

**The algorithmic approach is indeed this. The lidar ratio above the boundary layer in the region of Henties Bay is taken from the range of those considered for the operational product CALIPSO. Uncertainties are high on this parameter (±25 sr). The determination of the LR in the boundary layer based on lidar observations will therefore not be more accurate. Corrections have been made following the rapporteur's interesting recommendations.**

257: "significant changes in LR are also observed between P1/P2 and P3 with values in the FT evolving". This should be deleted. There is no observation (only assumptions) of lidar ratio in the free troposphere. If the authors want to keep something like this, I suggest something more like "CALIPSO and CATS retrievals suggest differences in the FT aerosols between P1/P2 and P3, with more occurrence of polluted dust in P1/P2 and polluted continental or smoke in P3."

**Agreed. The correction has been done.**

259: and for the PBL, again, since the retrieval here is quite coarse, I suggest rewording to something like "In the PBL, the low value of lidar ratio required to reproduce the sunphotometer AOT is consistent with the presence of clean marine aerosols in the PBL. The higher lidar ratios required in P3 indicate the presence of other aerosol types, which may include smoke or a mixture of smoke and terrigenous aerosols."

**Agreed. Thanks for suggesting this, we have added the sentence.**

268-269: The higher lidar ratio from this retrieval is not by itself indicative of dust. Any aerosol type other than marine aerosol or mixed with marine aerosol would produce a higher lidar ratio than the clean marine case; it does not require dust to explain this. I suggest deleting this sentence and simply waiting a few lines until the depolarization ratio is discussed before bringing up the hypothesis of a dust mixture.

**Agreed. The sentence has been removed.**

893: I believe this description is still somewhat confusing and I think the Author Response made it clearer. Consider replacing "obtained" with "selected from the discrete set of lidar ratios shown in Table 2" (if, in fact, this is an accurate description of the procedure).

**Agreed. The correction has been done.**

896: consider replacing "are representative of" to "are associated with" and delete "and consistent with the LRs from CALIOP for these aerosol types (compare Table 3 and Table 2)". If the lidar ratio were derived to a precision of a few sr, finding the lidar ratio to match the expected values for clean marine would be evidence of the presence of clean marine aerosol; however, since only a few lidar ratios are attempted, this evidence is rather weak. The statement that they are consistent with lidar ratios for CALIOP types seems to be merely redundant, since the retrieval used only those lidar ratios that correspond to CALIOP aerosol models (if I understand the response in the Author Response correctly).

**Agreed. The rapporteur has understood our answer. We have made the suggested corrections that we consider justified.**

[revised manuscript text omitted]

---

## Author Response (AR3)

**Dear Editor,**

**Please find hereafter the response to your comments. We thank you for thoughtful and**
**constructive proposals to improve our manuscript.**

This manuscript has improved a lot and reads very well. Below are some minor comments that are just meant to raise the level further; minor typos can be distracting to the reader and are worth removing.

abstract, line 22:- capitalize Lagrangian

**Agree. The correction has been done.**

P 2 line 37: correct St

**Agree. The correction has been done.**

P. 2 line 70: the aerosol radiative properties are held fixed in the Myhre and Stier papers. The model shortcomings are rooted more strongly in model diversity in the cloud fractions, e.g.,

Zuidema et al 2016 bams DOI:10.1175/BAMS-D-15-00082.1 ,and Stier et al. 2013, and this is what is affecting the sign of the regional DARE.

**We have added this important point and cited the reference Zuidema et al. (2016).**
p.6 line 139: analyse -> analyze

**We have chosen English UK. Also, we have used analyse consistently through the text,**

**and would rather stick with it.**

P. 9 line 256: there seems to be some sort of latex error here.

**Agree. The correction has been done.**

p. 18, line 340: evidence the->provide evidence of a 'evidence' isn't a verb and is used incorrectly in the next sentence also. And line 359.

**Agree. All the correction has been done.**

P. 18 line 363: 'the' before 'regional'

**Agree. The correction has been done.**

p. 22 line 454: check the brackets. Also on line 458. And line 463. Line 489.

**The brackets [a, b[ m are correct. This means, in the mathematical sense, that the**

**boundary b is excluded, as it is included in the next altitude range. It's to avoid counting**

**twice the same level.**

Figure 13, 14, 15: the figure labels and caption also have the same bracketing formatting problem.

**It is the same answer as the previous one.**

p. 33, lines 610-613: given that lines 600-602 indicate an AOT contribution of 10-15% from south America, the last sentence seems indulgent. I think the last paragraph is very strong without this last sentence. The authors are correct that this is the first time that the vertical structure of the aerosol has been characterized over coastal Namibia and related to transport patterns. That is enough to make this a very interesting and useful contribution. This study is not in a position to assess precipitation influences, but my suspicion is that it is difficult for aerosol from south America to survive via the baroclinic mid-latitude disturbance route. It is beyond the scope of this paper to assess and that is fine.

**Agree. The sentence has been removed.**

[revised manuscript text omitted]